# Longitudinal omics data and preclinical treatment suggest the proteasome inhibitor carfilzomib as therapy for ibrutinib-resistant CLL

Lavinia Arseni [1,16], Gianluca Sigismondo[2,3,16], Haniyeh Yazdanparast[1], Johanne U. Hermansen [4,5], Norman Mack[1], Sibylle Ohl[1], Verena Kalter[1], Murat Iskar [1], Mathias Kalxdorf [6], Dennis Friedel[7], Mandy Rettel [8], Yashna Paul [1], Ingo Ringshausen[9], Eric Eldering [10], Julie Dubois[11], Arnon P. Kater [11], Marc Zapatka [1], Philipp M. Roessner[1], Eugen Tausch[12], Stephan Stilgenbauer [12], Sascha Dietrich [13,14], Mikhail M. Savitski [8,15], Sigrid S. Skånland [4,5], Jeroen Krijgsveld[2,3], Peter Lichter[1] & Martina Seiffert [1] ✉

Chronic lymphocytic leukemia is a malignant lymphoproliferative disorder for which primary or acquired drug resistance represents a major challenge. To investigate the underlying molecular mechanisms, we generate a mouse model of ibrutinib resistance, in which, after initial treatment response, relapse under therapy occurs with an aggressive outgrowth of malignant cells, resembling observations in patients. A comparative analysis of exome, transcriptome and proteome of sorted leukemic murine cells during treatment and after relapse suggests alterations in the proteasome activity as a driver of ibrutinib resistance. Preclinical treatment with the irreversible proteasome inhibitor carfilzomib administered upon ibrutinib resistance prolongs survival of mice. Longitudinal proteomic analysis of ibrutinib-resistant patients identifies deregulation in protein post-translational modifications. Additionally, cells from ibrutinib-resistant patients effectively respond to several proteasome inhibitors in co-culture assays. Altogether, our results from orthogonal omics approaches identify proteasome inhibition as potentially attractive treatment for chronic lymphocytic leukemia patients resistant or refractory to ibrutinib.

Chronic lymphocytic leukemia (CLL) is a lymphoproliferative disorder characterized by the progressive accumulation of mature B cells in the peripheral blood, lymph nodes, bone marrow, spleen, and liver[1]. Ibrutinib, the first FDA-approved irreversible Bruton tyrosine kinase (BTK) inhibitor, is widely applied in first/second-line treatment in patients with CLL[2,3]. Although such treatment can induce remissions for many years, the development of drug resistance represents a major challenge[4,5]. Ibrutinib resistance is observed as primary resistance, the lack of initial response to ibrutinib, and secondary resistance generated after an initial response. While primary resistance is observed in 13–30% of patients, typically highly pretreated, with adverse genetic risk factors (del(17p), complex karyotype) and with an aggressive phenotype, acquired resistance is more frequent and appears normally after many years of continuous therapy with the BTK inhibitor (BTKi)[6].

The most frequent mutations responsible for acquired resistance affect *BTK* and *PLCG2*, detectable in approximately 85% of patients[7]. The molecular mechanisms underlying either primary ibrutinib resistance, or acquired resistance without such mutations, are not yet understood and their unraveling remains crucial. The BCL2 antagonist venetoclax shows efficacy in ibrutinib-resistant CLL patients, with an overall response rate of 65% and a median progression-free survival of 24.7 months[8]. But resistance to venetoclax is also occurring leaving the patients with hard-to-treat disease[9,10]. Carfilzomib (CFZ), a second-generation ubiquitin-proteasome pathway (UPP) inhibitor[11] approved for multiple myeloma by FDA and EMA, was recently identified as a potent drug that induces cytotoxicity in CLL cells before or after treatment with ibrutinib[12–17]. The UPP regulates many cellular processes including cell-cycle progression, transcriptional regulation, signal transduction, apoptosis, immune response, as well as degradation of damaged proteins, and has been clinically recognized as a therapeutic target[18]. Recognition of proteins by the proteasome is initiated by the formation of a K48-linked polyubiquitin chain[19], followed by their proteolytic degradation in the 26S proteasome by caspase-like, trypsin-like, and chymotrypsin-like activities[20]. CFZ (also known as PR-171) selectively and irreversibly inhibits the chymotrypsin-like activity of the proteasome[21–23] and differs from bortezomib, the first FDA-approved proteasome inhibitor (PI) that binds only reversibly to the proteasome. CFZ cytotoxic activity in CLL has been described previously[16,17], but its efficacy in ibrutinib-resistant patients has not been addressed yet.

In this work, we generate a CLL mouse model of ibrutinib resistance and perform comparative omics analyses to decipher the underlying mechanisms. As proteasome activity is suggested as a vulnerable target in this model, we perform preclinical treatment studies with CFZ and show its efficacy as salvage therapy in mice. Proteome analyses and ex vivo treatment studies with primary human CLL cells further verify the relevance of these findings for patients with ibrutinib-resistant CLL.

## Results

### Continuous ibrutinib treatment of TCL1-AT mice results in loss of drug sensitivity

To explore mechanisms of ibrutinib resistance in CLL, we generated a mouse model of resistance by continuous ibrutinib treatment of mice after adoptive transfer of Eμ-TCL1 leukemia (TCL1-AT mice) (Fig. 1A). Ibrutinib efficiently controlled leukemia development in TCL1-AT mice for several weeks. After 4 weeks of continuous treatment, a rise in TCL1-CLL counts in blood was observed leading to end-stage disease at week 6 (Fig. 1B). This suggested an acquired ibrutinib resistance in TCL1-CLL, similarly as observed in patients with CLL. The initial efficacy of ibrutinib was confirmed by Ki67 staining of TCL1-CLL cells showing a dramatic reduction of cell proliferation in blood, spleen, bone marrow and lymph nodes after 1 week of ibrutinib treatment (I-early) in comparison to the vehicle group (V-early) (Fig. 1C). This difference in proliferation rate was reduced when comparing TCL1-CLL cells at end-stage disease (Fig. 1D), when the number of TCL1-CLL cells reached in average $6.5 \times 10^8$ cells per spleen, which was after 3 weeks of vehicle (V-late), or 6 weeks of ibrutinib treatment (I-late). To verify that non-responsiveness to ibrutinib was intrinsic to TCL1-CLL cells, we first analyzed survival and proliferation of these cells ex vivo. To prevent spontaneous apoptosis of the cells in culture due to deprival from microenvironmental stimuli, we treated the cells with cytosine-phosphate-guanosine (CpG) as described before for human CLL cells[24]. TCL1-CLL cells from V-late mice showed a decreased survival upon ibrutinib treatment ex vivo, which was not observed in TCL1-CLL cells from I-late mice (Fig. S1A). Interestingly, CpG treatment strongly induced proliferation of TCL1-CLL cells from I-late mice, which could not be inhibited by ibrutinib in the culture setting (Fig. S1B). In contrast,

TCL1-CLL cells from V-late mice did not proliferate at all in vitro, suggesting a less aggressive phenotype of these cells.

We then transplanted splenocytes from end-stage disease mice of this first round of treatment into immunocompetent C57BL/6 mice that then underwent an analogue treatment regimen (Fig. 1A, 2nd treatment round). This confirmed an efficient but transient control of disease by ibrutinib in mice that were transplanted with V-late TCL1-CLL (Fig. 1E). In contrast, mice transplanted with I-late TCL1-CLL cells did not show any response to the drug (Fig. 1F), thus corroborating the hypothesis of a TCL1-CLL cell-intrinsic resistance mechanism.

In order to exclude that the observed ibrutinib-resistance in TCL1-AT mice is a tumor-specific feature, we repeated these experiments with TCL1-CLL cells from two further independent primary Eμ-TCL1 mice and replicated the results described above performing again two rounds of treatment (Fig. S1C–H). Altogether, our results show that TCL1-CLL cells acquire resistance to ibrutinib, an effect that was observed with all tumor samples analyzed.

### Ibrutinib resistance in TCL1-AT mice is not driven by genetic mutations but likely associated to phenotypic adaptations

To achieve a detailed characterization of TCL1-CLL cells under ibrutinib treatment and upon resistance, we sorted splenocytes from the 1st treatment round by flow cytometry to reach a purity of above 97% TCL1-CLL cells (Fig. S2A) and performed exome, transcriptome and proteome analysis (Fig. 2A). For the early, ibrutinib-sensitive state, we used samples from week 1 after treatment start. For the late, ibrutinib-resistant state, week 6-samples of ibrutinib-treated and week 3-samples of vehicle-treated mice were used, as these time points reflected a similar stage of disease as reflected by CLL counts in the spleen (Fig. S2B). By analyzing the tumor clone composition and the mutational landscape of the ibrutinib-resistant cells by WES we identified, according to SIFT, deleterious variants in *PIK3CD* (4:149658400 A/T) and *VMN2R37* (7:9206799 G/C). These variants were recurrent in all three ibrutinib-resistant samples, but detected only with low allele frequencies (<12.5%) (Fig. S2C, D and Supplementary Data 1). The protein encoded by *PIK3CD* gene is the phosphatidylinositol-4,5-bisphosphate 3-kinase catalytic subunit delta (PI3Kδ), which is involved in B-cell receptor (BCR) signaling. *VMN2R37* gene encodes the vomeronasal 2 receptor 37, a putative pheromone receptor, unrelated to immune responses. Supported by reports arguing for additional mechanisms of ibrutinib resistance besides mutations within the BCR signaling pathway[25], which often appear in low allelic frequency, we aimed to explore further novel potential targets beyond BCR signaling. Therefore, we hypothesized that transcriptional adaptations occur in CLL leading to ibrutinib resistance. Transcriptome analysis followed by unsupervised cluster and principal component analyses showed that mice responding to ibrutinib (I-early) clustered separately from the vehicle-treated and ibrutinib-resistant (V-early, V-late, I-late) groups (Fig. S2E). This is in line with the observation that I-early cells do not proliferate (Fig. 1C) or expand as a consequence of BTK inhibition. This further shows that loss of responsiveness to ibrutinib in I-late cells is associated with a considerable transcriptional adaptation. We identified differentially expressed genes between all 4 groups of samples with the largest fraction being significantly deregulated in the ibrutinib-resistant (I-late) group (Fig. 2B and Supplementary Data 2). These genes were associated to pathways of DNA synthesis, repair mechanisms, cell cycle and immune cell signaling, likely reflecting the high proliferative potential and state of the cells. Interestingly, the second most deregulated signature was associated with the proteasome feature (Fig. 2C), thus suggesting a major contribution of proteasomal processes in ibrutinib resistance.

In parallel to transcriptomics, we also performed LC-MS/MS-based proteomic analysis of FACS-sorted murine TCL1-CLL splenocytes (Figs. 2A and S2F). In brief, proteins were subjected to tryptic

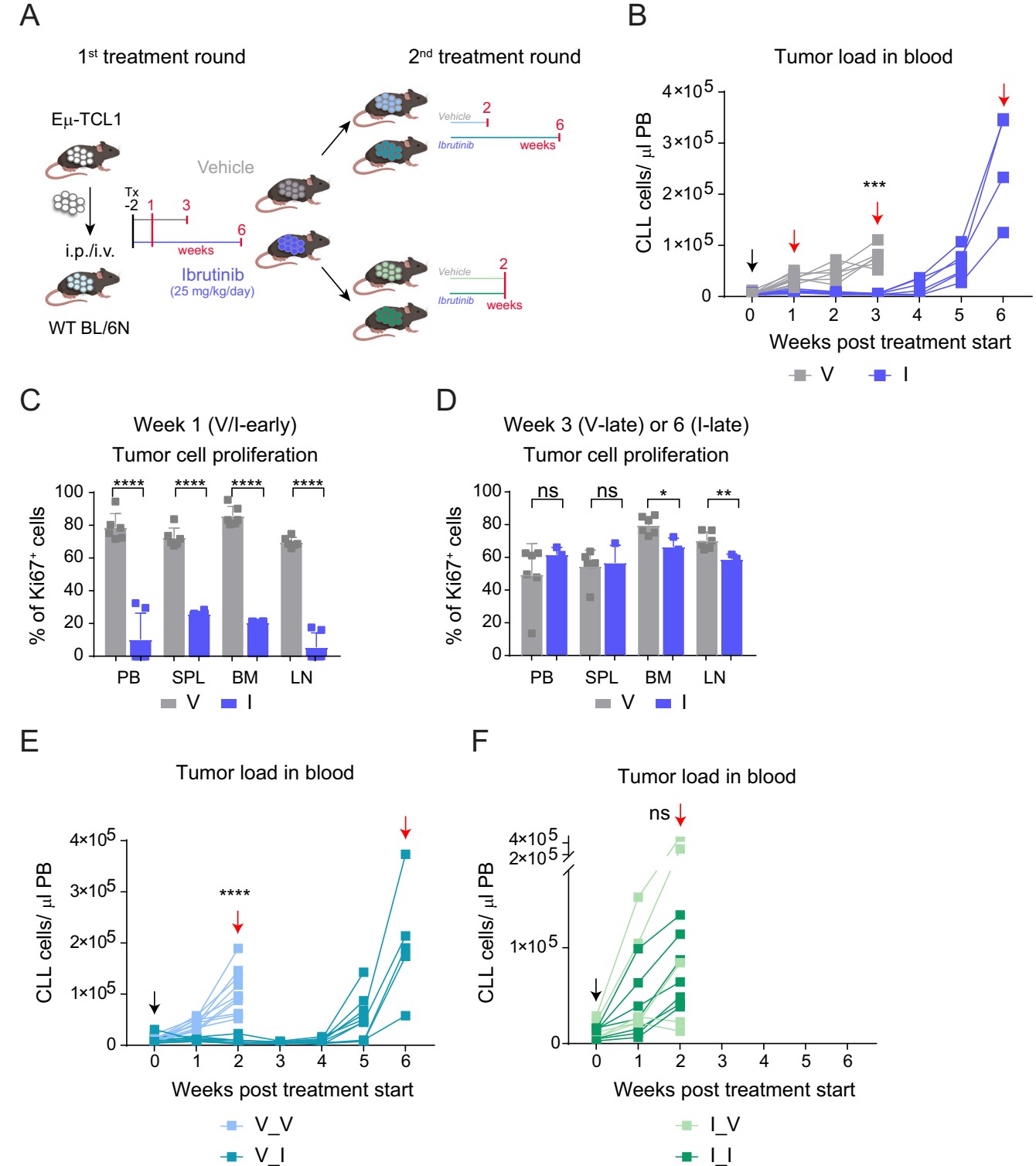

**Fig. 1 | Prolonged ibrutinib treatment of TCL1-AT mice results in cell-intrinsic resistance development. A** Experimental design and treatment schedule of in vivo assay. Created in BioRender. Zapatka, M. (2025) https://BioRender.com/a06v191. **B** Absolute numbers of CD45⁺CD5⁺CD19⁺ cells in the peripheral blood from mice adoptively transferred with malignant splenocytes isolated from leukemic Eμ-TCL1 mice (TCL1-AT; first treatment round). Mice were randomized 2 weeks post-transplantation ( = week 0) when tumor load was >5000 in the peripheral blood and treatment was started (black arrow) with vehicle (V, $n$ = 14) or ibrutinib (I, $n$ = 12). ***$p$ = 0.0003. **C, D** Percentages of Ki67-positive cells out of CD5⁺CD19⁺ cells in peripheral blood (PB), spleen (SPL), bone marrow (BM), and inguinal lymph nodes (LN) from TCL1-AT mice treated with vehicle (V, $n$ = 6, gray bars) or ibrutinib (I, $n$ = 6, blue bars) for one week (left), or three weeks for vehicle (V, $n$ = 6) and six weeks for ibrutinib (I, $n$ = 3) (right). PB: ns = 0.1842; SPL: ns = 0.7919; BM: *$p$ = 0.0256; LN: **$p$ = 0.0062, ****$p$ < 0.0001. Absolute numbers of CD45⁺CD5⁺CD19⁺ cells in peripheral blood of mice injected with vehicle-treated tumors (**E**) or ibrutinib-resistant tumors (**F**) and treated with vehicle or ibrutinib (second treatment round); ns = 0.2712, ****$p$ < 0.0001. Group sizes: V_V (light blue): $n$ = 10, V_I (dark blue): $n$ = 10, I_V (light green): $n$ = 7, I_I (dark green): $n$ = 6. Animals were sacrificed when reaching the moribund status (red arrows). Data in C and D are mean ± SD and were analyzed by two-tailed unpaired Student's $t$ test.

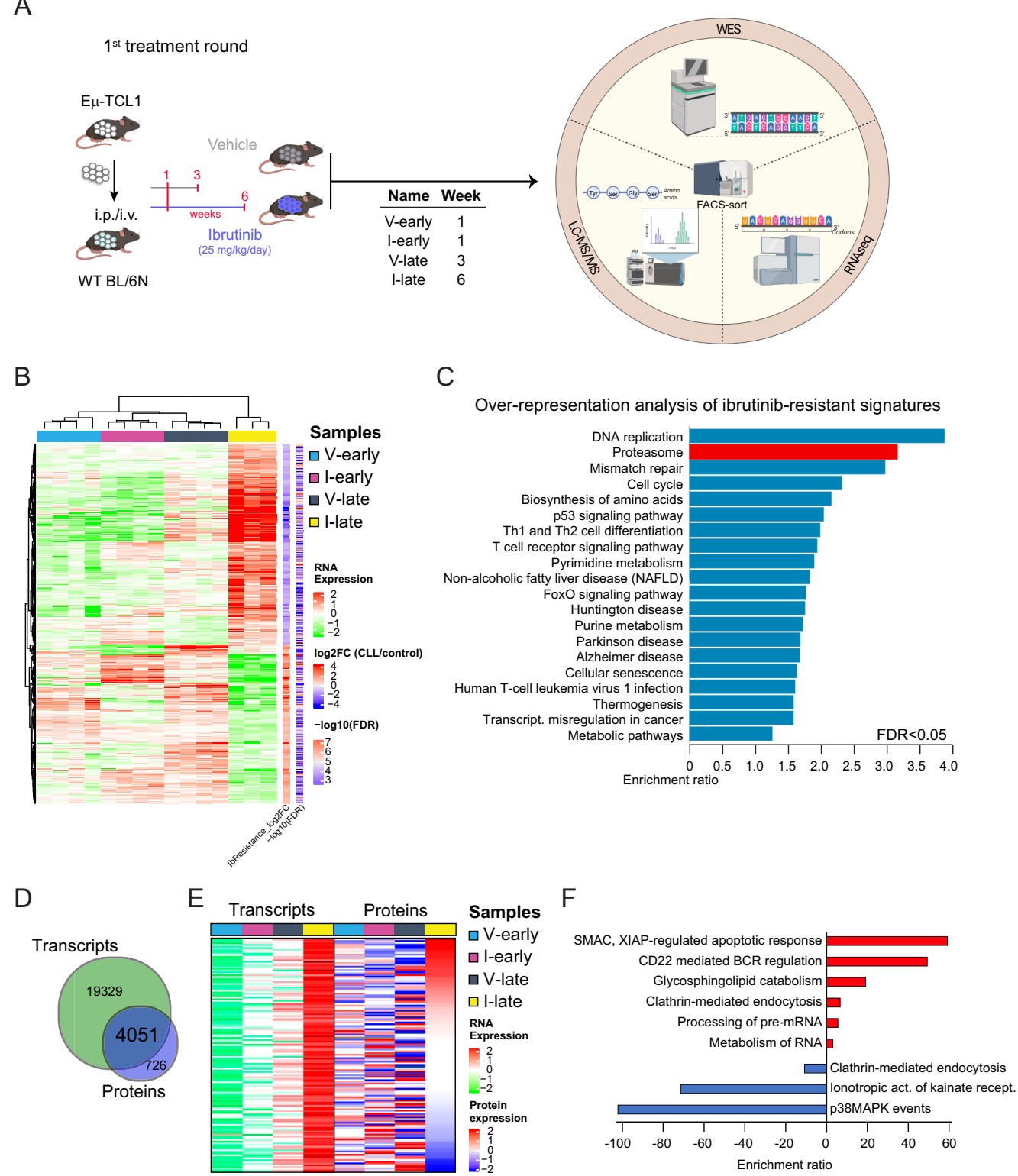

**Fig. 2 | Ibrutinib resistance is not driven by genetic mutations but likely related to post-translational modifications.** **A** Experimental design and treatment schedule of in vivo assay and overview of WES, RNA-sequencing and proteomic approaches on FACS-sorted murine CLL splenocytes. Created in BioRender. Zapatka, M. (2025) https://BioRender.com/a06v191. **B** RNA-sequencing of FACS-sorted murine CLL splenocytes. Unsupervised hierarchical clustering of the most variable transcripts comparing Ibr resistance (I-late, $n = 3$) vs Ibr sensitive (I-early, $n = 4$), and vehicle groups (V-late, $n = 4$ and V-early, $n = 4$) using LIMMA tool. **C** Over-representation analysis of significantly deregulated (FDR < 0.05) ibrutinib-

resistant-specific transcripts showing enriched terms. **D** Venn diagram showing the overlap between robustly detected transcripts and identified proteins. **E** Heatmap visualization of matched pairs between transcripts found either significantly up-(up) or down-regulated (down) in ibrutinib-resistant samples and the respective proteins, and ranked by abundance in the ibrutinib-resistant proteome. **F** Over-representation analysis of transcripts up-regulated in ibrutinib-resistant mice and showing normalized protein abundance in log 2 higher than 0.5 (red) or lower than −0.5 (blue). Source data are provided as a Source Data file Fig. 2.

digestion and peptide TMT labeling, followed by deep sample fractionation and LC-MS/MS analysis (Fig. S2F). We quantified roughly 5000 proteins in all samples (Fig. S2G and Supplementary Data 2), showing high correlation both at intra- and inter-group levels (Fig. S2H). Density distribution of peptide intensities showed a very similar distribution among the samples (Fig. S2I), thus allowing a robust direct comparison among the different conditions. As already observed at the transcriptome level, PCA of proteomic data showed a separate clustering for ibrutinib-responsive I-early samples (Fig. S2J), whereas ibrutinib-resistant I-late cells resembled vehicle-treated V-late cells. Multi-omics integration analysis of transcriptome and proteome data identified more than 4000 matched transcript-protein features (Fig. 2D and Supplementary Data 2). Analysis of features upregulated in ibrutinib-resistant cells at both transcriptional and translational level indicated increased CD22-mediated BCR regulation, which likely contributes to the lack of responsiveness to BTKi (Fig. 2E, F). As expected, our analysis indicated how transcriptional deregulation can be translated to significant differences in protein levels (Fig. 2E), therefore making proteomics a more accurate and suitable analysis to characterize the molecular mechanism driving ibrutinib resistance.

### Proteome analysis of TCL1-CLL splenocytes suggests an altered proteasome activity in ibrutinib-resistant tumors

Dimensionality reduction uMAP display of proteomic data of the TCL1-CLL cells showed a clear separation of samples into two clusters based on the disease stage (Fig. S3A), thus suggesting a different protein composition in TCL1-CLL cells between early and late state of disease. Unsupervised hierarchical clustering and heatmap visualization of proteomic data suggested that ibrutinib early samples have a distinct proteome expression profile, thus corroborating PCA visualization of transcriptome and proteome data (Figs. S2E and S2J). This unique pattern is likely linked to the reduced proliferation rate of TCL1-CLL cells upon ibrutinib treatment (Fig. 1C). Thorough investigation of proteins quantified across the four groups of samples (Fig. 3A and S3B) allowed us to identify a set of proteins specifically deregulated either in the ibrutinib-resistant (I-late vs V-late; Fig. 3B) or in the ibrutinib-responding (I-early vs V-early; Fig. 3C) groups by comparing vehicle- and ibrutinib-treated samples at a similar disease stage. Interestingly, both comparisons identified an enrichment of proteins involved in post-translational regulatory mechanisms. This analysis allowed us to define different dynamic clusters of proteins (Figs. 3D and S3C). We focused on clusters more prominently deregulated in the ibrutinib-resistant group (clusters 1, 8, 12 and 5). Cluster 1 consists of proteins massively downregulated in ibrutinib-resistant cells compared to the other groups. Proteins in cluster 8 follow an opposite dynamic profile in early and late stage of disease with an increase in vehicle-treated late vs early samples, and a decreased abundance in ibrutinib-resistant vs I-early cells. Respective GO analyses suggest that ibrutinib-resistant cells are less actively involved in immune response and protein modification (Fig. 3D). We further identified several clusters of proteins that showed higher abundance in ibrutinib-resistant vs I-early cells, including cluster 12 and 5 (Figs. 3D, and S3B, C). Whereas cluster 12 proteins are considerably downregulated with disease stage in vehicle-treated mice, but not with ibrutinib treatment in sensitive cells, cluster 5 proteins are downregulated at early stage ibrutinib-treatment (sensitive cells) which is reversed in the late ibrutinib group (resistant cells). Remarkably, in line with our transcriptional analysis (Fig. 2), GO classes associated with these two clusters are enriched in protein ubiquitination and ubiquitin proteasome pathway regulation, thereby corroborating our results and supporting the evidence that ubiquitination and proteasome regulation play a role in ibrutinib-resistance. To validate this observation, we investigated the level of ubiquitin in V-late and I-late cells, and observed that K48-linkage polyubiquitin is specifically reduced in ibrutinib-resistant TCL1-CLL cells compared to vehicle-treated cells (Fig. 3E). Interestingly, a similar treatment scheme with the BCL2 inhibitor

Venetoclax that resulted in therapy resistance of TCL1-AT mice after a comparable duration of treatment[26], elicited deregulation of proteins with different functional roles (Fig. S3D), thereby emphasizing the specificity of our findings for ibrutinib resistance.

### Proteasome inhibition by carfilzomib is an effective treatment for ibrutinib-resistant TCL1-CLL in vivo

In view of our multiomic results and based on previous clinical studies showing PIs to be effective in several lymphomas as well as in relapsed CLL patients[15,16,27–29], we tested the irreversible PI carfilzomib in the TCL1-AT mouse model. Sorted TCL1-CLL splenocytes isolated from ibrutinib-naïve or ibrutinib-resistant mice were adoptively transferred in C57BL/6 mice and after leukemia onset, mice were treated with CFZ or vehicle for three weeks (Fig. 4A). Quantification of TCL1-CLL cells in peripheral blood (PB) showed that the CFZ was able to significantly reduce the burden of ibrutinib-naïve and -resistant leukemic cells during disease course indicating its efficacy independently of ibrutinib-sensitivity (Fig. 4B). The reduction in tumor load was also evident in spleen and lymph nodes (Figs. 4C, D and S4A, B) and by the accordingly reduced spleen and liver size in the CFZ-treated mice (Fig. 4E, F). The infiltration of TCL1-CLL in the bone marrow was reduced or very low in CFZ-treated mice (Fig. S4C). Despite higher counts of TCL1-CLL cells observed in vehicle-treated ibrutinib-resistant compared to naïve tumors in blood and lymphoid tissues, CFZ treatment reduced the counts to similar levels in both treatment groups, indicating a more efficient control of disease in ibrutinib-resistant mice.

Overall our results show that in vivo treatment with CFZ is effective in the TCL1-AT model and its efficacy is even stronger in ibrutinib-resistant tumors, as the enhanced growth of these tumors is controlled to the same degree as for naïve tumors.

### Proteome analysis of CFZ-treated TCL1-CLL reveals patterns specifically altered in ibrutinib-resistant cells

To characterize the effect of CFZ on the proteome in ibrutinib-resistant TCL1-CLL, we performed global proteome analysis of FACS-sorted TCL1-CLL splenocytes via tandem mass spectrometry (Fig. S5A). We quantified 2692 proteins common to all samples (Fig. S5B). Density distribution of peptide intensities showed a very similar distribution among the samples (Fig. S5C), thus allowing a robust direct comparison of the different samples. The investigation of robustly quantified proteins across the four groups (Fig. 5A) allowed us to identify a set of proteins specifically deregulated in the ibrutinib-resistant TCL1-CLL cells upon CFZ treatment (Fig. 5B). Over-representation analysis of significantly deregulated proteins identified DNA damage-related signatures as well as ubiquitination and apoptosis pathways among the most represented ones (Fig. 5C). Moreover, Gene set enrichment analysis of ibrutinib-resistant samples treated with CFZ or vehicle, revealed a downregulation of ubiquitin ligase complex upon CFZ treatment (Fig. 5D), thereby confirming the sensitivity of ibrutinib-resistant mice to proteasome inhibitors. A limited number of differentially regulated proteins was identified in the ibrutinib-naïve arm (Fig. S5D). This can be explained by the higher heterogeneity and protein abundance variability observed in the naïve group, compared with the resistant counterpart. This low number of deregulated proteins is mainly involved in IRF7 activation and cytokine production (Fig. S5E). As expected, a greater number of deregulated proteins derived from the comparison between ibrutinib-resistant and -naïve cells, either untreated (Fig. S5F) or upon CFZ treatment (Fig. S5G). From these lists of deregulated proteins, we identified 307 proteins specific for ibrutinib-resistant CFZ-treated mice (Fig. S5H), and their association with pathways involved in IRF7 activation, RUNX1 activity and IL-6 signaling (Fig. S5I). These results therefore suggest that CFZ treatment has a distinct effect on ibrutinib-resistant in comparison to -naïve cells, which might explain the high efficacy of CFZ in tumors that are non-responsive to ibrutinib.

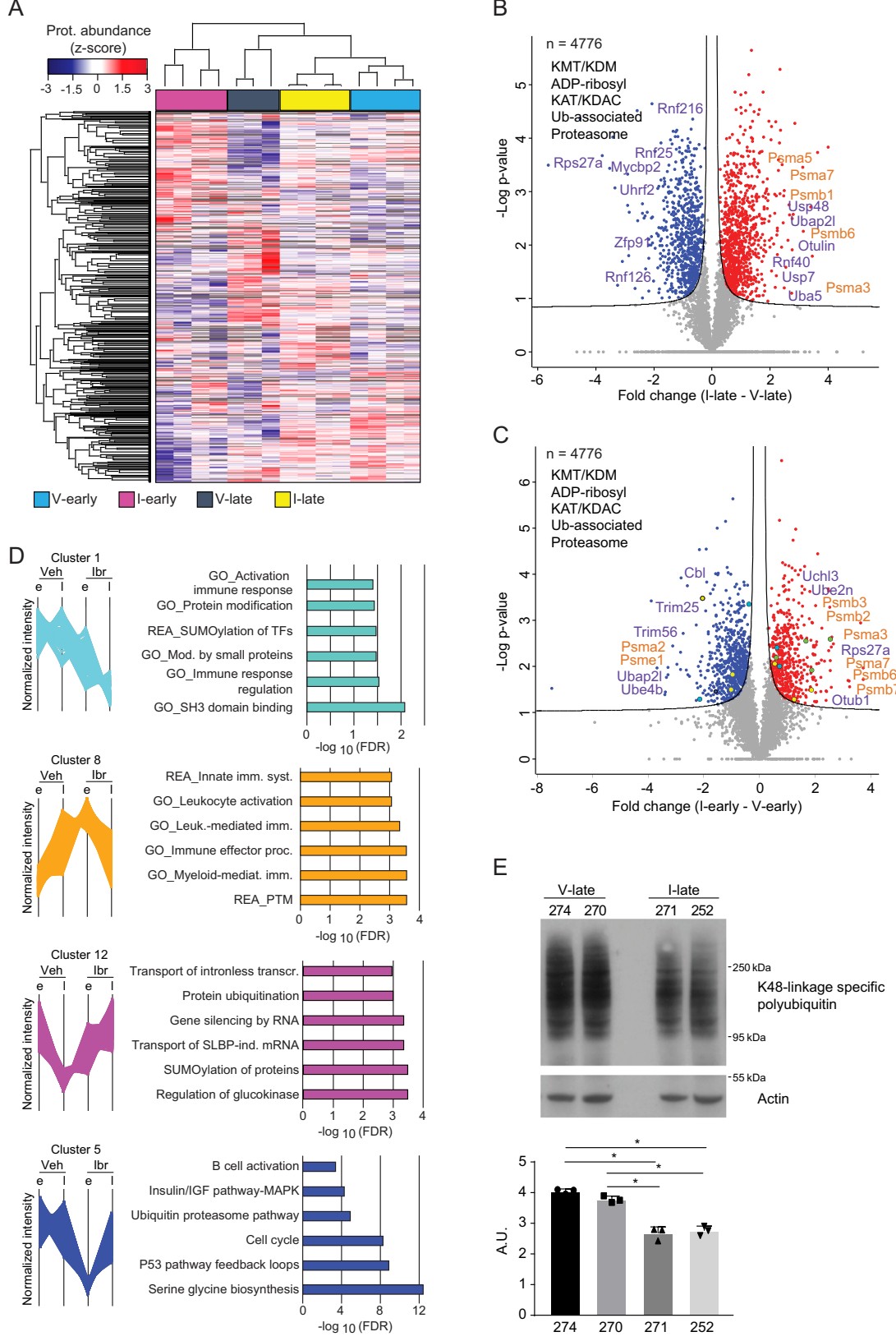

## Carfilzomib is effective as therapy for ibrutinib-resistant TCL1-CLL

In order to verify whether CFZ could help in delaying the onset of ibrutinib resistance, we transplanted ibrutinib-naïve tumors in C57BL/6 mice and after leukemia onset administered CFZ, either as mono- or combined therapy together with ibrutinib. Despite being effective as monotherapy, CFZ did not further improve disease control when simultaneously administered with ibrutinib from the beginning of the treatment (Figs. 6A and S6A), overall not resulting in a significant survival benefit compared to ibrutinib monotherapy (Fig. 6B). With the aim to investigate if CFZ acts more potently upon ibrutinib resistance onset, we performed a sequential treatment of CFZ in mice previously

**Fig. 3 | Proteome analysis of TCL1-CLL splenocytes suggests an altered proteasome activity in ibrutinib resistant tumors. A** Unsupervised hierarchical clustering of proteins reproducibly quantified in ibrutinib resistant (I-late, $n = 4$), ibrutinib sensitive (I-early, $n = 4$), and vehicle groups (V-late, $n = 3$ and V-early, $n = 4$) ($n = 5495$ proteins). Volcano plots display proteins significantly deregulated in ibrutinib-treated tumors vs vehicle-treated ones at late stage (**B**) or at early stage of disease (**C**) according to two-sided $t$ test statistics (FDR < 0.05, S0 = 0.1). The main categories of proteins are represented by colors (yellow, KMT/KDM; green, ADP-ribosyl; blue, KAT/KDAC; violet, Ub-associated; orange, Proteasome). Representative proteins associated with either ubiquitin or proteasome are highlighted.

**D** Selected protein clusters for significantly deregulated proteins in ibrutinib-resistant mice are shown, with the corresponding gene ontology signature. **E** Immunoblot analysis of K48-linkage specific polyubiquitination in PanB-enriched CLL splenocytes. Actin was used as loading control. Numbers indicate labels of mice. Experiment has been performed in triplicate and representative image is shown. Data are mean ± SD and were analyzed by two-tailed unpaired Student's $t$ test; 274 vs 271: $*p = 0.0234$; 274 vs 252: $*p = 0.0253$; 270 vs 271: $*p = 0.0261$; 270 vs 252: $*p = 0.0268$. Source data are provided as a Source Data file Fig. 3.

treated with ibrutinib. This approach revealed that, despite the intrinsic heterogeneity of the individual samples, CFZ addition after two weeks of ibrutinib treatment was indeed effective in slowing down TCL1-CLL development in mice. We observed a significant reduction in TCL1-CLL cell counts in blood two weeks after the combined treatment started, and this effect was maintained over time (Figs. 6C and S6B). Interestingly, this resulted in a significantly improved survival of mice compared to ibrutinib as monotherapy (Fig. 6D). Overall, these results show that CFZ is an effective therapy for ibrutinib-resistant TCL1-CLL in mice, thus suggesting proteasome inhibition as an effective therapeutic option for CLL patients that are refractory to ibrutinib treatment.

## Proteome analysis of patient-derived ibrutinib-resistant CLL cells reveals a common pattern in all patients

To verify the relevance of these findings for patients with ibrutinib-resistant CLL, we performed global proteomic profiling of isolated CLL cells from patient samples before start of treatment and upon refractoriness or relapse under ibrutinib. As the majority of ibrutinib-resistant patients harbor a mutation in *BTK* or *PLCG2*, our cohort consisted of 7 cases with a detectable mutation in *BTK* (BTKmut), while 2 cases had none (BTKwt). To determine if our mouse model of ibrutinib resistance accurately replicates the human disease state, with either mutant or wild-type (wt) *BTK*, we performed a correlation analysis between ibrutinib-resistant samples from mice and ibrutinib-resistant samples from BTKwt or BTKmut patients (Fig. 7A, Fig. S7A). This analysis shows a very high intraspecies correlation (i.e., 0.9) even between proteomes with different *BTK* mutational status, and a relatively high interspecies correlation of around 0.4–0.5. This result shows on the one hand that proteomes associated with ibrutinib resistance in our mouse model show a relatively good correlation with clinical samples, while on the other hand it suggests a very high correlation of proteomes from individuals with ibrutinib resistance independent of the *BTK* mutational status of the patients (Fig. 7A, Fig. S7A). Remarkably, we observed that specifically proteasome-related proteins have a significantly higher interspecies correlation compared to the entire proteome, or groups of proteins that cooperate in other functional complexes (e.g., eukaryotic translation initiation, SWI-SNF, or MAPK complex) or belong to the same organelle (e.g., mitochondrial or ribosomal proteins) (Fig. 7B, Fig. S7B). This suggests that ibrutinib resistance, both in the TCL1-AT mouse model and in patients with CLL, is associated with a similar regulation of proteasome-related proteins. Importantly, these results are valid in both individuals with BTKwt and BTKmut. In agreement with this, unsupervised hierarchical clustering segregates almost perfectly human CLL samples according to their disease state (i.e., untreated or resistant to ibrutinib) regardless of their *BTK* mutational status (Fig. 7C). This result may suggest that the mutational status of *BTK* plays an important role in the progression speed of the disease, but patients non-responsive to ibrutinib treatment might share a common proteomic program. This is in line with observations that show that mutations in *BTK* or *PLCG2* are often subclonal in ibrutinib-resistant patients[30,31].

To get further insights into the mechanism of ibrutinib resistance in human CLL, we collected PB of 3 patients at 3 sequential time points, before starting ibrutinib treatment, during ibrutinib responsiveness (within the first 6 months of treatment), and upon ibrutinib refractoriness, and performed proteome analysis via quantitative mass spectrometry. In brief, proteins from sorted human B cells were subjected to tryptic digestion and peptide TMT labeling, followed by deep sample fractionation and LC-MS/MS analysis (Fig. S7C). As a result, we quantified almost 6000 proteins (Fig. S7D, E), which showed a very high correlation both at intra- and inter-patient levels (Fig. S7F). Density distribution of peptide intensities showed a very similar distribution among the samples (Fig. S7G), thus making a robust direct comparison among the different conditions possible. Among the quantified proteins, 360 were significantly deregulated among the different conditions (Fig. 7D, and Fig. S7H) in at least two out of the three patients. Unsupervised hierarchical clustering of the 360 deregulated proteins identified a class of 181 proteins following a common profile of abundance in all three patient samples and being specifically altered in relapse (Fig. 7D, E). Half of these proteins were significantly up- and half down-regulated in relapsing patients compared to the ibrutinib responsive condition (Fig. 7F). Over-representation analysis of the downregulated proteins showed an enrichment of epigenetic and DNA damage signatures, as well as apoptosis-related signatures aligning with the increased cell death observed in ibrutinib-sensitive compared to resistant CLL (Fig. 7G, upper panel). BCR regulation was the top enriched signature in the upregulated group (Fig. 7G, lower panel), in agreement with the regained BCR hyperactivation in relapsed CLL. Gene ontology (GO) analysis of the 360 proteins showed regulation of proteolysis and protein stability among the top enriched GO terms (Fig. S7I), thereby corroborating proteome data of TCL1-AT mice showing an altered proteasome signature upon ibrutinib-resistance (Fig. 3). In line with these results, human proteome analysis revealed over-representation of proteasome-associated proteins and ubiquitin ligases in relapsing conditions (11 proteins with fold change >0.5) in comparison with the respective responsive states (2 candidates with fold change <0.5) (Fig. 7H). Despite observing different dynamics in the two categories of proteins, the majority of the candidates showed a higher abundance in relapse (Fig. 7I, J), further pointing towards a potential benefit of ibrutinib-refractory patients from PI treatment.

To test the efficacy of PIs, we treated CLL cells from both treatment naïve and ibrutinib-resistant patients ex vivo in cultures that mimic the lymph node microenvironment[32,33] with several PIs, including CFZ. In line with a case report demonstrating the efficacy of PIs in relapsed CLL after treatment with ibrutinib, idelalisib, alemtuzumab, and venetoclax/rituximab[32], all samples were sensitive to PIs, and CFZ was most potent (Fig. 7K). Responsiveness of these samples to ibrutinib could not be verified ex vivo (Fig. S7J) due to previously described small effect sizes after short term exposure of CLL cells to ibrutinib, which are in the range of 5-10%, and the expected response to treatment in vivo only after months[34]. Despite the fact that there is still no clear consensus on the activity of PIs in ex vivo cultures, mainly due to the concern that this effect would disappear in vivo, our murine data of CFZ treatment upon ibrutinib provides a solid observation in support of PIs as attractive treatment options for CLL patients not responding to ibrutinib.

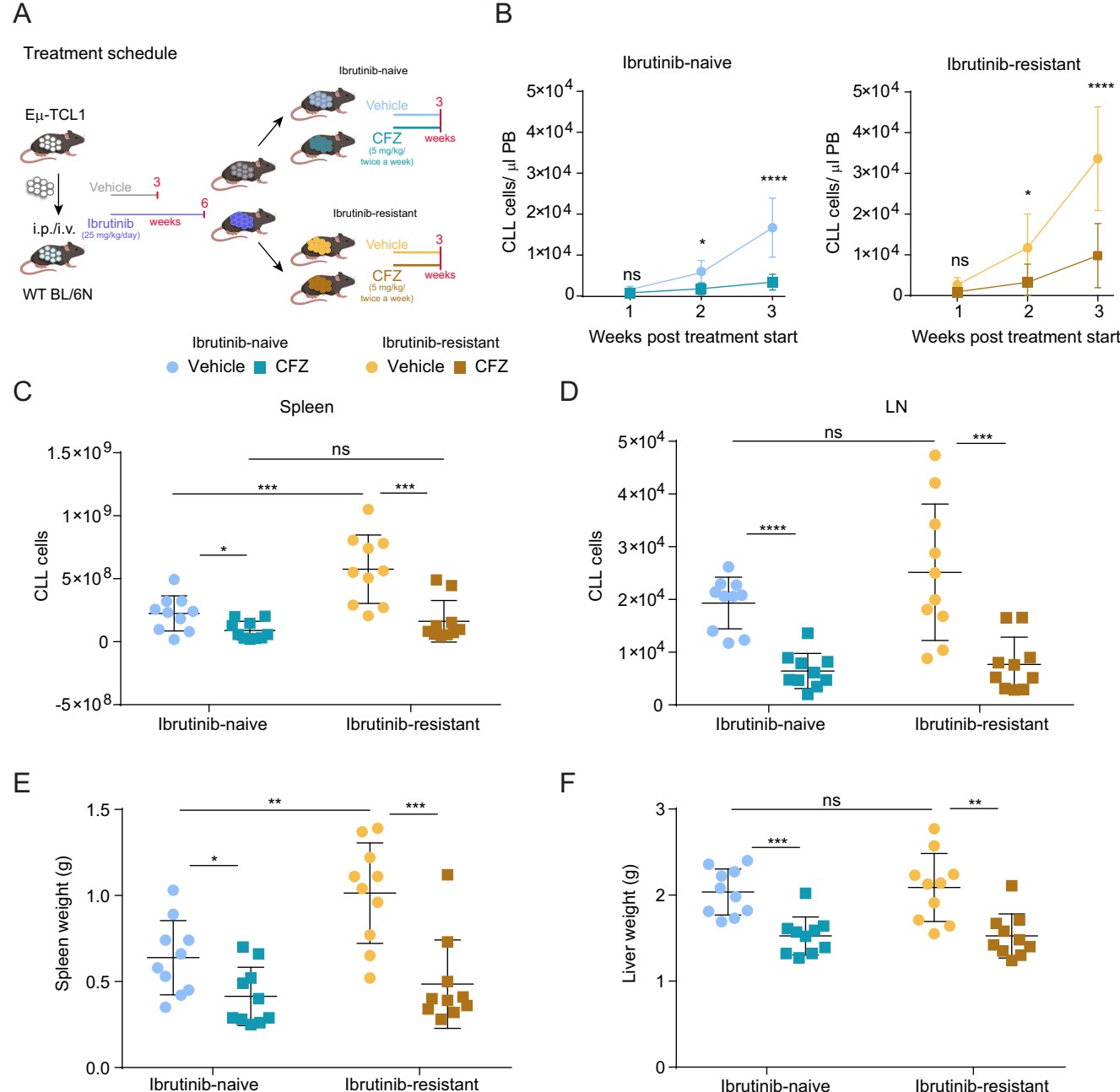

**Fig. 4 | Both ibrutinib-resistant and -naive TCL1-CLL respond to carfilzomib treatment. A** Experimental outline of CFZ treatment in TCL1-AT mice. Created in BioRender. Zapatka, M. (2025) https://BioRender.com/a06v191. **B** C57BL/6 mice were injected with either ibrutinib-naïve (left panel) or -resistant (right panel) TCL1-CLL cells. Ibrutinib-naïve: ns = 0.0535; ∗p = 0.0227, ∗∗∗∗p < 0.0001. Ibrutinib-resistant: ns = 0.0586; ∗p = 0.0282, ∗∗∗∗p < 0.0001. Absolute number of CD19+ CD5+ TCL1-CLL cells in spleen (**C**) and in the inguinal lymph nodes (LN) (**D**). Spleen:

∗p = 0.0232; ∗∗∗p = 0.0005, ns = 0.9318, ∗∗∗p = 0.0003. LN: ns = 0.3279, ∗∗∗p = 0.0002, ∗∗∗∗p < 0.0001. Spleen (**E**) and liver (**F**) weights of mice 5 weeks after TCL1-CLL transplantation. n = 10 each group. Spleen weight: ∗p = 0.0155, ∗∗p = 0.0062, ∗∗∗p = 0.0010. Liver weight: ∗∗∗p = 0.0002, ns = 0.9772, ∗∗p = 0.0012. Mice treated with vehicle are represented by circular symbols, mice treated with CFZ are represented by squared symbols. Data are mean ± SD and were analyzed by two-tailed unpaired Student's t test.

## Discussion

Since its discovery in 2007[35,36], ibrutinib went through a rapid drug development until the approval for CLL treatment in 2014. Despite its high initial response rates, resistance to ibrutinib monotherapy still represents a clinical challenge[7,37–39]. The most common and best characterized form of ibrutinib resistance is represented by genetic alteration, with mutations within *BTK* and its downstream substrate *PLCG2*, which can be found in CLL clones or subclones of variable size. Studies conducted both in vitro and in vivo on Waldenström's macroglobulinemia (WM) and diffuse large B-cell lymphoma (DLBCL)

reveal that even small subclones carrying these mutations can drive clinical resistance. *BTK*-mutant cells contribute to resistance by shielding wild-type cells from ibrutinib-induced killing, releasing IL-6 and IL-10 to activate pro-survival pathways such as JAK/STAT signaling. This mechanism may explain the clinical resistance observed in patients with *BTK*-mutated subclones[40]. The timing and frequency of these resistance-associated mutations suggest they are unlikely to be the initial drivers. Instead, several studies propose that epigenetic changes may be the initial adaptations allowing tumor cells to survive and persist during treatment with BTK inhibitors[41]. Such adaptations

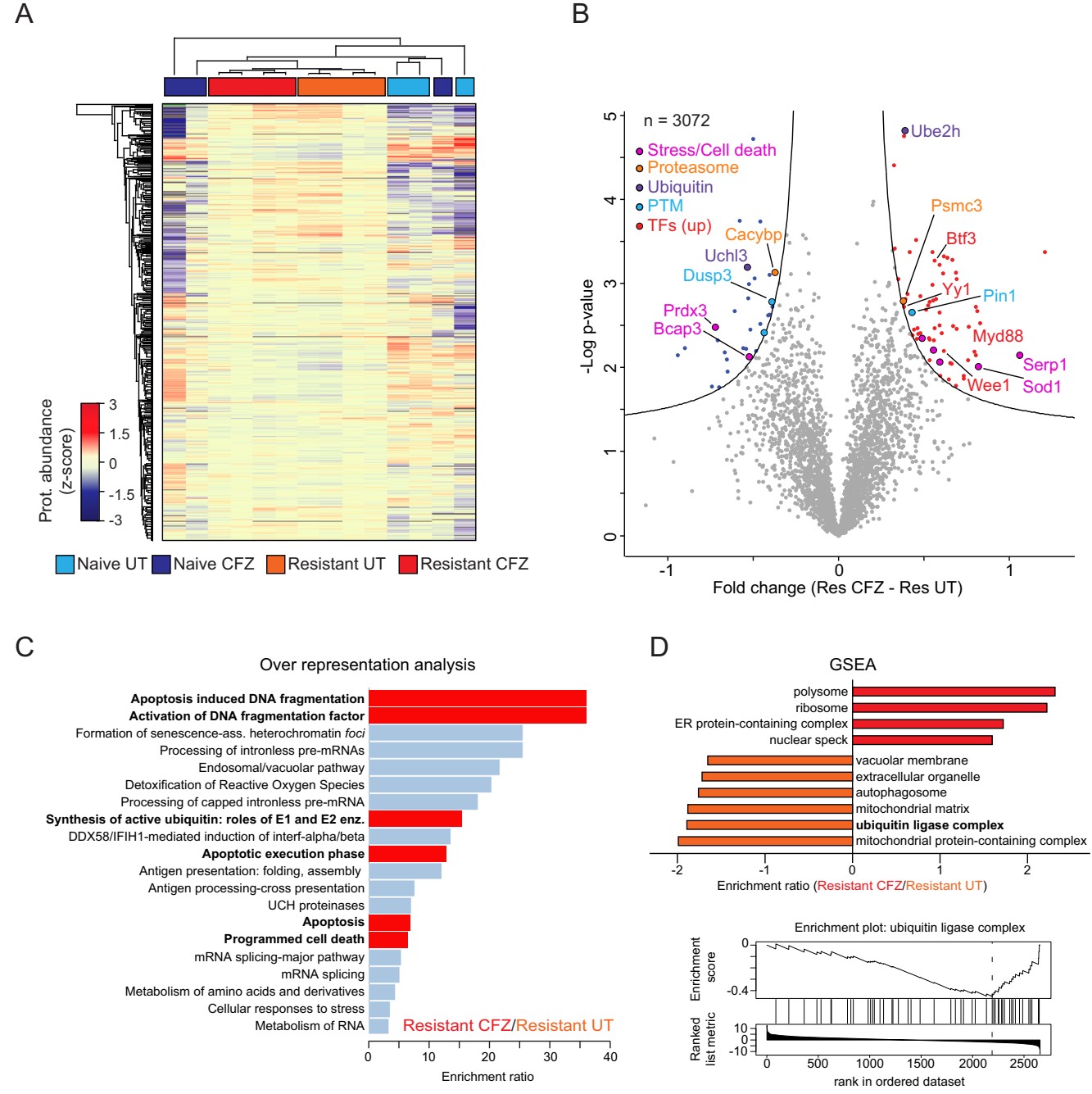

**Fig. 5 | Proteome analysis of CFZ-treated TCL1-CLL reveals patterns specifically altered in ibrutinib resistant tumors. A** Unsupervised hierarchical clustering of proteins reproducibly quantified in ibrutinib-naïve and ibrutinib-resistant TCL1-CLL treated with CFZ or the corresponding vehicle ($n = 3072$). Z-scored protein abundances are shown. **B** Volcano plot display of proteins significantly deregulated in ibrutinib-resistant TCL1-CLL treated with CFZ (Res CFZ; $n = 4$) or vehicle (Res UT; $n = 4$) according to two-sided $t$ test statistics (FDR < 0.05, S0 = 0.1). Main categories of proteins are represented by colors (pink, Stress/Cell death; orange, Proteasome; green, Ubiquitin; blue, PTM; red, upregulated transcription factors).

Representative proteins associated with the above-mentioned categories are highlighted. **C** Over-representation analysis of significantly deregulated ibrutinib-resistant proteins treated with CFZ showing enriched terms. **D** GSEA of proteins represented in B (Resistant CFZ/Resistant UT) and ranked by $t$ test statistics (top). Terms associated with proteins showing increased and decreased expression upon CFZ treatment are shown in red and orange, respectively. Enrichment plot for the downregulated ubiquitin ligase complex (bottom). Source data are provided as a Source Data file Fig. 5.

could facilitate the emergence of resistant subclones harboring *BTK* or *PLCG2* mutations. Evidence supporting this includes findings of malignant B-cell persistence without BTK or PLCG2 mutations in 15% to 43% of CLL patients[7,31,42]. In our study, the typical *BTK* and *PLCG2* mutations were absent in ibrutinib-resistant TCL1-CLL cells. Instead, we identified recurrent low-frequency variants in *VMN2R37* and *PIK3CD*. This suggests alternative mechanisms of resistance, consistent

with observations in CLL patients[25]. The molecular patterns of ibrutinib-resistant disease allow for the definition of two groups of patients. Transformation to aggressive disease predominantly occurred early after treatment start, with all cases manifesting within the first 15 months[31]. In contrast, progressive CLL emerged later, primarily after two or more years of treatment, with *BTK* and/or *PLCG2* mutations detected in 80% of these cases, consistent with their known association

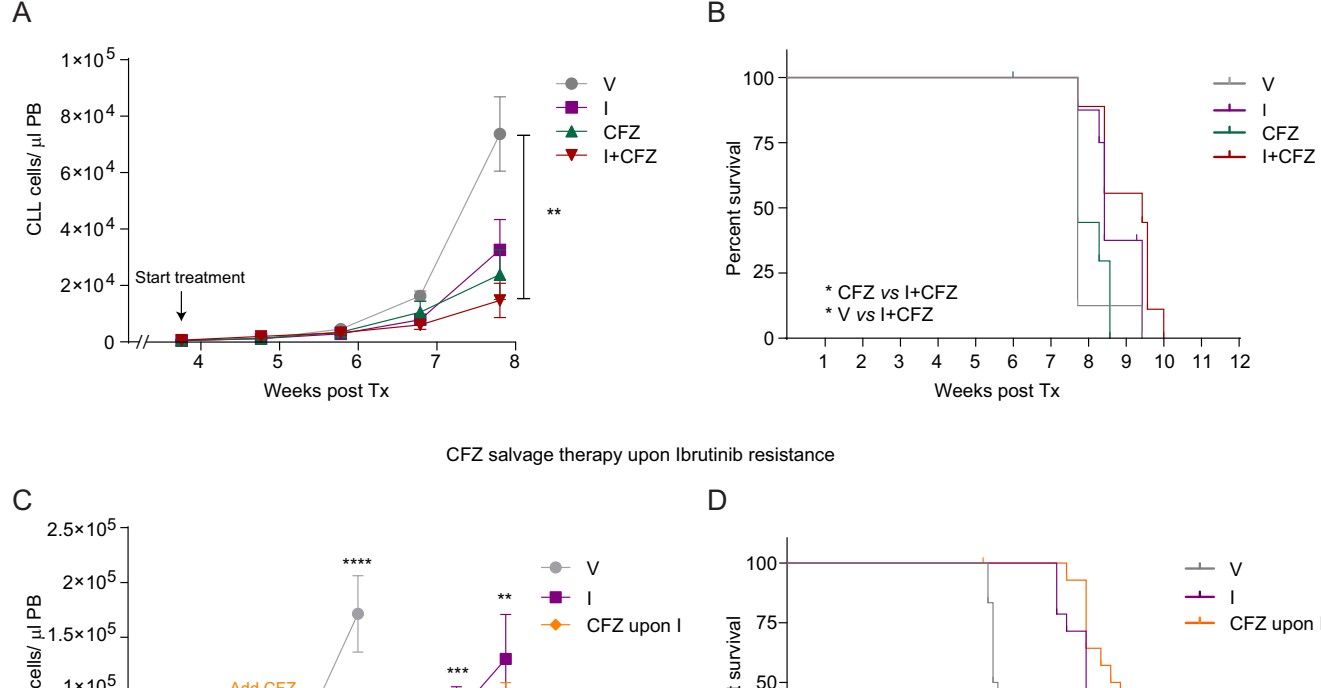

**Fig. 6 | Complementary administration of CFZ improves survival of mice developing ibrutinib resistance. A** Absolute numbers of CD19+ CD5+ TCL1-CLL cells in PB of TCL1-AT mice undergoing ibrutinib (I, $n = 8$, purple square) or carfilzomib (CFZ, $n = 10$, green triangle) monotherapy, the combined therapy (I + CFZ, $n = 9$, red upside-down triangle) or the corresponding vehicle treatment (V, $n = 8$, gray circle) over time. Data are mean ± SD and were analyzed by Kruskal–Wallis test with Dunn´s multiple comparisons test; $**p = 0.0025$. **B** Kaplan–Meier curves of the overall survival of the TCL1-AT mice described in (**A**). Data were analyzed by Mantel-Cox test; CFZ vs I + CFZ: $*p = 0.034$, V vs I + CFZ: $*p = 0.026$. **C** Absolute

numbers of CD19+ CD5+ TCL1-CLL cells in PB of mice under ibrutinib monotherapy (I, $n = 14$, purple square), the sequential administration of ibrutinib followed by CFZ (CFZ upon I, $n = 15$, orange rhombus) or the corresponding vehicle treatment (V, $n = 6$, gray circle) over time. Data are mean ± SD and were analyzed by one-way Anova with Tukey´s multiple comparison test (at 6 weeks) or with unpaired two-tailed Student's $t$ test (at 7, 8 and 9 weeks); $**p = 0.0016$, $***p = 0.0001$, $**p = 0.0039$, $****p < 0.0001$. **D** Kaplan–Meier curves of the overall survival of TCL1-AT mice described in C. Data were analyzed by Mantel-Cox test; $**p = 0.0027$, $****p < 0.0001$.

with ibrutinib resistance. The TCL1-AT model we used in this study resembles aggressive or transformed CLL with a rapid progression and short survival[43]. Therefore, the resistance to ibrutinib in this model likely resembles the mechanism of the first group of patients, which is not linked to mutations in *BTK* and *PLCG2*. Even though this concerns only a small group of patients (6–7% of ibrutinib-treated CLL patients, and 13–30% of ibrutinib resistant cases)[44,45], there is a high clinical need for these patients. Therefore, the results of our study are of particular importance to understand their mechanisms of refractoriness, and to develop alternative treatment strategies for this subgroup of patients that is hard to treat. To achieve these goals, we integrated multiple omics datasets obtained in the TCL1-AT model upon ibrutinib resistance and discovered posttranslational modifications and proteasome activity as majorly altered at the transcript and protein level. Pathway analysis identified TRAF6 and IL-6 signatures enriched in the proteome of ibrutinib-resistant mice, which is in accordance with recent studies in patients[41,46].

Among the several strategies to overcome ibrutinib resistance, second-generation BTKi like acalabrutinib and zanubrutinib[46] have been adopted and novel, third-generation, non-covalent BTKi have entered clinical trials to date, with pirtobrutinib being the most advanced in development[6]. Further, the combination of multiple

targeted agents with non-overlapping mechanisms of action, e.g., the simultaneous targeting of BCL2 and BTK with or without an anti-CD20 monoclonal antibody, is currently explored in clinical trials[47]. Avoiding constant drug exposure and selection of BTKi-resistant clones by time-limited treatment and exploiting synergism of BTKi with other therapeutic agents in killing malignant cells while minimizing toxicity and the development of resistance will likely result in improved long-term therapy responses.

Interestingly, recent research showed synergy between ibrutinib and PI in relapsed and refractory multiple myeloma[28,48] and promising results in CLL[15]. And several studies predicted a benefit when combining PI with BTKi[15,49,50]. Our preclinical studies showed that the PI carfilzomib was effective in ibrutinib-resistant human CLL samples in vitro and in the TCL1-AT mouse model where it reduced the tumor load to a similar level as observed in ibrutinib-naïve tumors. Several PIs, bortezomib, carfilzomib, and ixazomib, tested in combination with rituximab, have shown efficacy in prospective studies in WM[51]. Carfilzomib demonstrated its safety and tolerability in patients with relapsed/refractory CLL, therefore representing an interesting option as part of combination strategies[29]. Nevertheless, caution should be taken regarding its association with a dose-dependent increased risk of cardiovascular events in WM[52].

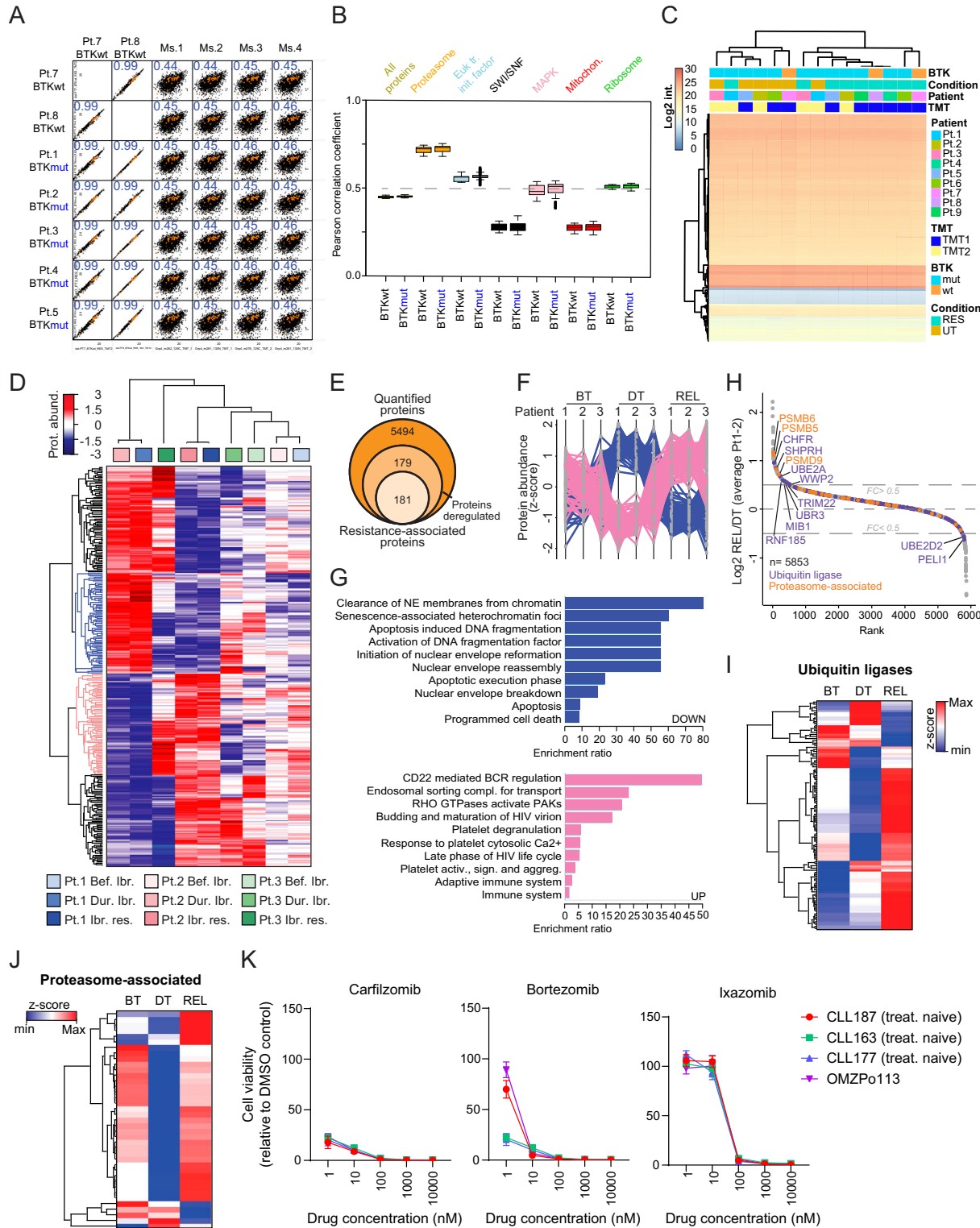

Ubiquitin-related proteins and the unfolded protein response (UPR) were shown to be prognostic in CLL, with UPR also predicted better response to BTKi[53]. In agreement, our results showed that ubiquitin alone has a low prediction power for ibrutinib responsiveness. Of note, mono- and polyubiquitination on K48 and K63 branches are associated with different biological outcomes, where K48 polyubiquitination is directly associated with proteasomal degradation. As a consequence, proteasome-related proteins likely have a higher

prediction for ibrutinib resistance, which is in line with our data, showing a drastic change in the expression of proteasome-related proteins at resistance. An upregulated UPR under BTKi treatment might be an adaptive response to address the stress of accumulating proteins, relying on proteasomal degradation. This might explain the efficacy of PI in CLL, also in patients after BTKi treatment.

By defining the protein landscape of CLL, the proteasome-related proteins PSMD9, USP5, UBE2N, and 10 proteasome subunits were

**Fig. 7 | Commonly deregulated trends in ibrutinib-resistant CLL patients revealed by proteome analysis. A** Multi-scatterplot showing the Pearson's correlation analysis of proteome data ($n = 2840$ proteins) between ibrutinib-resistant samples of the TCL1-AT mouse model (Ms.1–4, $n = 4$) and human ibrutinib-resistant samples with wild-type (wt, $n = 2$) or mutant (mut, $n = 5$) BTK. Pearson correlation coefficient as a measure of the strength of the linear relationship of the proteome of two samples each, is reported in blue. Orange dots represent proteasome-related proteins ($n = 35$). **B** Boxplot representation of Pearson's correlation coefficients between murine ibrutinib-resistant samples ($n = 4$) and human resistant samples with wild-type (BTKwt, $n = 2$) or mutant BTK (BTKmut, $n = 5$), for either all matched proteins, or proteins in specific complexes. Number of proteins: All proteins ($n = 2840$), Proteasome ($n = 35$), Euk tr. init. factor ($n = 23$), SWI/SNF ($n = 9$), MAPK ($n = 8$), Mitochon. ($n = 24$), Ribosome ($n = 86$). Center lines show the medians; box limits indicate the 25th and 75th percentiles; whiskers extend 1.5 times the interquartile range from the 25th and 75th percentiles. **C** Heatmap visualization of the unsupervised clustering of log2 protein abundance for human BTKwt or BTKmut CLL samples at either untreated (UT, $n = 6$) or ibrutinib-resistant (RES, $n = 9$) disease stage. TMT1 and TMT2 refer to two different sample batches. **D** Unsupervised hierarchical clustering of the significantly deregulated proteins in at least two out of three human CLL samples. Blue and pink clusters correspond to significantly down- and up-regulated proteins, respectively, in all three resistant samples in comparison with the other two conditions. **E** Venn diagram depicting the number of robustly quantified proteins across the three patients, the amount of significantly deregulated proteins and the ones specifically associated with resistance. Line plot display of significantly deregulated proteins in ibrutinib-resistant CLL patients ($n = 3$) are shown (**F**), with the corresponding gene signatures (**G**). Blue and pink clusters correspond to significantly down- and up-regulated proteins, respectively, in all three resistant samples in comparison with the other two conditions. **H** Intensity-based ranking distribution of relapse (REL, $n = 3$) over during treatment (DT, $n = 3$) average ratio expressed in log2. Proteins with log2 FC of at least $\pm0.5$ (represented by dotted lines) are highlighted. Ubiquitin ligases and proteasome-associated proteins are shown in violet and orange, respectively. Heatmap representation of ubiquitin ligases (**I**) and proteasome-associated proteins (**J**) across the three treatment stages: before treatment (BT, $n = 3$), during treatment (DT, $n = 3$), and at relapse (REL, $n = 3$). **K** CellTiter-Glo luminescent cell viability assay performed after 72 h exposure to proteasome inhibitors (in triplicate). Graph shows average +/- standard deviation (SD). Experiment was performed on cells from three treatment nave patients (CLL163, CLL177, CLL187) and one ibrutinib-resistant patient (OMZPo113). Source data are provided as a Source Data file Fig. 7.

shown to be upregulated in CLL patients with trisomy of chromosome 12[54]. This suggests a link between high proteasomal activity and CLL development and might explain the general efficacy of PI in CLL.

A recent study providing a rationale for the design of ibrutinib-based combination therapies combined epigenome with single-cell chemosensitivity profiling in CLL patient samples collected before and during ibrutinib therapy and showed that ibrutinib induced pharmacological sensitivity to PIs, amongst others[49]. In our proteome study on matched CLL samples before, during treatment and at relapse under ibrutinib, we observed an upregulation of proteasome-associated proteins and ubiquitin ligases in relapsing conditions in comparison with the respective responsive states, similar as observed in the murine TCL1-CLL proteome. The most represented signatures during treatment revealed increase in signaling pathways such as apoptosis, transcriptional deregulation and inflammation, and a decrease in VEGF signaling and DNA replication, which is in line with the chromatin accessibility results obtained in CLL patients[49], and likely due to ibrutinib-mediated suppression of proliferation and induction of apoptosis. These data, together with the observation that the mutational status of patients does not seem to be predictive of the drug response, corroborate the fundamental role of epigenetic changes as a basis of ibrutinib resistance and highlight the need of integrating multi-omics data instead of merely focusing on genetic aberrations as a means to understand ibrutinib resistance. Moreover, the interesting similarities between human and mouse proteome results in our study support the suitability of the TCL1-AT mouse model for studying disease biology and for pre-clinical drug testing, and further exemplifies the potential of integrating multiple orthogonal omics datasets. A limitation of our study is the lack of phosphoproteomic data which was mainly due to the low input material available, especially for clinical samples. The inclusion of such analyses would offer additional layers of functional and regulatory information that are crucial for understanding cellular signaling and disease mechanisms.

The overall survival benefit in TCL1-AT mice treated with carfil-zomib as therapy after ibrutinib monotherapy ultimately reinforces the hypothesis that PIs may represent a treatment option for CLL patients relapsing early on ibrutinib.

## Methods

### Study approval
The research of this study was carried out in accordance with all relevant ethical regulations and the Declaration of Helsinki. Peripheral blood (PB) samples were obtained from CLL patients after written informed consent and according to the declaration of Helsinki. Samples were provided by the B-cell malignancies Biobank at Amsterdam University Centres, Amsterdam, the Netherlands, by the Department of Hematology, Oslo University Hospital, Norway, and by the University Hospital Ulm, Germany. Ethical approval was provided by the Amsterdam University Centres ethical and biobank committee (METC 2013_159), by the Regional Committee for Medical and Health Research Ethics of South-East Norway (2016/ 947), and the Ethics committee of Ulm University, Ulm, Germany (96/08). Patients were selected based on their clinical and biological characteristics that made them suitable for the research question under study. Written consent was obtained by the patients for the use and publication of personal and medical data. No compensation was given to the patients.

All animal experiments were carried out according to governmental and institutional guidelines and approved by the Regierungs-präsidium Karlsruhe (permit numbers: G-39/19, G-77/19 and G-53/15).

### Human samples
Longitudinal samples from patients who matched the standard diagnostic criteria for CLL were taken before treatment start, during treatment with ibrutinib, and when patients showed resistance to therapy, either before or after discontinuation of ibrutinib. Peripheral blood mononuclear cells (PBMCs) were isolated on a Ficoll density gradient (1.077 g/mL) by collecting the interphase cells after centrifugation at $1000\,g$ for 20 min. Enrichment of B cells was performed with CD19 MicroBeads (Miltenyi Biotec, Bergisch Gladbach, Germany) according to manufacturers' instructions. LD columns were used to first deplete the unlabeled fraction (flow through) and then to isolate the magnetically labeled fraction (CD19+ cells). Before any subsequent procedure, the percentage of enriched cells (CD19+ CD5+) was measured by flow cytometry and only the samples with a double positive population >95% were included in the study.

Clinical information of patients is provided in Table 1.

### Mice and tumor models
C57BL/6 wild-type (WT) 6–8 months old mice were purchased from Charles River Laboratories (Sulzfeld, Germany). $E\mu$-TCL1 ((B6-Tg(Igh-V186.2-TCL1A)3Cro) – TCL1) mice on C57BL/6 background[55] and adoptive transfer (AT) of TCL1 tumors were previously described[56,57]. Briefly, malignant B cells were enriched from splenocytes of TCL1 mice using EasySep Mouse Pan-B Cell Isolation Kit (StemCell Technologies, Inc., Cologne, Germany), or by depletion of T cells using mouse CD90.2 microbeads (Miltenyi Biotec, Bergisch Gladbach, Germany), according to the respective manufacturer's protocol. The CD5+ CD19+ content of purified cells was typically above 95%, as measured by flow cytometry. $1–4 \times 10^7$ malignant

**Table 1 | Clinical features of CLL patients**

| Patient | Treatment stage | Mutation | IGHV-status | Other comments |
|---|---|---|---|---|
| 1<br>Male<br>*1967 | Before ibrutinib | SF3B1: K700E (VAF 51%) | unmutated V3-9 (IGHV DNA sequence homology in %:100) | No TP53 mutations, normal karyotype (FISH) |
|  | During ibrutinib | SF3B1: K700E (VAF 51%) |  |  |
|  | Relapse | SF3B1: K700E (VAF 50%); BTK: C481S (VAF 92%) |  |  |
| 2<br>Male<br>*1954 | Before ibrutinib | XPO1: E571 (VAF 20%);<br>TP53: G245V (VAF 73%);<br>TP53: R213X (VAF 21%) | unmutated V3-30 (IGHV DNA sequence homology in %:100) | karyotype: del(13q), del(17p), del(14q), TP53 exon 6 c.637 C > T p.R213X, exon 7 c.734 G > T p.G245V |
|  | During ibrutinib | XPO1: E571 (VAF 5%);<br>TP53: G245V (VAF 19%);<br>TP53: R213X (VAF 81%);<br>BTK: C481S (VAF 3%) |  |  |
|  | Relapse | XPO1: E571 (VAF 24%);<br>TP53: G245V (VAF 77%);<br>TP53: R213X (VAF 18%) |  |  |
| 3<br>Female<br>*1953 | Before ibrutinib | BIRC3: Q547Nfs* (VAF 19%) BIRC3: K558Q (VAF 4%)<br>BIRC3: I427R (VAF 3%) | unmutated V1-69 (IGHV DNA sequence homology in %:100) | karyotype: del(13q), del(11q), del(17p), del(14q), TP53 exon 8 c.797 G > A p.(Gly266Glu) |
|  | During ibrutinib | BIRC3: Q547Nfs* (35%) BIRC3: K558Q (VAF 8%) TP53: G266E (VCF 7%) |  |  |
|  | Relapse | EGR2: E412G (VAF 7%);<br>BIRC3: K558Q (VAF 67%);<br>PLCG2: D993G (VAF 7%);<br>TP53: G266E (VAF 7%);<br>RPS15: S139F (VAF 4%);<br>BTK: C481S (VAF 41%) |  |  |
| 4<br>Male<br>*1957 | Before ibrutinib | DNMT3A: R882H (VAF 4,2%); TP53: wt | mutated | karyotype:<br>del(13q14) |
|  | Relapse | BTK: wt, PLCG2: wt |  |  |
| 5<br>Male<br>*1955 | Before ibrutinib | EGR2: E356K (VAF 18%);<br>ATM: c.3994-1 G > T (VAF 82%) | unmutated V1-69 (IGHV DNA sequence homology in %:100) | karyotype: (partial) trisomy 8q24.21,<br>del 11q22 (83%),<br>del 14q32,<br>IGH break, translocation with unknown partner |
|  | Relapse | EGR2: E356K (VAF 23%);<br>ATM: c.3994-1 G > T (VAF 96%)<br>BTK: C481S (VAF 6%);<br>BTK: C481Y (VAF 92%) |  |  |
| 6<br>Female<br>*1958 | Before ibrutinib | ATM: S2375Kfs*2 (VCF 35%) RPS15: S145F (VAF 50%) TP53: c.376-2 A (31%) | unmutated V1-02 (IGHV DNA sequence homology in %:99.5) | karyotype: 6q21 deletion (78,5%)<br>del 13q14.3 (8,5%) |
|  | Relapse | ATM: S2375Kfs*2 (VAF 5%);<br>RPS15: S145F (VAF 44%);<br>TP53: c.376-2 A;<br>BTK: C481R (VAF 28%);<br>BTK: C481S (VAF 1%) |  |  |
| 7<br>Male<br>*1941 | Before ibrutinib | SF3B1: K700E (48) ATM: C2770W (99%) | mutated V3-48 (IGHV DNA sequence homology in %:97.9) | karyotype:<br>del 14q32 (87,5%) |
|  | Relapse | SF3B1: K700E (VAF 37%);<br>ATM: C2770W (VAF 74%);<br>BTK: wt; PLCG2: wt |  |  |
| 8<br>Male<br>*1953 | Relapse | TP53: G105Afs*18 (VAF 43%); R249T (75%);<br>EZH2: Y646F (VAF 15%);<br>BTK: wt; PLCG2: wt | unmutated | karyotype: del(17p13) |
| 9<br>Male<br>*1963 | Relapse | BTK: C481S (VAF 86%); PLCG2: wt;<br>TP53: wt | unmutated | karyotype: del(13q14) |
| 10<br>Male<br>*1975 | Relapse | BTK: C481S (VAF 2.7%); C481S (VAF 3.9%); T474F (VAF 2.4%);<br>PLCG2: wt; TP53: wt | unmutated | karyotype: del(11q22) |

*wt* wild-type.

TCL1 splenocytes were transplanted by intravenous (i.v.) or intraperitoneal (i.p.) injection into 6 week-old C57BL/6 WT mice. Tumor progression was monitored in PB. Spleens were dissected when animals were moribund and tumor load was analyzed. For sequential adoptive transfer experiments, viably-frozen mouse splenocytes coming from a single donor mouse were thawed, counted to check for cell viability, and injected in recipient mice as described above. All animal experiments were carried out according to governmental and institutional guidelines and approved by the local authorities (Regierungspräsidium Karlsruhe, permit numbers: G-39-19, G-77/19 and G-53/15).

Female mice were used for all experiments as disease development was more homogeneous compared to male mice. All animals were housed at the central animal facility of the German Cancer Research Center (DKFZ) in groups of up to six mice in Tecniplast GM500 IVC cages with a 12 h light/dark cycle. Mice had *ad libitum* access to water and food. The colony was regularly controlled for infections using sentinel mice to ensure a healthy status. The maximal tumor burden permitted by the ethics committee of 120,000 malignant B cells per $\mu l$ blood was not exceeded. All mice were sacrificed by means of $CO_2$ asphyxia in increasing concentrations.

**Sample preparation for mass spectrometry analysis and MS data acquisition.** Either mouse or human CLL cell pellets were resuspended in 150 µL of 0.1% RapiGest Surfactant (Waters Corporation) in 100 mM ammonium bicarbonate (AmBic) containing 10 mM chloroacetamide (Sigma) and 40 mM tris(2-carboxyethyl)phosphine, and sonicated for 15 cycles with a PicoBioruptor (Diagenode). Samples were quantified, heated at 90 °C for 5 min and subjected to tryptic digestion. Acidified peptides were subjected to buffer exchange via SP3 clean up protocol[58,59], labeled with TMT10plex (Thermo Scientific) according to manufacturer's instructions, pulled fractionated by high pH, and concatenated into 12 or 24 fractions. Peptides were loaded on a trap column (PepMap100 C18 Nano-Trap 100 µm × 2 cm) and separated over a 25 cm analytical column (Waters nanoEase BEH, 75 µm × 250 mm, C18, 1.7 µm, 130 Å,) using the Thermo Easy nLC 1200 (Thermo Fisher Scientific), and analyzed on a Tri-Hybrid Orbitrap Fusion or Lumos mass spectrometer (Thermo Fisher Scientific) in data-dependent mode. Both MS1 and MS2 scans were acquired in the Orbitrap.

**Mass spectrometry data processing analysis and visualization.** RAW data were processed with Maxquant software (2.0.3.0)[60,61], or Proteome Discoverer v2.1 (Thermo Scientific), with default parameters: In MaxQuant, match between runs option was enabled. Perseus software[62] was used for further protein analysis on proteins identified with at least one unique peptide in each biological replicate. Limma-based batch effect r package was used for batch effect correction. Two-sided $t$ test statistics was used for volcano plots generation with FDR was 0.05 and S0 constant was 0.1. Pathway enrichment analysis was performed using the Metascape resource[63], while WebGestalt with default parameters was used for GSEA[64].

A detailed description of all methods is available as supplementary information.

### Reporting summary
Further information on research design is available in the Nature Portfolio Reporting Summary linked to this article.

## Data availability
The WES raw data generated in this study have been deposited at BioProject under accession code PRJNA889569. The raw RNA sequencing data generated in this study have been deposited in GEO under accession code GSE215414. The mass spectrometry raw and processed data generated in this study have been deposited to the ProteomeXchange Consortium via the PRIDE[65] partner repository with the dataset identifier PXD037314 and PXD053512 [http://www.ebi.ac.uk/pride/archive/projects/PXD037314, http://www.ebi.ac.uk/pride/archive/projects/PXD053512]. The remaining data are available within the Article, Supplementary Information or Source Data file. Source data are provided with this paper.

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

## Acknowledgements

We acknowledge the Central Animal Laboratory, the Flow Cytometry and Genomics and Proteomics Core Facilities at the DKFZ. The study has been supported by a fellowship of the DKFZ Clinician Scientist Program, supported by the Dieter Morszeck Foundation to PMR, by the DFG SFB1074 subproject B1 to ET and StSt, by the Research Council of Norway under the frame of ERA PerMed (CLL-CLUE: project number 322898) to SSS, by the Stiftelsen Kristian Gerhard Jebsen (Grant 19) to SSS, by the Norwegian Centre for Clinical Research (MATRIX: Research Council of Norway and the Norwegian Cancer Society joint grant 328827) to SSS, by the German Cancer Aid (grant 70114114) to MS, and the German José Carreras Foundation (grant 03 R/2021) to MS.

## Author contributions

L.A. conceptualized the study, designed and performed experiments, analyzed and interpreted data, generated figures and wrote the manuscript. G.S. designed and performed proteomic experiments, analyzed and interpreted data, generated figures and wrote the manuscript. H.Y.

designed and performed experiments, analyzed data and generated figures. J.U.H. performed experiments under the supervision of S.S.S., and they generated figures. D.F., M.R., and M.M.S. contributed to acquire mass spectrometry data. N.M. and V.K. generated in vivo data. S.O. and I.R. generated in vitro data. M.I., M.K., D.F., Y.P., and M.Z. performed bioinformatic analyses. J.D., A.P.K., E.T., and StSt provided clinical samples. E.E. and A.P.K. provided murine samples of the venetoclax treatment study. S.D. and P.M.R. provided intellectual feedback to the study. J.K. and P.L. provided logistic and budget support and intellectual feedback to the paper. M.S. conceptualized and supervised the study, interpreted data, provided logistic and budget support and wrote the manuscript. All the authors read, reviewed, and revised the manuscript.

## Funding

## Competing interests

S.S.S. has received honoraria from AbbVie, AstraZeneca, BeiGene, and Janssen, and research support from BeiGene and TG Therapeutics outside of this work. E.T. has received honoraria from Abbvie, Janssen, AstraZeneca, BeiGene and research support by Abbvie, Hofmann-La Roche and Janssen-Cilag. StSt has received advisory board honoraria, research support, travel support, speaker fees from AbbVie, AstraZeneca, BeiGene, BMS, Gilead, GSK, Hoffmann-La Roche, Janssen, Novartis, and Sunesis. JD has received research funding from Roche/Genentech. MS has received advisory board honoraria from AbbVie and research support from Bayer AG. The remaining authors declare no competing interests.

## Additional information

[1]Division of Molecular Genetics, German Cancer Research Center (DKFZ), Heidelberg, Germany. [2]Division of Proteomics of Stem Cells and Cancer, German Cancer Research Center, Heidelberg, Germany. [3]Heidelberg University, Medical Faculty, Heidelberg, Germany. [4]Department of Cancer Immunology, Institute for Cancer Research, Oslo University Hospital, Oslo, Norway. [5]K. G. Jebsen Centre for B Cell Malignancies, Institute of Clinical Medicine, University of Oslo, Oslo, Norway. [6]Cellzome, a GSK Company, Heidelberg, Germany. [7]Division of Neuropathology, German Cancer Research Center (DKFZ), Heidelberg, Germany. [8]EMBL, Proteomics Core Facility, Heidelberg, Germany. [9]Department of Haematology, Jeffrey Cheah Biomedical Centre, University of Cambridge, Cambridge, United Kingdom. [10]Department of Experimental Immunology, Cancer Center Amsterdam, Amsterdam Institute for Infection and Immunity, Amsterdam University Medical Centers, University of Amsterdam, Amsterdam, The Netherlands. [11]Department of Hematology, Cancer Center Amsterdam, Amsterdam Institute for Infection and Immunity, Lymphoma and Myeloma Center Amsterdam, Amsterdam University Medical Centers, University of Amsterdam, Amsterdam, The Netherlands. [12]Division of CLL, Department of Internal Medicine III, Ulm University, Ulm, Germany. [13]Heidelberg University, Department of Hematology, Heidelberg, Germany. [14]Düsseldorf University, Department of Hematology, Düsseldorf, Germany. [15]European Molecular Biology Laboratory (EMBL), Genome Biology Unit, Heidelberg, Germany. [16]These authors contributed equally: Lavinia Arseni, Gianluca Sigismondo. ✉e-mail: m.seiffert@dkfz.de

