## [Transparent Peer Review file · Nature Communications]

Longitudinal omics data and preclinical treatment suggest the proteasome inhibitor carfilzomib as therapy for ibrutinib-resistant CLL

Corresponding Author: Dr Martina Seiffert

Version 0:

Reviewer comments:

Reviewer #1

(Remarks to the Author)

The manuscript written by Arseni and colleagues describes experiments in a mouse model of ibrutinib resistance to investigate if proteasome inhibition could be a potential salvage therapy. Later on, also patient samples of ibrutinib resistance CLL are included to confirm and translate their findings. By applying transcriptomics and proteomics on different timepoints in the mouse model, the authors identify the proteasome activity as driver of ibrutinib resistance. Treatment with CFZ is then tested on different timepoints and shows efficacy after ibrutinib resistance as a salvage regimen. Finally, proteome profiling is performed in CLL patient samples on different timepoints before and upon ibrutinib resistance which confirms the findings of their preclinical model. Finally, proteasome inhibition shows to effectively induce CLL apoptosis in vitro.

The idea of combining proteasome inhibition with ibrutinib has been around for some time, the novelty is that the authors wish to uncover mechanisms of ibrutinib resistance and arrive at proteasome regulation. The authors made a very extensive effort to characterize the ibrutinib resistance cells, although truly integration of the multi-omics is lacking and the datasets are more standalone per experiment. The translation of their preclinical data to patient samples is interesting and relevant for the field. I think there are a few major issues to be resolved before publication would be appropriate. Please see my suggestions:

1 - In line 147-150 the authors state that the I-early group clusters away from the others which is evident in the PCA plot in S2B. However the heatmap in fig 2B clearly shows that I-late is distinct from the others. The transcriptomic data then does not match the proteomic data in 3A showing again that I-early clusters away from the other groups. Is this correct and is the data reliable?

2 - In line with the previous point, it would be helpful if the authors share in more detail the overlap between the transcriptome and proteome besides a broad pathway that is dysregulated. Are similar genes/proteins changing in the same direction? From the pathway plot in figure 2D it's not clear if direction is taken into account, are the majority of genes in the proteasome pathway upregulated or downregulated in the ibrutinib resistant mice? The authors state in line 175-182 that proteasome regulation pathways and ubiquitination are downregulated in ibrutinib resistance, it seems counterintuitive that then proteasome inhibition could be effective. It would be helpful if the authors discuss this more clearly.

3 - In figure 5 the authors analyze the proteome after CFZ. One would expect quite strong differences if the proteasome is efficiently inhibited because any type of protein marked for degradation stays intact. Could it be that the proteasome is already downregulated in resistant cells and therefore further inhibition does not have much effect? Or is the hypothesis that due to ER stress and apoptosis there is more cell death but the protein composition is not strongly affected? Elaboration on the (hypothetical) mechanism here would increase the attractiveness of this manuscript.

4 - Even though the survival benefit of adding CFZ after ibrutinib resistance is significant, it is still a minor benefit. Also when looking at the CLL cells in the blood, there is quite some overlap between data points where some mice with ibrutinib

monotherapy have less CLL burden than mice treated with the combination. Be cautious with the conclusions here.

Minor comments:

5 – Statement made in lines 76-79, suggests that CFZ has been tested in CLL and identified as a “potent drug that induces cytotoxicity in CLL [..]”. None of the references mention anything about CFZ in CLL, only later in line 87-88 this is properly described

6 – Methods on MS analysis are overlapping from within the main text and the supplementary methods. Better would be to provide short and summarized methods in the main text and refer to the supplement for more detailed methods. A reference to the supplemental methods is also not made.

7 – In the sequential adoptive transfer experiments it is unclear how many donor mice are used. Is one ibrutinib resistant mouse supplying the cells for all the subsequent adoptive transfer or are there multiple donors? It would be good to make sure the methods more closely align with the ARRIVE guidelines for reporting on animal experiments

Reviewer #2

(Remarks to the Author)

Arseni et al. manuscript has a high significance to the field of CLL at both preclinical and clinical level addressing the issue of resistance to current target therapy in particular BTK inhibitors. The work is clearly written and previous works acknowledged with relevant references. The work also supports the conclusions in particular regarding the in vivo part, while for the part regarding the patients additional evidence or discussion is needed (see following comments). The methodology sounds and meets the expected standards but more details are needed for the work to be reproduced.

Major comments:

- In Figure 4 is evident a general higher variability in untreated ibrutinib resistant mice, this was the case also for the mice in figure 1? In this case the graphical representation does not allow to see it. If this was the case please comment. It would be interesting to see an immunohistochemistry analysis of the clone and Ki67 to reinforce the model.

- How does author count the absolute number of CD19+CD5+ cells in the tissues in figure 4? It is not described in the methods, as the data are not in line with the % of the clone it is important to understand how the analysis has been done.

The last part is very interesting and important to reinforce the in vivo part and confirming the predictivity of the model. However, some details are missing:

- In the figure 7A a magnetic separation of CD19+ cells is represented while in the methods a cell sorting for the CD19+CD5+ is mentioned. Which is the correct method? If authors used the isolation of CD19+ cells it is important to know the % of the clone in the population analysed in order to consider the contribution of healthy b cells to the analysis.

- In figure 7I authors mention that they use an ex-vivo lymph node microenvironment culture system. They acknowledge the reference but no details at all reported in the methods please add. In this case again the cells are CD19+ or CD19+CD5+, see question before.

Minor comments:

- In figure 1,2,4 the TCL1-AT model has been established by i.p o i.v. injection it is not clear.

- Figure 3 panels B and C are not readable the dots with different colours are not visible author should identify a clearer way to show these results.

- In Figure 4 I would suggest to use a different colour for treated and untreated mice not just a different tone of blue it is difficult to follow over all panel B.

Reviewer #3

(Remarks to the Author)

This manuscript used a multi omic approach to identify features associated with resistance to BTK inhibitors in CLL, and identified proteasome inhibition (PI) as a potential therapeutic modality. BTKi resistance historically was a problem in a notable fraction of CLL cases, but the magnitude of the problem is decreasing as BTKi is more commonly being employed in newly diagnosed, or first line settings, so primary resistance arising from prior resistance and mutational evolution is less. It is also becoming less relevant as newer triple combinations with BCL2i +BTKi + anti CD20 antibody combinations, which lead to deeper MRD negative remissions (whether they are cures, or longer lasting needs more time) are being increasingly employed. But resistance remains a problem for some, and identifying means of counteracting it retains importance, albeit for a smaller number of cases then when the researchers likely began this project.

My first major concern about the data relates to the model, and its ability to replicate what is seen in patients, and thus be a

useful guide. Most BTKi resistance (BTKi-Res) to date has involved mutations in BTK and PLCG2, as noted in line 70, and this more commonly evolves over a long time period (years). Truly “chronic” CLL has been nearly impossible to develop models for. This research team uses the TCL1-AT model, which leads to moribund or dead mice in a matter of weeks, not months to years as is typical of CLL. In the presented studies BTKi Resistance is seen with outgrowth after only ~ 6 weeks (Fig 1B), not the years typically seen in newly diagnosed CLL cases. So the “BTKi-Res” seen here does not arise slowly over time, after an initial response, and therefore its relevance to clinically seen BTKi-Res is suspect. Furthermore, after generating their BTKi-Res CLL molecular testing did not show the common mutation (or any dominant mutation as discussed below). We therefore have a model that doesn’t replicate the patterns seen in patients with respect to frame, or initial response followed by emergence of resistance, and the resistance that is generated doesn’t mutationally mirror that seen in clinical practice. Hence, my first major concern is the relevance of the findings of this model to actual clinical findings and the generalizability of these findings, from a very aggressive disease model, without commonly found mutations, to that of the actual clinical setting. The same pattern and timing are seen with two other primary tumors (supp fig 1), again suggesting that true BTKi-Res is not being selected for given the time frame.

With that concern defined, the next question is whether there are compelling discoveries worth pursuing?

A second concern is the specificity of the changes observed after BTKi-TX. I think that some crucial control experiments were omitted. Fig 1 and associated text looked at vehicle vs BTKi, and saw delayed growth (multiple endpoints) but outgrowth that they feel represents BTKi-Res by 6 weeks. They only compared vehicle vs BTKi, so we cannot know if the differences that follow are limited to BTKi therapy, or are the result of therapy in general. Would treatment with venetoclax, or bendamustine, or Fludarabine/Cytosin, have led to similar results? The data shows that I-Long cells were resistant to rechallenge with BTKi in serially transplanted mice, suggesting that cells that were not sensitive to BTK were selected for. But are the changes seen BTK specific? Or would similar changes have been seen with a different therapy? Some comparator would allow better definition of what subsequent findings were BTK specific, vs. treatment specific.

There is a curious finding (line 117-18) that I-late cells proliferate in vitro, but V-late cells don’t proliferate in vitro, yet these same V-L cells were leading to moribund mice that required euthanasia. So, they were clearly proliferating in the mice, but not in vitro. So does this represent a problem with the experimental system, or can we truly believe that there is a fundamental difference in I-Late vs V-late cells? Part of this may come from comparing a different treatment condition to a different control condition. Notably V-L are collected at three weeks, and I-L collected at six weeks, so they are from different in vivo durations. As discussed below, the other data shows changes from time in vivo. So, how much of the V vs I-late proliferation difference is from time in culture and how much from the BTKi-Res effect? Did more time allow for selection of a more robust cell, independent of BTK effect, hence more proliferation?

I do not understand the data in line 138. When they sequenced the BTKi-Res cells they noted some cells with variants in PIK3CD and VMN2R27, but the VAF was low (<12.5%). There is a paradox here. If they selected for a unique resistant cell then it should be clonal, and dominant. But the fact that these new changes were seen in a paucity of cells argues that the majority of BTKi-TX I-long cells at 6 weeks are outgrowth of the original population, rather than selection for a resistant population. This casts further doubt that they have selected for a truly BTKi-Res cell.

Figure 2 B (Heatmap) shows that there are clear changes between the vehicle treated cells at 1 vs 3 weeks. I see at least three blocks of genes that are red at 3 weeks and green (or much less red) at 1 week. This demonstrates that time in the mouse is leading to changes in these cells. As noted above, the BTKi-TX cells were taken at 6 weeks, and the V-late at 3 weeks. So if there are “time in mouse” dependent changes, how can we have confidence that the changes in I-late, taken 3 weeks later, don’t merely reflect additional “time in mouse” related changes? They would need a vehicle control at 6 weeks, or a BTKi-TX match at 3 weeks, for a better comparison. Since they had to euthanize the vehicle mice at 3 weeks the 6 week comparator isn’t possible with vehicle alone, but might be with an alternative treatment. So, once again, I cannot answer concerns about whether the findings are time in mouse or BTK-TX related. Minor comment, Fig 2b, a version with the gene names readable should be presented somewhere.

Next they did proteomics, using cells from three patients, with three BTK TX related timepoints for each, pre, during and at resistance. Uncommented on is whether they measured total protein levels, or also measured phosphorylation and other post translational modifications? These cases have different mutation profiles, with two having BTK-mut at relapse, and case two selecting for a TP53 mutated clone, but no BTK-mut and case three having both. So we might expect different protein profiles between them, given 3 different putative reasons for resistance. However, Sup Fig 3 shows that the protein expression correlation between any of these has almost no variation from each other. This result seems illogical to me. The plots in Fig4D are also confusing. In cluster 11 (up reg) and Cluster 1 (down reg), the changes are seen after only 1 week, but are somewhat more pronounced after 6 weeks. If resistance required more time to develop then this suggest that these are rapidly induced BTKi induced changes, rather than changes associated with a mutation that leads to resistance being selected for. The time frame is impossible for that. What would happen to these changes if passaged into mice and treated with V or BTKi after that? Would they see reversion in those where BTKi selective pressure is removed? Or change to yet another pattern? And would those treated with BTKi for a second course maintain this pattern? Or change further? In clusters 7 and 8, there are up regulated groups in the V treated, but in the BTKi-TX these are more (cluster 8) or less (cluster 7) at 1 week, but this effect is lost with BTKi-Res at 6 weeks. This is further evidence that time in mouse is leading to changes in the cells, and the reversion of the BTKi-TX late to look like V-E (7) or V-L(8) suggest that these changes are time, not treatment related. In fig 3E, if Ubq proteins were measured with the MS-proteomics, then a more detailed reporting of what proteins had changes in Ubiquitination could be presented. It is a bit confusing regarding the data in lines 160-170, is this for 1 of the three cases? The three cases pooled? The proteins that are differentially expressed in the noted clusters should be identified somewhere. A heatmap showing the protein expression by cluster for the 3 cases should be presented.

From this they got a lead to look at PI in CLL leading to the next set of experiments in which they tested PI vs the TCL1 CLL, showing that it slowed growth in I-naïve and I-TX cells (Fig 4B) in multiple sites. They next did proteomics on BTKi naïve and treated, +/- carfilzomib (Fig 5A [minor comment the red and orange are hard to distinguish, especially when the figure is miniaturized for printing.]) to ID proteins uniquely changed by CFL in BTKi-Res cells. In the figure all the prior TX are normalized to close to 0, but it would be superior to show actual fold changes, A protein change from -2 to -3 likely is irrelevant, one that goes from -1 to +1 is likely relevant, but this cannot be appreciated as normalized.

Next they tested PI alone or in combo (PI + BTKi) in TX naïve mice, but surprisingly the benefit of CZL was lost. It was only seen in BTKi-Res when used after BTK was stopped. Fig 6D and text (lines 235-240) claims that this was effective in slowing CLL development and led to longer survival, and while the p-values are “significant” the visual impression is that there was minimal effect and that all the mice still died with a 1 to 1.5 week delay. This does not “clearly show that CZF is an effective salvage therapy” line 239. A similar delay may have been seen with any chemotherapy and doesn’t clearly show that there is a PI specific effect. So again, a crucial control was omitted. So I find this data confusing and unconvincing.

Finally they tested different PI drugs vs. samples from the three patients, with proteomics showing a set of proteins both up/down regulated during therapy, that mostly revert to the pretreatment level at relapse/resistance. It is unclear if the relapse/resistance samples were taken while the patients were still on the BTKi, or after they had been off it for a while. If they had been off BTKi for a while then these changes represent the effects of BTKi therapy, rather than changes associated with resistance. Fig 7G does show a block of proteins that are different at relapse than during therapy, or pre treatment, so some of these changes may be BTKi-Res related. They do show that all three different patients were sensitive to three different PI (FIG 7I).

Of relevance to this manuscript: Griffen, T.L., Hoff, F.W., Qiu, Y. et al. Proteomic profiling based classification of CLL provides prognostication for modern therapy and identifies novel therapeutic targets. *Blood Cancer J.* 12, 43 (2022). <https://doi.org/10.1038/s41408-022-00623-7> Proteomic profiling based classification of CLL provides prognostication for modern therapy and identifies novel therapeutic targets | *Blood Cancer Journal* (nature.com) Figure 1 in that study shows that ubiquitin proteins and especially UPR, and epigenetics proteins were prognostic in CLL, but in figure 4 UPR, and one of the epigenetic groups but, not UBq protein were also prognostic for BTKi treated cases. Epigenetic proteins were prognostic in all CLL cases and in the BTKi treated. So if the unfolded protein response is truly important in CLL, then that would go along with generalized PI sensitivity in CLL. UPR upregulation may be an adaptive response to address the stress of accumulating proteins, relying on proteasomal degradation as part of this, hence PI use blocks the effect of this adaptive response, sensitizing the cells. But if BTKi is upregulating UPR, it is unclear why adding PI would not be effective as found in this study. I’d like to see more thought on that puzzling result in the discussion. They should also look at another large proteomic dataset presented in Meier-Abt F, Lu J, Cannizzaro E, Pohly MF, Kummer S, Pfammatter S, et al. The protein landscape of chronic lymphocytic leukemia (CLL). *Blood.* 2021;138:2514–25 for other corroborating data.

In summary, we have somewhat intriguing data and a possible therapeutic lead. But it arises from a model that has uncertain relevance to clinical CLL, with findings that may be related more to time, than therapy, with “resistance” that is not clearly related to the BTK therapy. There are too many uncertainties to have confidence in the data as presented.

Reviewer #4

(Remarks to the Author)

Comments to Authors

In this project, the group develops a mouse model of ibrutinib resistance by treating mice upon adoptive transfer of Eu-TCL1 leukemia continuously with ibrutinib. Using cells from these mice with ibrutinib resistant CLL, authors perform transcriptome and proteome analyses during therapy (sensitive to ibrutinib) and after disease progression (resistant to ibrutinib). These data were compared to those obtained from 3-4 patients with CLL with ibrutinib-resistant disease. Based on omics data that suggested over-activation of proteasome activity, authors use proteasome inhibitors (PI) such as carfilzomib in mouse model to suggest utility of this drug in treating ibrutinib-resistant CLL. This was then extended to CLL cells obtained from patients. Authors need to be congratulated for creating ibrutinib-resistant mouse model. There are, however, challenges to extrapolate from this model to the disease in human that becomes resistant to ibrutinib.

Major comments

1. In humans, most common ibrutinib resistance mechanism is identified as alterations in BCR pathway proteins that include mutations in the binding site of ibrutinib (i.e. C481 substitutions), gate-keeper mutations (T474 substitutions) and downstream PLCG2 alterations. There are some patients who develop disease resistant to ibrutinib without these mutations. In mice, in the current project, when the disease progresses and is coined as ibrutinib-resistant CLL, authors do not find any mutations in the BTK. However, authors compare mice data to 4 patient samples (Figure 7) with respect to omics and biological response to proteasome inhibitors including carfilzomib. This extrapolation needs to be revisited by comparing mice data with human data where CLL is progressed without any BTK or PLCG2 mutations. Considering these fundamental differences in mouse versus human ibrutinib-resistant cells, it is challenging to accept mouse model as a replica of human ibrutinib-resistant disease.

2. Throughout the manuscript, authors propose and suggest that ibrutinib-resistant CLL in mice respond well to carfilzomib indicating sensitization of cells to carfilzomib after disease progression. However, data suggest that carfilzomib treatment responded similarly in ibrutinib-naïve and ibrutinib-resistant disease. For this comparison, authors should statistically compare carfilzomib treated ibrutinib-naïve and ibrutinib-resistant cohorts in Figure 4D-F and Supplemental Figure 4 A-C. Similarly, in Figure 5, authors should include a volcano plot to compare carfilzomib-treated ibrutinib-naïve with carfilzomib-

treated ibrutinib-resistant CLL (Res CFZ – Naïve CFZ). Currently, this is a supplemental figure. In general, it appears that both ibrutinib-sensitive and resistant CLL respond to carfilzomib.

3. Carfilzomib was selected based on proteasome pathway overexpression data in Figure 2C. Several other pathways, such as replication, DNA repair, cell cycle etc were also upregulated. Did authors test other drugs that target these cellular processes?

4. Similarly, other targeted agents such as venetoclax should be tested for treatment of ibrutinib resistant mice CLL cells.

5. For data in figure 1C and D, authors should compare week 1 vehicle treated with week 3 vehicle treated cells. Why do vehicle-treated cells show less Ki-67 stain at week 3? Did authors observe lymphocytosis after treatment of diseased mice with ibrutinib?

6. Number of CLL patient samples is very low (Figure 7). Authors should add additional patient samples including those that are resistant to ibrutinib without any BTK or pLCG2 mutations.

7. Authors should elaborate data regarding carfilzomib monotherapy results in patients with CLL (reference 29). It appears that there was very limited activity of carfilzomib in CLL.

Minor comments

1. Based on text for Figure 4B, ibrutinib-naïve should be on the right panel.

2. On line 87, previous should be previously.

3. On line 288, it would be important to cite Honigberg paper (ref 12).

Version 1:

Reviewer comments:

Reviewer #1

(Remarks to the Author)

The manuscript has been significantly improved and the authors have sufficiently addressed my comments.

Reviewer #2

(Remarks to the Author)

Authors addressed clearly to all the questions I raised.

Reviewer #3

(Remarks to the Author)

TO THE AUTHORS:

I have read the response to the initial review and can see that the authors extended significant effort to try and address the critiques. In some ways it is improved, but in many ways I still have the same concerns that I had in the first review.

Comments on "Response to Reviewers"

I've read their points about the validity of their model, but still have major concerns about its applicability to clinical ibrutinib resistance.

1. As noted before, this is resistance for single agent ibrutinib, but CLL therapy has moved on to doublets and triplets with venetoclax and obinituzumab and acalabrutinib, yielding unprecedented deep MRD negative remissions, that may prove to be cures. There is significantly less resistance with these newer therapies. Consequently single agent resistance is less relevant.

2. As noted before, clinical resistance to ibrutinib takes months to years to develop, and has characteristic mutations in most. Their model arises in weeks and lacks these mutations. So, while I think they performed their experiments well, and I concur with the conclusions they derived from the data, the relevance of this model to clinically seen resistance remains uncertain to me. As they note, there is a small set of early resistance patients for whom this may be representative.

I didn't feel that their responses to the issue of comparing V-L to I-L cells from 3 and 6 week timepoints really answered what was due to time vs what was due to ibrutinib.

Likewise, the point about time in mouse changes vs ibrutinib induces changes (original fig 2B) suggests that they are comparing identical disease states and that the time is irrelevant, but I didn't see data demonstrating that these were similar disease states as claimed. They also don't clearly define what these "different states" are and how they know that these are differ/similar.

The lack of phosphoproteomic data limits the findings. This limitation should be commented on in the discussion

Comments on new manuscript

The manuscript has two sections. The first part used transcriptomic and proteomics to assess what changes in a cell line defined as being ibrutinib resistant after short term ibrutinib exposure. From this they identified protein degradation as a possible therapeutic weakness for the I-resistant cells. In the second part they conduct experiments to show that proteasome inhibition therapy works on the I-resistant cells. In both the first and second reviews, most of my concerns relate to the first part.

PART 1: Multiomic analysis of ibrutinib resistant cell line.

On pages 5-6 data.the issue of whether the resistance is "acquired" or "selected" for is evaluated. It's too quick for new

mutations, which would require that it was selection for a small sub clone with existing mutations. But since they don't see mutation it must be acquired. They later posit that it is the transcriptional and protein profiles that show the adaptation, arguing that it is not mutation causing resistance, but rather is the protein signature.

THERE IS A FIGURE THAT I GENERATED FOR THIS RESPONSE THAT I CANNOT CUT AND PASTE HERE. I WILL ASK THE EDITORS HOW TO GET IT TO THE AUTHORS.

In Figure 2B. They claim (page 7 line 154-158) that this figure shows that I-E cluster separately from V-E V-L and I-L samples, but then say that I-L (Line 157-158, "loss of responsiveness to I" = I-L) look like Vehicle V-E and V-L treated cells. I disagree that VE, VL and IL look alike. I looked at the heatmap and by eyeball boxed areas in which the expression was much higher in I-E vs. V-E (Magenta Boxes ABCDE) then areas in VE- that were higher than I-E, but that looked like V-L (Light blue boxes H & F), and G which is common to V-E, V-L and I-E but not V-L, then those that were high only in V-L relative to I-E (gray boxes I,J,K,L, K). And finally for V-L identifying two boxes of high and low expressed genes (yellow box M,N). It is clear that none of these four look like each other.

I agree that I-E is unique as it shares only region G with V-E, only B and E with V-L and virtually nothing with V-L.

It's also clear that V-L looks nothing like V-E cells, which implies that either time in culture, or the vehicle is inducing many changes. This is also true from Supp fig2 PCA where VE and VL are in different spaces. VE is close to the reference cells in Sup2I, but V-L are far away, again suggesting a strong effect of the time in culture, or a vehicle induced effect. Furthermore it is overwhelmingly clear that I-L look nothing like the other three groups, and most importantly, nothing like the V-E cells.

This poses a major unexplained dilemma: If the protein pattern, achieved either by induction or selection, is crucial for resistance in I-E cells, and if the I-L cells are maintained under the same selection criteria, by continued exposure to Ibrutinib, until they were retested, then how can the I-L cells look nothing like (and in most cases, the exact opposite of) the I-E samples? This defies logic. If it is important for resistance, and exposure continues, they why doesn't the expression pattern continue?

Figure 2E shows a complete disconnect between protein expression and mRNA expression. There are as many completely inversely correlated proteins in I-L-Pro w.r.t. I-L-RNA as there are strongly positively correlated proteins. The other three comparisons also seem to show the same low correlation. (the correlation coefficient for these should be stated). In further support of this Fig S2D (PCA of protein transcription) shows that V-L and I-L are closest, while in S2I (protein PCA) V-L and I-L are far apart. Furthermore, S2I shows that I-L revert to be close to the reference and V-E profiles, suggesting that the I-E changes are lost. This argues against the selection for, or induction of a protein expression pattern that confers resistance. So there is neither mRNA-Protein correlation, nor PCA pattern correlation between the samples in mRNA vs. protein. There is ample evidence in the literature that mRNA is not equal to protein expression, so this reaffirms what others have seen. But, since it is the protein that acts, not the mRNA, this then makes the mRNA derived data for the rest of the manuscript suspect as we cannot know if high transcription means high protein, or the inverse. Furthermore there is the extremely puzzling observation on PCA, that the early changes leading to the I-E protein pattern needed for resistance is lost with additional time, despite continued "pressure" from Ibrutinib and reverts back towards baseline with additional time.

The authors then look to show that the proteome Page * line 183. Sup3A PCA now has IE and VE, which were very distant in S2I, close together, and far from the two late samples, which curiously sit close to the reference samples. If I exposure leads to the massive transcriptional changes seen in Fig 2B, and the very different protein profiles seen in Fig 2E, then why are the two early clustering together in S3A? This again makes me think that there is a lot of time in culture effect. Harder to explain, if exposure to I is leading to selection of a resistance profile, is why the VL and IL should be close to each other in S3A, and closer to the reference samples. This further argues against the idea that there is a unique I induced protein expression pattern being detected here.

The lack of consistency between the trends in the data from the different omics raises serious concerns.

Part 2: Proteasome inhibition in Ibrutinib resistant cells.

Page 9, line 219-235 & Figure 4. I agree with the interpretation. CZL alone had minimal effect on I naïve cells, but did inhibit growth (spleen, node size) in the I-resistant cells.

Page 10-11 lines 237-261. Figure 5. Proteomics in CZL treated -I-resistant cells.

Need to note line 240 that these are total proteins and that phospho, or other PTM are not assessed.

For Figure 5A. Per the legends at the lower left, this is total protein abundance. The authors confirmed in the "response to reviewers" that this was indeed normalized based on the expression of the resistant cells, but I still don't see that mentioned in either the methods or figure legend. I find the unnormalized figure in the RTR (page19) superior to the one used in the manuscript. For one the samples are labelled at the top (Naïve 1 2 3, Resistant 1 2 3 4, UT/CZL) so that we can compare how a sample did/did not change with CZL treatment. That cannot be done with the current Fig 5B. 5. Since the areas below the arms of a dendrogram can be freely spun to improve visual analysis, without changing the statistical implication of the clustering I would suggest flipping the order to move the left most ((NAÏVE_CZL_R3 to the far right next to NAÏVE_UT-R1. I would also reverse From NAÏVE-CZL-R2 to RES_UT_r1. This would put the Res together on the left and the Naïve on the R. Once reordered, it is clearer that there is not a lot of change with CZL tx in either set, but that the Res cells seem to maintain their starting point (compare UT vs CZL) and the UT tend to maintain their starting profile (UT vs CZL), supporting the contention from part 1 that resistance selection/induction induces a protein expression profile and that this is maintained over time Fig 5B. The purpose of this figure (line 244-245) is to ID a set of proteins that were specifically deregulated I the I-

res cells upon CZL treatment.

Page 11-12 Line 263-278 CZL as therapy for CLL. I agree with the laboratory work and interpretation until line 277. In fig 4 it is claimed that there was a significant decrease in growth (Fig4b ****) and in Spleen and node size (Fig 4 C E * and D, F **** and ***) in ibrutinib naïve cells, in some cases a statistical magnitude equal to that of the I-resistant cells. Yes in fig 6B & C adding CZL to V treated cells had no effect. So this is a discrepancy to be commented on. Why should CZL only work in I-res cells and not V in the fig6 experiments when it seemed to work in the fig 4 experiments? The lack of consistency needs some explanation.

Page 12-13, lines 281-337 Proteomics of patient derived I-res. Small numbers, but I don't think this can be helped. There is a question of what is treatment time related vs I-resistant related, similar to the issue with the mouse models (Line 310). This data might be stronger if you had a comparator from patients that were I-sensitive and maintained sensitivity (obviously you can probably only get the pre treatment and maybe 3 month timepoint samples, as they were unlikely to have many circulating cells by 6 months).

Discussion

As they note on lines 344-345 on page 14, the time course of their model and clinical resistance are very different, so their own comments raise questions about the relevance of their model to the clinic. I think they need to stress this more in the discussion, perhaps as a lead in to line 372 when they argue for alternative mechanisms.

Because there is conflicting data (Fig4 and 6) regarding whether PI works in I-naïve cells, its unclear if PI should be tried in all CLL cases, or only those with their variant form of I-resistance. Their data using I-res cells from patients with the typical BTK mutations suggest that it could be all I-res cases.

SUMMARY:

We have some interesting data that supports looking at PI in CLL, possibly in I-resistant cases, which supports the findings of others. But this is based on multiomic data that is confusing, and inconsistent in places, using a model with uncertain clinical relevance.

Minor comments.

Some color choices in the figures are still hard to discern.

Lines 145. It remains difficult to say that these mutations are crucial when they are present at a low VAF. Supplemental 1C seems to claim that 4 deleterious SNV were seen in all 3 cases. Why is that not mentioned here?

Supp fig 2D seems to be missing a I-lates dots. Others have 4, but only 3 for I-late. +

Reviewer #4

(Remarks to the Author)

Comments to authors

Authors have revised their manuscript by rebuttal comments from all four reviewers. Want to thank them for adding additional patients' data in the current work. Revised manuscript is improved and strengthened to support conclusions. However, there are a few comments that need to be addressed.

Major comments

1. Throughout the manuscript, including title, authors state proteasome inhibition and proteasome inhibitor for ibrutinib-resistant CLL. While omics data indicate proteasome pathway in ibrutinib-resistance, in preclinical treatment mostly carfilzomib (CFZ) was tested. Hence it is more appropriate to replace proteasome inhibitor by carfilzomib throughout the manuscript except for Figure 7K.
2. Authors have added seven additional patient samples (Yellow-highlighted in Table 1) to the original three patients. This enhances strength of the manuscript. Clinical features are provided for each patient however it is not uniformly described or presented, and important information is missing for some individuals. For example, VAF value should be added for first four patients. BTK and/or PLCG2 mutation information should be provided for patients 4, 7 and 8. For some genes, chromosomal location is included for some patients, but it is not incorporated for other patients.
3. Is there statistical difference between ibrutinib naïve versus ibrutinib-resistant CLL cells, spleen, and liver that are treated with carfilzomib in Figure 4 C, D, E, F? That is to compare data represented by blue solid squares and brown solid squares in panels C-F for Figure 4. If it is not significantly different, then these data would suggest that ibrutinib and carfilzomib sequential combination is effective in ibrutinib-sensitive as well as ibrutinib-resistant CLL.
4. For SF6A, is ibrutinib (I) treatment alone or carfilzomib (CFZ) treatment alone different than I+CFZ? A p value and statistical analysis should be included.
5. References 15 and 26 are same (Lamothe et al. Blood 2015). Based on the text of the manuscript it appears that authors are referring to single agent carfilzomib in CLL (one reference) and carfilzomib addition after ibrutinib therapy in CLL (another reference). Authors should appropriately cite the following two references
Single agent CFZ in CLL.
Lamothe B, Wierda WG, Keating MJ, et al. Carfilzomib triggers cell death in chronic lymphocytic leukemia by inducing proapoptotic and endoplasmic reticulum stress responses. Clin Cancer Res 22(18):4712-26, 2016.
Ibrutinib and CFZ combination in CLL.
Lamothe B, Cervantes-Gomez F, Sivina M, et al. Proteasome inhibitor carfilzomib complements ibrutinib's action in chronic

lymphocytic leukemia. Blood 125(2):407-10, 2015.

Minor comments

1. References 7 and 41 are same. Reference 41 needs to be deleted.
2. References 16 and 30 are same. Reference 30 needs to be deleted.
3. References 29 and 51 are same. Reference 51 needs to be deleted.
4. Reference 45, 46, 49, 64 and many others are incomplete. Authors should check all references for completeness and duplication.
5. SF4 title should be changed from 'Proteasome inhibitor treatment on '...to 'CFZ treatment on'.
6. For SF5B, Naïve CZF should be Naïve CFZ.

Version 2:

Reviewer comments:

Reviewer #3

(Remarks to the Author)

O THE AUTHORS

The comments are not numbered in my prior critique or in the authors response to reviewers, so I'll count the response # to ID it and give a summary sentence and the page # from the response to reviewers to clarify what I'm referring to.

As before, my concerns center on the multi omic analysis and its applicability to routine BTK resistance, and not on the attempts to overcome this with proteasome inhibition.

1st Response. Regarding model suitability for CLL. The authors agree that their model is "therefore resembling an advanced or aggressive form of CLL more similar to accelerated or transformed disease". This affirms my concern that the findings, while intriguing, may have little relevance to BTK resistance in non-advanced forms of CLL. This cannot be remedied. It doesn't make the findings incorrect, but raises concerns regarding their relevance to the typical BTK resistance they are trying to study. As they note the "frequency of this type of disease may be low, but represents patients with the highest clinical need." I concur, that it reflects a rare but difficult patient population. So, the findings may have value, but to a smaller subset of patients than implied, and not to general BTKi resistance.

2nd Response. Regarding comparing 3 vs 6 week time points and "time in mouse" changes. The authors have adequately clarified that they are using the peripheral blood CLL cell count to align the samples that they are comparing, and the BTK treated mice have a lag before outgrowth. I can accept this. I think they need to add more text about this model and this why this endpoint was selected for disease state alignment to the manuscript and most readers will not be intimately familiar with this model.

3rd Response. Regarding lack of phosphoproteomics. Adequately addressed in revised manuscript.

4th Response & 5th Response. Regarding former fig 2B interpretation and the different protein patterns of the different condition, and the lack of mRNA and protein correlation. The authors responded that the prior manuscript referred to the wrong figure, and redirect me to S2D. The text here and elsewhere makes the point that the I-E cells differ from the I-L cells, in that the I-E are not proliferative, and the I-L, having escaped BTKi suppression are now proliferative again. In the prior review I was concerned that we couldn't tell what changes were truly related to resistance vs. time in mouse and other effects. The response shifts my concern to be that now we cannot tell what changes are related to suppressed proliferation (I-E) and which to actual resistance in the I-L cells. The data (fig S2D for RNA-GEP and Response page 7 for protein) shows that the I-E cells, which decrease their proliferation rate in their initial response to BTKi, clearly move away from the V-E cells and that the V-L cells drift a bit from the V-E over time (a time in culture effect for the control cells), and that the I-L cells, once they have regained proliferation, move to a PCA space close to that of V-L cells. So, to me this says that the majority of the detected differences are reflective of proliferation state, and possibly buried within this are changes related to resistance. The analysis as performed did not try to separate what are proliferation state related changes, which are likely unrelated to BTKi resistance, and which DEG or protein are likely related to resistance. This would be a lot of work to perform, but I think that is crucial to teasing out the changes that they want to identify. Perhaps there are changes in I-E vs V-E that are also present in I-L vs V-L, as opposed to those common to I-L and V-L, and maybe these represent resistance, rather than proliferation?

I did not understand what point the authors were trying to make with the up and down regulated transcript/protein correlation plots on page 6 of the review when they say that this suggests "a prominent post transcriptional impact on deregulated transcripts". To me, there is no significant correlation in either.

6th Response. Regarding PCA plots for protein (Response page 7 lower). Generally these responses raise the same concern that most of what we are being shown are proliferation related changes, and that they have not teased out what are related to actual resistance.

7th Response. Regarding total proteome vs phosphoproteome. (Response bottom 8/top page 9) . Adequately addressed in

revised manuscript.

8th Response. Regarding figure 5A and rearranging it. page 9. I think the “guided” heatmap (middle figure top of page 10) better shows both the variability in the 3 naïve UT and 3 naïve CZL samples, and better shows that Res UT and Res CZL are heterogeneous within each group, and share many commonalities across groups, than the prior and current figure 5A. Is there overlap between the proteins in the yellow, magenta and green boxes in the right most figure on page 10, and the proteins that are appearing in the volcano plot fig 5B ? Minor comment: Fig5B volcano plot has 5 colors, but the description for red is not included in the legend.

9th Response. Regarding figure 6B. page 10. Corrected in revised manuscript.

10th Response. Regarding I-sensitive comparators. page 10. As expected, this is not a sample they can access, so this cannot be further addressed. No remaining issue.

11th Response. Regarding discussion. page 11. Corrected in revised manuscript.

Minor comments were addressed.

Reviewer #4

(Remarks to the Author)

Authors have addressed my prior comments.

Point-by-point response letter to Reviewers' comments

Reviewer #1, expertise in CLL models and multi-omics (Remarks to the Author):

The manuscript written by Arseni and colleagues describes experiments in a mouse model of ibrutinib resistance to investigate if proteasome inhibition could be a potential salvage therapy. Later on, also patient samples of ibrutinib resistance CLL are included to confirm and translate their findings. By applying transcriptomics and proteomics on different timepoints in the mouse model, the authors identify the proteasome activity as driver of ibrutinib resistance. Treatment with CFZ is then tested on different timepoints and shows efficacy after ibrutinib resistance as a salvage regimen. Finally, proteome profiling is performed in CLL patient samples on different timepoints before and upon ibrutinib resistance which confirms the findings of their preclinical model. Finally, proteasome inhibition shows to effectively induce CLL apoptosis in vitro.

The idea of combining proteasome inhibition with ibrutinib has been around for some time, the novelty is that the authors wish to uncover mechanisms of ibrutinib resistance and arrive at proteasome regulation. The authors made a very extensive effort to characterize the ibrutinib resistance cells, although truly integration of the multi-omics is lacking and the datasets are more standalone per experiment. The translation of their preclinical data to patient samples is interesting and relevant for the field. I think there are a few major issues to be resolved before publication would be appropriate. Please see my suggestions:

1 - In line 147-150 the authors state that the I-early group clusters away from the others which is evident in the PCA plot in S2B. However the heatmap in fig 2B clearly shows that I-late is distinct from the others. The transcriptomic data then does not match the proteomic data in 3A showing again that I-early clusters away from the other groups. Is this correct and is the data reliable?

Response: Thank you for the thorough revision of our manuscript and your highly valuable feedback. The PCA plot in Fig. S2D is based on raw transcriptomic data, whereas the heatmap in Fig. 2B uses z-score normalization on rows. The latter might emphasize different aspects of the data, highlighting I-late distinct features. However, the PCA plot of the transcriptome data in Fig. S2D aligns perfectly with the proteomic data in Fig. 3A and the respective PCA plot of proteomic data (New Figure S2I), both showing that the I-early samples cluster away from the other groups.

To address this comment in our revised manuscript, we have added a new PCA plot of the proteomic data in the new Figure S2I and commented the results in text on page 8.

2 - In line with the previous point, it would be helpful if the authors share in more detail the overlap between the transcriptome and proteome besides a broad pathway that is dysregulated. Are similar genes/proteins changing in the same direction? From the pathway plot in figure 2D it's not clear if direction is taken into account, are the majority of genes in the proteasome

pathway upregulated or downregulated in the ibrutinib resistant mice? The authors state in line 175-182 that proteasome regulation pathways and ubiquitination are downregulated in ibrutinib resistance, it seems counterintuitive that then proteasome inhibition could be effective. It would be helpful if the authors discuss this more clearly.

Response: We thank the reviewer for this discussion point. In our revised version of the manuscript we have deepened into the integration between transcriptome and proteome (New Figure 2D, E, F), and we have presented novel protein clusters important for describing the molecular mechanisms of ibrutinib-resistance (Figure 3). These revisions resulted in a global improved quality of the manuscript.

To address the first point, we performed multi-omics integration and looked specifically at the directionality of transcriptional and translational regulations. In order to provide an easy visualization of such data, we generated Figure 2D, E, and F, where we provided also functional analysis of positively correlating and anti-correlating features.

For dissecting the mechanism of ibrutinib-resistance, we presented in Figure 3D two novel clusters (cluster 12 and 5) that showed higher abundance in ibrutinib-resistant vs I-early cells, (Updated Figure 3D, and S3B, C). Whereas cluster 12 proteins are considerably downregulated with disease stage in vehicle-treated mice, but not with ibrutinib treatment in sensitive cells, cluster 5 proteins are downregulated at early stage ibrutinib-treatment (sensitive cells) which is reversed in the late ibrutinib group (resistant cells). Functional analysis of proteins belonging to these clusters show the enrichment for protein ubiquitination and ubiquitin/ proteasome pathway, thus corroborating in vivo and transcriptomics pieces of evidence. The reduced ubiquitination level that we observed at the protein level in the western blot (Figure 3E) is in line with an increased proteasome activity and sustain the use of proteasome inhibitors in ibrutinib resistance.

The description of this data was added on page 9.

3 - In figure 5 the authors analyze the proteome after CFZ. One would expect quite strong differences if the proteasome is efficiently inhibited because any type of protein marked for degradation stays intact. Could it be that the proteasome is already downregulated in resistant cells and therefore further inhibition does not have much effect? Or is the hypothesis that due to ER stress and apoptosis there is more cell death but the protein composition is not strongly affected? Elaboration on the (hypothetical) mechanism here would increase the attractiveness of this manuscript.

Response: To corroborate ubiquitin deregulation upon proteasome inhibition, we performed GSEA comparing ibr-resistant samples treated with Carfilzomib or untreated. The results of this study showed a downregulation of the ubiquitin ligase complex in carfilzomib treated cells (New Figure 5D). Our treatment data show that ibrutinib-resistant cells are sensitive to Carfilzomib, and that the proteasome is not intrinsically down-regulated in ibrutinib-resistant cells. These results of murine samples are also corroborated by the human data presented in Figure 7K.

We agree that the data show a rather low perturbation of the proteome by Carfilzomib. As suggested by the referee, this could be explained by published observations showing that proteasome inhibition leads to the accumulation of proteins in tumor cells, triggering cellular stress responses like the unfolded protein response, which could then induce cell death. There is further evidence that carfilzomib treatment doesn't significantly alter global average protein half-life. Instead, it appears to selectively affect the turnover of specific protein groups.¹ In addition, it was shown that iPSC-derived cardiomyocytes show an increase in autophagy after 48 hours of carfilzomib treatment. This may help compensate for reduced proteasome function by providing an alternative protein degradation pathway.¹

4 – Even though the survival benefit of adding CFZ after ibrutinib resistance is significant, it is still a minor benefit. Also when looking at the CLL cells in the blood, there is quite some overlap between data points where some mice with ibrutinib monotherapy have less CLL burden than mice treated with the combination. Be cautious with the conclusions here.

Response: We thank the reviewer for these observations. We agree that the survival benefit with CFZ post-ibrutinib resistance, while significant, is modest. In Fig. 6A, we note that Ibr monotherapy and the combination have similar impacts on CLL burden, consistent with our conclusions. However, Fig. 6C and D highlight a significant difference in CLL development and survival benefit with the combination, which we believe warrants emphasis. We appreciate your cautionary note and have ensured our conclusions reflect the nuanced benefits observed.

We think that the “minor benefit” that the reviewer points out should be seen in the context of a potential translation of this therapeutic approach to the human setting. In particular, it would be interesting for those patients lacking any alternative therapeutic options, and it would be valuable to quantify this benefit in each individual, taking into account the intrinsic variability among different patients.

Despite a slight heterogeneity among the individual samples which is strictly related to the model and to the modality of measuring CLL cells in blood, a reduced tumor load was measured in the majority of the samples, thus resulting in a significant decrease, as shown in Figures 6C, D and in S6B.

We have revised the manuscript on page 12 accordingly.

Minor comments:

5 – Statement made in lines 76-79, suggests that CFZ has been tested in CLL and identified as a “potent drug that induces cytotoxicity in CLL [..]”. None of the references mention anything about CFZ in CLL, only later in line 87-88 this is properly described

Response: We thank the reviewer for spotting this mistake. The references have been corrected accordingly.

6 – Methods on MS analysis are overlapping from within the main text and the supplementary methods. Better would be to provide short and summarized methods in the main text and refer to the supplement for more detailed methods. A reference to the supplemental methods is also not made.

Response: We thank the reviewer for this suggestion. The methods have been edited accordingly, and a reference to the supplemental methods made.

7 – In the sequential adoptive transfer experiments it is unclear how many donor mice are used. Is one ibrutinib resistant mouse supplying the cells for all the subsequent adoptive transfer or are there multiple donors? It would be good to make sure the methods more closely align with the ARRIVE guidelines for reporting on animal experiments

Response: We thank the reviewer for this suggestion. This point has been clarified in the Methods in the main text on page 19.

Reviewer #2, expertise in CLL models (Remarks to the Author):

Arseni et al. manuscript has a high significance to the field of CLL at both preclinical and clinical level addressing the issue of resistance to current target therapy in particular BTK inhibitors. The work is clearly written and previous works acknowledged with relevant references. The work also supports the conclusions in particular regarding the in vivo part, while for the part regarding the patients additional evidence or discussion is needed (see following comments). The methodology sounds and meets the expected standards but more details are needed for the work to be reproduced.

Major comments:

- In Figure 4 is evident a general higher variability in untreated ibrutinib resistant mice, this was the case also for the mice in figure 1? In this case the graphical representation does not allow to see it. If this was the case please comment. It would be interesting to see an immunohistochemistry analysis of the clone and Ki67 to reinforce the model.

Response: We thank the reviewer for this observation and we agree that it is interesting. We provide below a representation of tumor load in blood extracted from Figures 1E and 1F at week 2, to compare V_V and I_V. From the graph it is possible to appreciate a certain variability in the untreated ibrutinib resistant samples, similar as observed in Figure 4. The variation that we see in the proteome of mice within this treatment group might explain different growth patterns of the Ibr-res cells upon retransplantation. The groups are too small to draw firm conclusions of causal links between proteome alterations and growth rate. That is why we can at that stage comment on this interesting observation, but not explain it. Further analyses that follow up on this are beyond the scope of this manuscript.

Spleens from treated mice have been processed for mass spectrometry and for flow cytometric analysis. Especially mass spectrometry requires high amounts of tissue in order to achieve robust and reliable quantification of proteins. As we decided to give priority to these methods,

which are more quantitative than IHC, we did not collect spleen sections to be subjected to this assay. This is why we cannot provide IHC analysis of the samples. But we provide Ki67 data from flow cytometry analysis (shown in Fig. 1C+D) which does not show a higher rate or variability in lbr-resistant vs vehicle-treated mice in spleen and blood, the 2 main organs of disease development. We even see slightly less Ki67 in lbr-res mice in bone marrow and lymph nodes.

- How does author count the absolute number of CD19+CD5+ cells in the tissues in figure 4? It is not described in the methods, as the data are not in line with the % of the clone it is important to understand how the analysis has been done.

Response: We thank the reviewer for this observation. The tumor load calculation is described in the “Flow cytometry” paragraph on page 2 of the Supplemental Methods. We apologize for omitting the corresponding calculation in the organs. The description has been now included on page 2 of the Supplemental Methods.

The last part is very interesting and important to reinforce the in vivo part and confirming the predictivity of the model. However, some details are missing:

- In the figure 7A a magnetic separation of CD19+ cells is represented while in the methods a cell sorting for the CD19+CD5+ is mentioned. Which is the correct method? If authors used the isolation of CD19+ cells it is important to know the % of the clone in the population analysed in order to consider the contribution of healthy b cells to the analysis.

Response: We thank the reviewer for raising this point. FACS-sorting was used to isolate murine samples, whereas the magnetic separation adopted for human samples. Upon isolation of PBMC by Ficoll, human B cells were enriched by magnetic separation via CD19+ beads (Miltenyi Biotec). After the magnetic enrichment, prior to the following steps, the % of CD19+CD5+ was defined by flow cytometry and demonstrated to be always >95%. Details have been provided in the Methods section on page 18.

- In figure 7I authors mention that they use an ex-vivo lymph node microenvironment culture system. They acknowledge the reference but no details at all reported in the methods please add. In this case again the cells are CD19+ or CD19+CD5+, see question before.

Response: The ex-vivo lymph node microenvironment culture system is described in the Supplementary material, in the paragraph “CellTiter-Glo luminescent cell viability assay”.

Regarding the cell population, we included an additional section in the “Human samples” paragraph in the Methods on page 18, where we specified that the used samples were CD19+CD5+ cells.

Minor comments:

- In figure 1,2,4 the TCL1-AT model has been established by i.p o i.v. injection it is not clear.

Response: We thank the reviewer for this observation. In some experiments, the TCL1 CLL cells were injected i.p., in others i.v., which both leads to the development of CLL-like disease in the mice. Therefore, we used this representation.

- Figure 3 panels B and C are not readable the dots with different colours are not visible author should identify a clearer way to show these results.

Response: Thank you for this comment. We have increased the size of the corresponding text in Figure 3B and C.

- In Figure 4 I would suggest to use a different colour for treated and untreated mice not just a different tone of blue it is difficult to follow over all panel B.

Response: We thank the reviewer for this observation. The color in Figure 4 have been changed according to the reviewer's comment.

Reviewer #3, expertise in CLL proteomics (Remarks to the Author):

This manuscript used a multi omic approach to identify features associated with resistance to BTK inhibitors in CLL, and identified proteasome inhibition (PI) as a potential therapeutic modality. BTKi resistance historically was a problem in a notable fraction of CLL cases, but the magnitude of the problem is decreasing as BTKi is more commonly being employed in newly diagnosed, or first line settings, so primary resistance arising from prior resistance and mutational evolution is less. It is also becoming less relevant as newer triple combinations with BCL2i +BTKi + anti CD20 antibody combinations, which lead to deeper MRD negative remissions (whether they are cures, or longer lasting needs more time) are being increasingly employed. But resistance remains a problem for some, and identifying means of counteracting it retains importance, albeit for a smaller number of cases than when the researchers likely began this project.

My first major concern about the data relates to the model, and its ability to replicate what is seen in patients, and thus be a useful guide. Most BTKi resistance (BTKi-Res) to date has involved mutations in BTK and PLCG2, as noted in line 70, and this more commonly evolves over a long time period (years). Truly “chronic” CLL has been nearly impossible to develop models for. This research team uses the TCL1-AT model, which leads to moribund or dead mice in a matter of weeks, not months to years as is typical of CLL. In the presented studies BTKi Resistance is seen with outgrowth after only ~ 6 weeks (Fig 1B), not the years typically seen in newly diagnosed CLL cases. So the “BTKi-Res” seen here does not arise slowly over time, after an initial response, and therefore its relevance to clinically seen BTKi-Res is suspect. Furthermore, after generating their BTKi-Res CLL molecular testing did not show the common mutation (or any dominant mutation as discussed below). We therefore have a model that doesn’t replicate the patterns seen in patients with respect to frame, or initial response followed by emergence of resistance, and the resistance that is generated doesn’t mutationally mirror that seen in clinical practice. Hence, my first major concern is the relevance of the findings of this model to actual clinical findings and the generalizability of these findings, from a very aggressive disease model, without commonly found mutations, to that of the actual clinical setting. The same pattern and timing are seen with two other primary tumors (supp fig 1), again suggesting that true BTKi-Res is not being selected for given the time frame.

With that concern defined, the next question is whether there are compelling discoveries worth pursuing?

Response: Thank you for your detailed feedback on our manuscript. We appreciate your insights and the opportunity to clarify and discuss the points raised regarding the relevance and applicability of the TCL1-AT model for studying BTKi resistance in CLL.

Although the TCL1-AT model progresses more quickly compared to the primary E μ -TCL1 mouse model, it mirrors critical aspects of CLL pathogenesis, such as the influence of BCR signaling and (auto)antigen-driven clonal selection.² These features are fundamental in the development and progression of CLL in patients. The model’s aggressiveness may partly be

attributed to the gain of additional driver mutations, such as MYC amplification,³ which is associated with more aggressive disease and transformation in CLL patients. This suggests that while the timeline is compressed, the underlying disease mechanisms in the TCL1-AT model are highly relevant to understanding CLL biology and treatment resistance.

In our study, BTKi resistance developed within six weeks in TCL1-AT mice, which is faster than typically observed in patients. However, the rapid emergence of resistance still offers valuable insights into the mechanisms of BTKi resistance under a controlled setting. The absence of common resistance mutations, such as those in BTK and PLCG2, suggests that other mechanisms may contribute to resistance. This aligns with findings from CLL patients showing that early refractoriness or resistance to ibrutinib is observed in a subset of patients without mutations in the pathway. In addition, the mutations in BTK or PLCG2 are often subclonal in ibrutinib-resistant patients, arguing that CLL cells without the mutations lose sensitivity to the inhibitors also in absence of the mutations.

To address the critique raised and to determine if our mouse model of ibrutinib resistance accurately replicates the human disease state, we have now generated new proteome data of six additional patients with paired samples before or during treatment and after ibrutinib resistance. This included patients with either mutant or wild-type BTK. We performed a correlation analysis between ibrutinib-resistant samples from mice and ibrutinib-resistant samples from BTKwt or BTKmut patients (New Figure 7A, Figure S7A) and showed a very high intraspecies correlation (i.e. 0.9) even between proteomes with different BTK mutational status, and a relatively high interspecies correlation of around 0.4 - 0.5. This result shows on the one hand that proteomes associated with ibrutinib resistance in our mouse model show a relatively good correlation with clinical samples, while on the other hand it suggests a very high correlation of proteomes from individuals with ibrutinib resistance independent of the *BTK* mutational status of the patients (New Figure 7A, Figure S7A). Remarkably, we observed that specifically proteasome-related proteins have a significantly higher interspecies correlation compared to the entire proteome, or groups of proteins that cooperate in other functional complexes (e.g. eukaryotic translation initiation, SWI-SNF, or MAPK complex) or belong to the same organelle (e.g. mitochondrial or ribosomal proteins) (New Figure 7B, Figure S7B). This suggests that ibrutinib-resistance, both in the mouse model and in patients with CLL, is associated with a similar regulation of proteasome-related proteins. Importantly, these results are valid in both individuals with BTKwt and BTKmut. In agreement with this, unsupervised hierarchical clustering segregates almost perfectly human CLL samples according to their disease state (i.e. untreated or resistant to ibrutinib) regardless of their BTK mutational status (New Figure 7C). This result may suggest that the mutational status of BTK play an important role in the progression speed of the disease, but patients non-responsive to ibrutinib treatment might share a common proteomic program. This is in line with observations that show that mutations in *BTK* or *PLCG2* are often subclonal in ibrutinib-resistant patients.^{4,5}

The new data and this paragraph are now included in Fig. 7 and Fig. S7 and the text on pages 12 and 13.

A second concern is the specificity of the changes observed after BTKi-TX. I think that some crucial control experiments were omitted. Fig 1 and associated text looked at vehicle vs BTKi, and saw delayed growth (multiple endpoints) but outgrowth that they feel represents BTKi-Res by 6 weeks. They only compared vehicle vs BTKi, so we cannot know if the differences that follow are limited to BTKi therapy, or are the result of therapy in general. Would treatment with venetoclax, or bendamustine, or Fludarabine/Cytosar, have led to similar results?

Response: We agree that comparing the effects of different therapies is crucial to establish specificity and would like to thank the referee for this very helpful suggestion. In Kater AP et al.,⁶ similar BTKi resistance patterns were observed in the TCL1 AT mouse model around 6-7 weeks after starting treatment with ibrutinib, aligning with our findings. Furthermore, this study included also a treatment arm with venetoclax, and resistance to treatment developed also after 6-7 weeks. In collaboration with Kater et al., we performed now proteome analysis of venetoclax-treated vs vehicle-treated mice of this study at an early (sensitive) and late (resistant) time point. Pathway analyses of this data did not identify proteasome-related alterations, but as expected amongst other pathways, regulation of apoptotic signaling being deregulated (Figure S3D).

We further performed a proteome comparison of Ibrutinib-treated and Venetoclax-treated mice at early and late time points separately. T-test statistics and volcano representations highlight the intrinsic difference of the two treatments as shown by a high number of deregulated proteins (indicated by blue and red dots). Moreover, the GO term SUMOylation was enriched at early and late disease stages in Ibrutinib-specific deregulated proteins, but not in the Venetoclax counterpart (see below).

These results indicate that the changes in proteasome activity we observed are specific to BTKi rather than general therapy effects. We will further explore these differences and include relevant comparisons in future work.

This novel data is now included as new Fig. S3D and described in the text on page 9.

Early:

Late:

The data shows that I-Long cells were resistant to rechallenge with BTKi in serially transplanted mice, suggesting that cells that were not sensitive to BTK were selected for. But are the changes seen BTK specific? Or would similar changes have been seen with a different therapy? Some comparator would allow better definition of what subsequent findings were BTK specific, vs. treatment specific.

Response: According to the new proteome data described above, we can exclude that resistance to venetoclax does induce the same alterations as ibrutinib resistance in the TCL1 mouse model. That does not exclude that venetoclax-resistant cells might respond to Carfilzomib. As seen in the TCL1 mouse model and in patients' samples with and without Ibrutinib-resistance, CLL cells show generally a good response to carfilzomib, which is also supported by published data. The main finding and statement of our paper is, that this responsiveness does not get lost in ibr-resistant cells, but is rather slightly better. Whether this is also the case for venetoclax-resistant cells has to be further explored in the future, but is beyond the scope of our manuscript.

Published data exploring resistance mechanisms to venetoclax describe that BH3 mimetic resistance is characterized by decreased mitochondrial apoptotic priming, both in PDX models and human clinical samples, due to alterations in BCL-2 family proteins.⁷ This argues for additional alternate resistance mechanisms to this drug.

There is a curious finding (line 117-18) that I-late cells proliferate in vitro, but V-late cells don't proliferate in vitro, yet these same V-L cells were leading to moribund mice that required euthanasia. So, they were clearly proliferating in the mice, but not in vitro. So does this represent a problem with the experimental system, or can we truly believe that there is a fundamental difference in I-Late vs V-late cells? Part of this may come from comparing a different treatment condition to a different control condition.

Response: Proliferation of CLL cells in vitro typically requires external stimuli such as CD40 or TLR activation. In our study, we induced proliferation using the TLR9 agonist CpG, which selectively stimulated proliferation in ibrutinib-resistant cells. This may be due to the higher sensitivity of these cells to CpG or their reduced reliance on external signals for cell cycle entry, leading to increased proliferation ex vivo.

In vivo, both vehicle- and ibrutinib-treated cells showed significant expansion at later time points (Figure 1). Unlike in vitro conditions, in vivo growth is not limited by external growth signals. Additionally, Figure 1D shows that Ki67+ cell content in various tissues (peripheral blood, spleen, bone marrow, lymph nodes) is comparable between vehicle-late and ibrutinib-late groups, indicating similar proliferation rates.

To ensure accurate comparisons of cells at similar in vivo expansion rates, we conducted proteome analyses at various time points, focusing on comparable disease stages. We categorized the disease into early (ibrutinib-sensitive) and late (ibrutinib-resistant) stages, ensuring comparable tumor loads in the blood for both treatment groups. This approach enabled us to identify proteome changes associated with ibrutinib resistance.

Part of this may come from comparing a different treatment condition to a different control condition. Notably V-L are collected at three weeks, and I-L collected at six weeks, so they are from different in vivo durations. As discussed below, the other data shows changes from time in vivo. So, how much of the V vs I-late proliferation difference is from time in culture and how much from the BTKi-Res effect? Did more time allow for selection of a more robust cell, independent of BTK effect, hence more proliferation?

Response: As illustrated in Figure 1D, despite originating from different time points, the in vivo proliferation capacity of I-late and V-late cells is nearly the same in the spleen, the source tissue of cells for the proteome analysis. It is crucial to compare cells at similar stages of the disease rather than focusing on the duration of treatment. Our previous research has demonstrated that changes in the tumor microenvironment are closely associated with the disease stage (e.g. tumor load), which in turn indirectly affects CLL cells.

I do not understand the data in line 138. When they sequenced the BTKi-Res cells they noted some cells with variants in PIK3CD and VMN2R27, but the VAF was low (<12.5%). There is a paradox here. If they selected for a unique resistant cell then it should be clonal, and dominant. But the fact that these new changes were seen in a paucity of cells argues that the majority of BTKi-TX I-long cells at 6 weeks are outgrowth of the original population, rather than selection for a resistant population. This casts further doubt that they have selected for a truly BTKi-Res cell.

Response: We argue against the selection of a genetically-modified clone as the only reason for resistance, but rather show that proteasome alterations impact on responsiveness. Also in CLL patients, mostly subclonal BTK or PLCG2 mutations are seen in ibrutinib-resistant patients. And there are several studies that argue for alterations (e.g. epigenetic) within resistant CLL cells beyond these mutations. This discussion is included on page 15 of the manuscript.

Figure 2 B (Heatmap) shows that there are clear changes between the vehicle treated cells at 1 vs 3 weeks. I see at least three blocks of genes that are red at 3 weeks and green (or much less red) at 1 week. This demonstrates that time in the mouse is leading to changes in these cells. As noted above, the BTKi-TX cells were taken at 6 weeks, and the V-late at 3 weeks. So if there are “time in mouse” dependent changes, how can we have confidence that the changes in I-late, taken 3 weeks later, don’t merely reflect additional “time in mouse” related changes?

Response: Our main point was to compare conditions where the disease state is similar, therefore comparable. As shown in Fig. 1D, although analyzed at 3 and 6 weeks after treatment start, the V-late and I-late mice show very similar tumor proliferation and comparable numbers of CLL cells. Analysis of Ibrutinib-treated mice at the same time as vehicle-treated mice (3 weeks for the late group) would have resulted in substantially different CLL cell content and incomparable tumor cell proliferation and therefore, disease state.

To address this important question, we generated now proteome data of mice treated with venetoclax, where therapy resistance was observed at a similar time point as in our study with ibrutinib. Comparative analysis of this data showed that the alterations of the proteome are different in ibrutinib- and venetoclax-resistant cells and are therefore not a mere reflection of “time in mouse”, but rather linked to the specific drug treatment. This new data is included as new Fig. S3D.

They would need a vehicle control at 6 weeks, or a BTKi-TX match at 3 weeks, for a better comparison. Since they had to euthanize the vehicle mice at 3 weeks the 6 week comparator isn’t possible with vehicle alone, but might be with an alternative treatment. So, once again, I cannot answer concerns about whether the findings are time in mouse or BTK-TX related.

Response: We agree with the referee that this question is hard to address in the Ibr treatment study, as the vehicle-treated mice reached endpoint stage disease much earlier, and at this time point the Ibr-treated mice have barely CLL detectable. Therefore, analysis of samples after the same time of treatment will be massively biased by disease stage. We instead followed the

advice of the referee and included now data of an alternative treatment, namely venetoclax. And we can clearly show that this treatment after the same treatment time did not induce the same proteome changes (linked to proteasome activity) as observed in ibrutinib-treated mice.

Minor comment, Fig 2b, a version with the gene names readable should be presented somewhere.

Response: The full list of transcripts (23.000) cannot be displayed in the figure but we provide now a new Table S2 with the matched feature list as supplementary online material.

Next they did proteomics, using cells from three patients, with three BTK TX related timepoints for each, pre, during and at resistance. Uncommented on is whether they measured total protein levels, or also measured phosphorylation and other post translational modifications?

Response: We would like to thank the referee for this comment which we agree is very relevant. As visualized in the experimental design in Figure S2E, S5A and S7C, as well as described in the methods section, the proteomic analysis did not comprise analysis of phosphorylated or differentially modified peptides. The main limitation that prevented this approach was the low input material available, especially for clinical samples.

These cases have different mutation profiles, with two having BTK-mut at relapse, and case two selecting for a TP53 mutated clone, but no BTK-mut and case three having both. So we might expect different protein profiles between them, given 3 different putative reasons for resistance. However, Sup Fig 3 shows that the protein expression correlation between any of these has almost no variation from each other. This result seems illogical to me.

Response: The extremely high linear correlation we observed between human samples is a reflection of the ~6,000 proteins identified and the fact that a minor fraction drives resistance, while a good “core” is not significantly affected. On top, these data come from a peptide labeling processing with tandem mass tag (or TMT) where different samples are pulled and analyzed together, therefore boosting the correlation among samples in the same batch. The variability within the three clinical samples, most likely developing resistance in different ways, is reflected by the rather high data dispersion observed in the volcano plots in Figure S7H, and the heatmap representation of the data in Fig. S7E.

But we completely agree with the referee, that given the genetic heterogeneity of the three analyzed patients, firm conclusions are impossible to make. Therefore, we have performed global proteome analysis of six further patients, including patients with and without BTK mutation, where we were able to obtain samples at two time points, before/during treatment and upon Ibr-resistance. We have included Figure 7A, B and C, in addition to Figure S7A and B, showing a limited proteome variability despite BTK mutational status. This new data confirms that the proteome alterations between BTK-mutated and -unmutated cases with Ibr-resistance are not considerably different, so that no segregation of these two groups of patients was

observed by unsupervised clustering (Figure 7C). We hypothesize that the mutation in BTK or PLCG2 (often subclonal in patients) contributes to resistance, but beyond that, alterations on proteome level, including the proteasome activity, are part of the resistance mechanism.

As outlined above, this new data and new comparative analyses are now included in the manuscript.

The plots in Fig4D are also confusing. In cluster 11 (up reg) and Cluster 1 (down reg), the changes are seen after only 1 week, but are somewhat more pronounced after 6 weeks. If resistance required more time to develop then this suggest that these are rapidly induced BTKi induced changes, rather than changes associated with a mutation that leads to resistance being selected for. The time frame is impossible for that. What would happen to these changes if passaged into mice and treated with V or BTKi after that? Would they see reversion in those where BTKi selective pressure is removed? Or change to yet another pattern? And would those treated with BTKi for a second course maintain this pattern? Or change further ?

Response: We thank the referee for this comment and agree that we observed proteome changes that are on the one side associated with Ibr treatment, and on the other side with Ibr resistance. The display of data allows us to look at both effects. This is why we have chosen this format.

Cluster 1 corresponds to Ibrutinib resistant-specific deregulations. Although the deregulation is somehow already visible between Ibr-early and veh-early, this deregulation is way more pronounced in the Ibr-late vs veh-late comparison.

We further identified proteins in cluster 4 and 10 shown in Fig. S3B and C which are associated with Ibrutinib treatment regardless of the early or late time point.

For dissecting the role of Ibrutinib-resistance, we presented in Figure 3D and S3B, C two novel clusters (cluster 12 and 5) identifying proteins with a differential abundance in Ibrutinib-resistant cells. Whereas cluster 12 proteins are considerably downregulated with disease stage in vehicle-treated mice, but not with Ibrutinib treatment in sensitive cells, cluster 5 proteins are downregulated at early stage Ibrutinib-treatment (sensitive cells) which is reversed in the late Ibrutinib group (resistant cells).

Functional analysis of proteins belonging to these two clusters show the enrichment for protein ubiquitination and ubiquitin proteasome pathway, thus corroborating in vivo and transcriptomics pieces of evidence. The reduced ubiquitination level that we observed at the protein level in the western blot is in line with an increased proteasome activity and sustain the use of proteasome inhibitors in Ibrutinib resistance.

Our retransplantation approach of V-late and I-late cells into mice followed by treatment with veh or Ibr showed that resistance to Ibr is CLL cell intrinsic. Our mouse and human data further suggest, that this resistance is not only driven by gene mutation and selection, but rather by changes on the proteome level, maybe by posttranslational events.

In clusters 7 and 8, there are up regulated groups in the V treated , but in the BTKi-TX these are more (cluster 8) or less (cluster 7) at 1 week, but this effect is lost with BTKi-Res at 6 weeks. This is further evidence that time in mouse is leading to changes in the cells, and the reversion of the BTKi-TX late to look like V-E (7) or V-L(8) suggest that these changes are time, not treatment related.

Response: We agree with the referee that the malignant cells change over the course of disease also in the veh-treated mice, evidenced also by a lower proliferation rate (measured by Ki67 shown in Fig. 1C+D). But we also show that this is not predominantly time-dependent, but rather stage of disease-dependent. There is a lot of published evidence that alterations in the CLL microenvironment correlate with disease stage in mice and human, and that this altered TME acts back on the CLL cells. Therefore, it is important to compare mice with a similar disease-stage rather than the same time on treatment.

As eluded to above, the proteome differences between V-early and I-early are due to the effects of the drug in still sensitive cells, whereas the most interesting changes are the ones between I-late and V-late, when cells are resistant to Ibr. Both comparisons that reflect the same stage of disease are displayed in Fig. 3B+C.

As visualized in Fig. 3D and Fig. S3B and C, proteins in cluster 7 and 8 are strongly different in V-late and I-late. These proteins go up with disease development (V-late > V-early) and go up also with Ibr treatment (I-early > V-early). But in both clusters, a strong drop back to the level of V-early was observed in the resistant I-late samples. This means, that these proteins somehow revert to or mirror the expression value observed in the vehicle early mice. This drop seems unlikely to be a consequence of time, but rather an alteration associated with therapy resistance.

In fig 3E, if Ubq proteins were measured with the MS-proteomics, then a more detailed reporting of what proteins had changes in Ubiquitination could be presented.

Response: Fig. 3E shows the differential ubiquitination of proteins in Ibrutinib late and Vehicle late analyzed by western blot. Limited by the input starting material, specific ubiquitin enrichment and analysis of the so-called ubiquitinome has not being carried out for these samples. As further validation of the western blot results, we detected a significant enrichment of proteasome-related proteins in I-late compared to V-late samples, thereby strongly suggesting proteasome hyper-activation as mechanism for ibrutinib resistance.

It is a bit confusing regarding the data in lines 160-170, is this for 1 of the three cases? The three cases pooled? The proteins that are differentially expressed in the noted clusters should be identified somewhere. A heatmap showing the protein expression by cluster for the 3 cases should be presented.

Response: It is hard for us to completely follow this comment, as there might be a typo concerning the line numbers mentioned. The text in lines 160-170 summarizes the experimental

design for the LC/MS-proteomics of mouse samples. Here four “conditions” were evaluated (vehicle early, ibrutinib early, vehicle late, ibrutinib late). Ubiquitin- and proteasome-related proteins are highlighted in the respective volcano plots in Fig. 3B and Fig. 3C. If the reviewer is referring to clinical human samples (three cases), Fig. 7I and Fig. 7J show the heatmap visualization of ubiquitin ligases and proteasome-associated proteins of all three cases averaged in the three conditions (before treatment, during treatment, and at relapse) explored for the clinical human samples.

We hope that we could answer the referee’s question.

From this they got a lead to look at PI in CLL leading to the next set of experiments in which they tested PI vs the TCL1 CLL, showing that is slowed growth in I-naïve and I-TX cells (Fig 4B) in multiple sites. They next did proteomics on BTKi naïve and treated, +/- carfilzomib (Fig 5A [minor comment the red and orange are hard to distinguish, especially when the figure is miniaturized for printing.] to ID proteins uniquely changed by CFL in BTKi-Res cells. In the figure all the prior TX are normalized to close to 0, but it would be superior to show actual fold changes, A protein change from -2 to -3 likely is irrelevant, one that goes from -1 to +1 is likely relevant, but this cannot be appreciated as normalized.

Response: We thank the reviewer for this comment. Data here are indeed shown upon median normalization (to make samples more comparable) and not upon z-score normalization (used to highlight differences between samples).

As suggested by the referee, we have generated a z-scored heatmap representation (see below), which leads to differences in the color intensities of the samples, but does not impact on the hierarchical clustering.

Next they tested PI alone or in combo (PI + BTKi) in TX naïve mice, but surprisingly the benefit of CZL was lost. It was only seen in BTKi-Res when used after BTK was stopped. Fig 6D and text (lines 235-240) claims that this was effective in slowing CLL development and led to longer survival, and while the p- values are “significant” the visual impression is that there was minimal effect and that all the mice still died with a 1 to 1.5 week delay. This does not “clearly show that CZF is an effective salvage therapy” line 239. A similar delay may have been seen with any chemotherapy and doesn’t clearly show that there is a PI specific effect. So again, a crucial control was omitted. So I find this data confusing and unconvincing.

Response: We agree with the referee that combining PI with Ibr does not lead to a benefit in the TCL1 mouse model. But if one looks at the treatment effects of both monotherapies, it becomes quite clear that the monotherapies are already highly effective, leaving not much room for improvement. A further conclusion of this observation is that there is no synergistic effect of the two therapies. This was the reason for us to concentrate on sequential therapies. Our observations are in line with clinical data, where the majority of patients with Ibr respond very

well. A large clinical need is however new therapy regimen that can be given after patients relapse on Ibr. Our data showing prolonged survival of mice when given in sequence suggests this treatment specifically for those patients developing resistance to Ibr.

Concerning the survival benefit of 1.5 weeks, we would like to comment that this benefit in a mouse model with a highly aggressive disease upon Ibr resistance is considerable. Our data further suggests that adding CFZ even at a later time point, when Ibr-resistance becomes more apparent would likely enhance the survival difference even more strongly.

To address the critique of the referee, we have now adapted the text on page 12 to better reflect our observations.

Finally they tested different PI drugs vs. samples from the three patients, with proteomics showing a set of proteins both up/down regulated during therapy, that mostly revert to the pretreatment level at relapse/resistance. It is unclear if the relapse/resistance samples were taken while the patients were still on the BTKi, or after they had been off it for a while. If they had been off BTKi for a while then these changes represent the effects of BTKi therapy, rather than changes associated with resistance.

Response: Thanks for pointing this out. Longitudinal patient samples were taken before treatment start, during treatment with ibrutinib, and when patients showed resistance to therapy. This was either before or after discontinuation of ibrutinib. But as we have not observed major differences in the proteome of all ibr-resistant samples (Figure 7C), the continuation or discontinuation of ibrutinib in these cases did not further influence proteosomal alterations of the resistant cells. We have adapted the text on page 18 to better describe the treatment status of the patient samples.

Fig 7G does show a block of proteins that are different at relapse than during therapy, or pre treatment, so some of these changes may be BTKi-Res related. They do show that all three different patients were sensitive to three different PI (FIG 7I).

Of relevance to this manuscript: Griffen, T.L., Hoff, F.W., Qiu, Y. et al. Proteomic profiling based classification of CLL provides prognostication for modern therapy and identifies novel therapeutic targets. Blood Cancer J. 12, 43 (2022). <https://doi.org/10.1038/s41408-022-00623-7> Proteomic profiling based classification of CLL provides prognostication for modern therapy and identifies novel therapeutic targets | Blood Cancer Journal (nature.com) Figure 1 in that study shows that ubiquitin proteins and especially UPR, and epigenetics proteins were prognostic in CLL, but in figure 4 UPR, and one of the epigenetic groups but, not UBq protein were also prognostic for BTKi treated cases. Epigenetic proteins were prognostic in all CLL cases and in the BTKi treated. So if the unfolded protein response is truly important in CLL, then that would go along with generalized PI sensitivity in CLL. UPR upregulation may be an adaptive response to address the stress of accumulating proteins, relying on proteasomal degradation as part of this, hence PI use blocks the effect of this adaptive response, sensitizing

the cells. But if BTKi is upregulating UPR, it is unclear why adding PI would not be effective as found in this study. I'd like to see more thought on that puzzling result in the discussion.

Response: We would like to thank the referee for pointing this out. We have now included a paragraph in the discussion on page 17 to hypothesize on the role of UPR and ubiquitin in CLL outcome and the responsiveness to BTKi.

In agreement with our Fig. 3B, Fig. 3C, Fig. 7I, ubiquitin alone has a low prediction power as ubiquitin ligases are linked also to signaling transduction. Indeed, mono and polyubiquitination on K48 and K63 branches are associated with different biological outcomes. In line with Fig. 3E K48 polyubiquitination is directly associated with proteasomal degradation. Therefore, a PFG on proteasome would have a higher prediction for Ibrutinib resistance, in line with Fig. 3B, Fig. 3C and Fig. 7J, showing a drastic change in the expression of proteasome-related proteins at resistance.

We would like to further point out, that PI is found effective in our in vitro assays with human CLL cells and in the in vivo treatment study using the mouse model. We did not observe a synergistic effect of carfilzomib with ibrutinib, but this might be explained by the high efficacy rate of BTKi in this model. Once response to Ibr got lost, carfilzomib addition allowed for a longer control of disease progression. This is likely due to the proteome changes in the Ibr-resistant cells. This is why our observations suggest, that PI represent a viable treatment option for patients that do not respond to BTKi treatment.

They should also look at another large proteomic dataset presented in Meier-Abt F, Lu J, Cannizzaro E, Pohly MF, Kummer S, Pfammatter S, et al. The protein landscape of chronic lymphocytic leukemia (CLL). Blood. 2021;138:2514–25 for other corroborating data.

Response: We agree with the referee that a comparison of our data with the data in this manuscript is very helpful and we have added a concluding statement on common targets between our manuscript and this study on page 17.

In summary, we have somewhat intriguing data and a possible therapeutic lead. But it arises from a model that has uncertain relevance to clinical CLL, with findings that may be related more to time, than therapy, with "resistance" that is not clearly related to the BTK therapy. There are too many uncertainties to have confidence in the data as presented.

Response: We would like to thank the referee for a very thorough revision of our manuscript and extremely helpful and constructive critique which helped us to improve the content and conclusions of our manuscript. We have added now additional proteome data from TCL1 AT mice treated with venetoclax to exclude that the observed proteome changes are merely a result of time, but rather specific to the Ibr treatment. We have also enlarged the analyzed patient cohort, showing now that Ibr resistance with and without BTK mutation appears similar in our proteomic analysis, suggesting that mutations in the pathway are one but not the only resistance mechanism.

Our suggestion, that Carfilzomib might be a drug candidate for CLL patients after Ibr resistance is supported also by other published studies as outlined below and now included in the discussion of our manuscript on page 16:

- Hezkiy EE et al., Cells 2022. Bioinformatic analysis to search for synergistic drug combinations to treat CLL predicts a benefit of combining BTKi + CFZ. The authors state that *“inhibitors of proteasome and mTORC1 could synergize with ibrutinib both in vitro and in vivo”*.

- Schmidl C et al., Nat Chem Biology 2019. Combined chemosensitivity and chromatin profiling prioritizes drug combinations in CLL. The authors state *“we observed an ibrutinib-induced gain of CLL cell selectivity for proteasome inhibitors, PLK1 inhibitors, and mTOR inhibitors, which was validated in several models and further supported by the high ranking of the (interconnected) proteasome and autophagy pathways in our chromatin data. These results suggest that ibrutinib renders CLL cells more sensitive to the pharmacological disruption of protein turnover and cellular homeostasis, and they provide a basis for further mechanistic dissection and/or clinical evaluation in CLL patients undergoing ibrutinib therapy”* which perfectly matches our findings.

- Lamothe B et al., Blood 2015. Proteasome inhibitor carfilzomib complements ibrutinib's action in CLL. The authors state *“we performed an ex vivo drug screen using targeted agents on CLL cells isolated from ibrutinib-treated patients... We identified carfilzomib (PR-171), a second-generation proteasome inhibitor and ABT-199 (a Bcl-2 antagonist) as the most cytotoxic agents, as indicated by increased annexin V/propidium iodide (PI) double positivity after 24-hour incubation... Cells isolated from 7 patients (Table 1) were treated for 16 hours with concentrations of ibrutinib and carfilzomib that resulted in modest toxicity individually but, when combined, showed at least an additive cytotoxic effect”*.

Reviewer #4, expertise in CLL therapy and ibrutinib resistance (Remarks to the Author):

Comments to Authors

In this project, the group develops a mouse model of ibrutinib resistance by treating mice upon adoptive transfer of Eu-TCL1 leukemia continuously with ibrutinib. Using cells from these mice with ibrutinib resistant CLL, authors perform transcriptome and proteome analyses during therapy (sensitive to ibrutinib) and after disease progression (resistant to ibrutinib). These data were compared to those obtained from 3-4 patients with CLL with ibrutinib-resistant disease. Based on omics data that suggested over-activation of proteasome activity, authors use proteasome inhibitors (PI) such as carfilzomib in mouse model to suggest utility of this drug in treating ibrutinib-resistant CLL. This was then extended to CLL cells obtained from patients. Authors need to be congratulated for creating ibrutinib-resistant mouse model. There are, however, challenges to extrapolate from this model to the disease in human that becomes resistant to ibrutinib.

Major comments

1. In humans, most common ibrutinib resistance mechanism is identified as alterations in BCR pathway proteins that include mutations in the binding site of ibrutinib (i.e. C481 substitutions), gate-keeper mutations (T474 substitutions) and down-stream PLCG2 alterations. There are some patients who develop disease resistant to ibrutinib without these mutations. In mice, in the current project, when the disease progresses and is coined as ibrutinib-resistant CLL, authors do not find any mutations in the BTK. However, authors compare mice data to 4 patient samples (Figure 7) with respect to omics and biological response to proteasome inhibitors including carfilzomib. This extrapolation needs to be revisited by comparing mice data with human data where CLL is progressed without any BTK or PLCG2 mutations. Considering these fundamental differences in mouse versus human ibrutinib-resistant cells, it is challenging to accept mouse model as a replica of human ibrutinib-resistant disease.

Response: We would like to thank the referee for in-depth reviewing our manuscript and for very helpful and constructive comments which we used to improve the paper.

Finding BTKwt resistant patients has been very challenging. Nevertheless, we were able to perform comparative proteomic analysis of mouse (BTKwt) ibrutinib-resistant cells and samples from Ibr-resistant patients with or without BTK mutation. As shown in the multi-scatterplot, proteins identified in resistant samples of the mouse model and in resistant samples of patients have a Pearson correlation around 0.5. This value is not significantly different whether the correlation was calculated against BTKwt or BTKmut human CLL samples. The correlation between human resistant samples from patient with or without BTK mutation is above 0.9.

In addition, as eluded to in the response to referee 3, the observation that mutations in BTK or PLCG2 in BTKi resistant patients are often subclonal, suggests that additional events are contributing to this resistance. Our data brings evidence for a similar proteome pattern in BTKwt and BTKmut CLL cells from patients with Ibr-resistance. Therefore, proteasomal changes upon development of resistance likely contribute to non-responsiveness to Ibr in both patient groups.

The new data and a paragraph describing the results are now included in Fig. 7 and Fig. S7 and the text on pages 12 and 13.

Taken together, these data suggest that the mutational status of BTK might be important for predicting the disease progression velocity, but at the stage of resistance the CLL cells are quite similar irrespective on the BTK status. This effect is similar to what seen for example in high grade IDH mutant gliomas, that do not rely anymore on IDH and 2-HG production.

Moreover, our data suggest that our mouse model does not show a bias in preferentially recapitulating human disease with BTKwt.

To overcome the limitation of results obtained in a mouse model, we have tested PI also in ex vivo culture models of primary CLL cells and showed that ibr-resistant CLL cells respond efficiently to PI treatment.

2. Throughout the manuscript, authors propose and suggest that ibrutinib-resistant CLL in mice respond well to carfilzomib indicating sensitization of cells to carfilzomib after disease progression. However, data suggest that carfilzomib treatment responded similarly in ibrutinib-naïve and ibrutinib-resistant disease. For this comparison, authors should statistically compare carfilzomib treated ibrutinib-naïve and ibrutinib-resistant cohorts in Figure 4D-F and Supplemental Figure 4 A-C. Similarly, in Figure 5, authors should include a volcano plot to compare carfilzomib-treated ibrutinib-naïve with carfilzomib-treated ibrutinib-resistant CLL (Res CFZ – Naïve CFZ). Currently, this is a supplemental figure. In general, it appears that both ibrutinib-sensitive and resistant CLL respond to carfilzomib.

Response: Carfilzomib has for sure a strong impact on leukemia development in TCL1 AT mice, and treated mice, also without Ibr-resistance, show a response to this drug. Nevertheless, Fig. 4 shows how, especially in the spleen, carfilzomib response is stronger in ibrutinib-resistant mice. Moreover, in comparison with ibrutinib-naïve mice, ibrutinib resistant mice have in general a higher number of CLL cells and spleen weight, yet upon carfilzomib treatment both these values sharply decreased and reach the values observed in ibrutinib-naïve mice.

The distinct effect of carfilzomib treatment on either naïve or ibrutinib-resistant mice is also highlighted by the distinct alterations of the proteome and the different gene ontologies overrepresented in carfilzomib-treated naïve or ibrutinib-resistant mice (i.e. inflammatory cytokine vs apoptosis, respectively).

Given the rather low number of proteins significantly deregulated upon carfilzomib treatment in naïve conditions (Fig. S5D), the direct comparison of proteomes from carfilzomib-treated ibrutinib-resistant and naïve mice (Fig. S5G) intrinsically shows how the ibrutinib treatment dramatically impacts on the proteome. Figure S5F corroborate this hypothesis, showing a high number of deregulated proteins between Ibrutinib-resistant and naïve conditions already in absence of carfilzomib treatment.

3. Carfilzomib was selected based on proteasome pathway overexpression data in Figure 2C. Several other pathways, such as replication, DNA repair, cell cycle etc were also upregulated. Did authors test other drugs that target these cellular processes?

Response: We agree that our data pointed to several upregulated pathways, including replication, DNA repair, and cell cycle, which could potentially be targeted with other drugs. Our choice to focus on proteasome inhibition was influenced by several factors. Firstly, proteasome inhibitors have shown promise in the treatment of CLL in both literature and clinical trials (statement in now included in the manuscript on page 16). These inhibitors specifically target the proteasome pathway, which plays a crucial role in protein degradation and cellular homeostasis, particularly in cancer cells that are highly dependent on proteasome activity for survival and proliferation.

While the manuscript does not explore drugs targeting other pathways such as DNA repair or cell cycle regulation, this was a deliberate decision to maintain a focused scope. Exploring these pathways and corresponding drugs would indeed be a valuable line of inquiry for future studies, and we agree that they hold potential for yielding promising results. However, it is important to consider the potential toxicity of targeting broad cellular processes like DNA replication and cell cycle regulation. Proteasome inhibitors, on the other hand, have demonstrated a more favorable toxicity profile in clinical settings, making them a more viable therapeutic option for CLL.

4. Similarly, other targeted agents such as venetoclax should be tested for treatment of ibrutinib resistant mice CLL cells.

Response: Clinical data show that the majority of CLL patients with ibrutinib resistance respond to venetoclax therapy. Therefore, we assumed that this is also the case in the TCL1 mouse model. As our aim was to find alternative novel therapies for ibrutinib-insensitive patients, we searched for novel targets.

We have now performed proteome analysis of TCL1 mice treated with venetoclax which leads to therapy resistance in a similar timeframe as with ibrutinib, and observed that the two drugs deregulate different proteins (both at early and late time points). We did not identify a link between venetoclax resistance in proteasome activity, as seen for ibrutinib resistance. Interestingly, at both time points of this comparison, only ibrutinib treatment was associated with an increase in SUMOylation. Based on this one might speculate that proteasome inhibitors might work better as second line therapy in ibrutinib resistant compared to venetoclax-resistant patients.

This data is now included as new Fig. S3D.

5. For data in figure 1C and D, authors should compare week 1 vehicle treated with week 3 vehicle treated cells. Why do vehicle-treated cells show less Ki-67 stain at week 3? Did authors observe lymphocytosis after treatment of diseased mice with ibrutinib?

Response: We would like to thank the referee for pointing this out. Indeed, in a comparable treatment study with ibrutinib in the TCL1 mouse model by Kater AP et al., lymphocytosis was observed 0.5 weeks after treatment start in the mice (see Figure 1 below).⁶ We have no clear evidence for this in our own data, but that might be due to not analyzing blood samples at such an early time point. Our first analysis was after 1 week of treatment, a time point where also in the published study lymphocytosis was not evident any more. Therefore, the higher frequency of Ki67-positive CLL cells at the early compared to the late time point might be due to mobilization of CLL cells from the tissues in the first few days of treatment. But this conclusion has to be done with caution, as comparisons of flow cytometry data that were acquired at different days are not always reliable, especially if the analyzed signal does not resolve a positive and negative cell population, but rather leads to a shift of the whole cell cluster.

Figure 1 of Kater AP et al: Treatment groups: red – untreated; blue – ibrutinib; green – venetoclax; yellow – ibrutinib + venetoclax combination

6. Number of CLL patient samples is very low (Figure 7). Authors should add additional patient samples including those that are resistant to ibrutinib without any BTK or pLGG2 mutations.

Response: We agree with the referee that the cohort of patients with ibrutinib resistance included in our manuscript was small and that the comparison of patients with and without mutations in BTK or PLCG is of high relevance. That is why we added now analyses of six more ibrutinib resistant/refractory patients, 4 BTKmut and 2 BTKwt, of which we performed proteome analyses before treatment and upon resistance. Our proteome data show that patient samples cluster (almost perfectly) according to their disease state (untreated or ibrutinib resistant) irrespective to their BTK status (shown in new Fig. 7C). In addition, we performed a correlation analysis between ibrutinib-resistant samples from BTKwt or BTKmut patients (New Figure 7A, Figure S7A) and showed a very high correlation (i.e. 0.9) even between proteomes with different BTK mutational status. Although not significant, the median correlation between Ibrutinib-resistant samples from patients with BTKmut and BTKwt (figure below; light blue, 0.94) is slightly higher than the respective correlation at the untreated state (yellow, 0.84). This data is in

line with patient proteomes at relapse state being more “close to each other” irrespective from BTK status, compared to the proteomes of BTKwt and BTKmut at untreated state.

The inclusion of this new data strengthens our statements and shows similarities in the resistance mechanisms of BTKwt and BTKmut patients. We hypothesize that the mutation in BTK or PLCG (often subclonal in patients) contributes to resistance, but beyond that, alterations on proteome level, including the proteasome activity, are part of the resistance mechanism.

7. Authors should elaborate data regarding carfilzomib monotherapy results in patients with CLL (reference 29). It appears that there was very limited activity of carfilzomib in CLL.

Response: Thanks for pointing this out. We were aware of the limited response of CLL patients to carfilzomib treatment. That is why we have tested in our study a sequential treatment of CFZ following ibrutinib-resistance. Based on our findings, the proteasomal changes observed in ibr-resistant CLL might result in improved treatment responses.

Minor comments

1. Based on text for Figure 4B, ibrutinib-naïve should be on the right panel.

Response: We thank the reviewer for spotting this. We corrected the figure legend accordingly.

2. On line 87, previousl should be previously.

Response: Thank you. We corrected the typo.

3. On line 288, it would be important to cite Honigberg paper (ref 12).

Response: We thank the reviewer for this suggestion. The reference has been included.

References for the Response Letter

1. Currie, J., *et al.* Simultaneous proteome localization and turnover analysis reveals spatiotemporal features of protein homeostasis disruptions. *Nat Commun* **15**, 2207 (2024).
2. Floerchinger, A. & Seiffert, M. Lessons learned from the E μ -TCL1 mouse model of CLL. *Semin Hematol* (2024).
3. Öztürk, S., *et al.* Longitudinal analyses of CLL in mice identify leukemia-related clonal changes including a Myc gain predicting poor outcome in patients. *Leukemia* **36**, 464-475 (2022).
4. Burger, J.A., *et al.* Clonal evolution in patients with chronic lymphocytic leukaemia developing resistance to BTK inhibition. *Nat Commun* **7**, 11589 (2016).
5. Ahn, I.E., *et al.* Clonal evolution leading to ibrutinib resistance in chronic lymphocytic leukemia. *Blood* **129**, 1469-1479 (2017).
6. Kater, A.P., *et al.* Combined ibrutinib and venetoclax treatment vs single agents in the TCL1 mouse model of chronic lymphocytic leukemia. *Blood Adv* **5**, 5410-5414 (2021).
7. Bhatt, S., *et al.* Reduced Mitochondrial Apoptotic Priming Drives Resistance to BH3 Mimetics in Acute Myeloid Leukemia. *Cancer cell* **38**, 872-890.e876 (2020).

Point-by-point reply to reviewers's comments

Reviewer #1 (Remarks to the Author):

The manuscript has been significantly improved and the authors have sufficiently addressed my comments.

Reviewer #2 (Remarks to the Author):

Authors addressed clearly to all the questions I raised.

Reviewer #3 (Remarks to the Author):

TO THE AUTHORS:

I have read the response to the initial review and can see that the authors extended significant effort to try and address the critiques. In some ways it is improved, but in many ways I still have the same concerns that I had in the first review.

Comments on "Response to Reviewers"

I've read their points about the validity of their model, but still have major concerns about its applicability to clinical ibrutinib resistance.

1. As noted before, this is resistance for single agent Ibrutinib, but CLL therapy has moved on to doublets and triplets with venetoclax and obinituzumab and acalabrutinib, yielding unprecedented deep MRD negative remissions, that may prove to be cures. There is significantly less resistance with these newer therapies. Consequently single agent resistance is less relevant.

2. As noted before, clinical resistance to ibrutinib takes months to years to develop, and has characteristic mutations in most. Their model arises in weeks and lacks these mutations. So, while I think they performed their experiments well, and I concur with the conclusions they derived from the data, the relevance of this model to clinically seen resistance remains uncertain to me. As they note, there is a small set of early resistance patients for whom this may be representative.

Response: We completely agree with the referee, that the treatment landscape for CLL patients has completely changed within the last years and we are aware that the majority of patients respond very well to ibrutinib and its combination with additional drugs. Also, the addition of venetoclax to the treatment regimens has further improved outcome, leading now often to situations that are close to curing this disease. We would like to point out, that the E μ -TCL1 adoptive transfer model that we are using in our study, is likely not reflecting CLL disease with long-term responsiveness to these treatments. This model is based on transplantation of malignant B cells which have already adapted throughout the development of a CLL-like disease during the lifespan of a mouse. Among the adaptations is a gain of Myc activity, as we have described before (Öztürk et al, 2022). The transplantation model is therefore resembling an advanced stage or aggressive form of CLL, more similar to accelerated or transformed disease (reviewed by Floerchinger and Seiffert, 2024). Even though the frequency of this type of disease

is low, it represents patients with the highest clinical need, as they do not benefit long-term from ibrutinib or venetoclax treatment or novel combination therapies. In the clinic, these patients stand out as the cases showing early resistance to ibrutinib, without mutations in the pathway. Our study suggests that patients with this aggressive form of CLL will benefit from treatment with carfilzomib.

We have adapted our manuscript to better reflect the clinical relevance of the used model and the obtained data of our study.

I didn't feel that their responses to the issue of comparing V-L to I-L cells from 3 and 6 week timepoints really answered what was due to time vs what was due to ibrutinib.

Likewise, the point about time in mouse changes vs ibrutinib induces changes (original fig 2B) suggests that they are comparing identical disease states and that the time is irrelevant, but I didn't see data demonstrating that these were similar disease states as claimed. They also don't clearly define what these "different states" are and how they know that these are differ/similar.

Response: We would like to thank the referee for pointing this out. We observed both for transcriptome and proteome data that I-early cells are most distinct from the other groups (Suppl. Fig. 2D and I) which is likely due to their non-proliferative state induced by inhibition of BTK by ibrutinib. In contrast to that, ibrutinib-resistant I-late cells resemble more V-late cells, as both these cell types are highly proliferative and rapidly expanding in the mice. We agree with the referee that the cells also change with the course of disease. This is seen by the differences observed in V-early vs V-late cells (Suppl. Fig. 2D and I). From many years of experience with the TCL1-AT mouse model, we know that most alterations in the CLL microenvironment are clearly correlated with the disease stage, which we define as the number of TCL1-CLL cells in the blood and lymphoid tissues. For comparative analyses of CLL-related features, it is therefore essential to analyze samples at a similar stage of disease. This is why we analyzed in our study V-late at week 3 and I-late at week 6, which reflected samples with a similar number of CLL cells (as can be observed in the data represented below) and proliferation rate in spleen (Fig. 1D).

Figure legend: Spleen weight (left panel) and absolute count of CLL in spleens (right panel) of V-late (veh) and I-late (lbr) samples. Data are mean \pm SD and were analyzed by Mann-Whitney test; p values are reported.

The lack of phosphoproteomic data limits the findings. This limitation should be commented on in the discussion

Response: Thanks for this comment. We have added a paragraph to the discussion describing the lack of phosphoproteomic data as limitation of our study.

Comments on new manuscript

The manuscript has two sections. The first part used transcriptomic and proteomics to assess what changes in a cell line defined as being Ibrutinib resistant after short term Ibrutinib exposure. From this they identified protein degradation as a possible therapeutic weakness for the I-resistant cells. In the second part they conduct experiments to show that proteasome inhibition therapy works on the I-resistant cells. In both the first and second reviews, most of my concerns relate to the first part.

PART 1: Multiomic analysis of Ibrutinib resistant cell line.

On pages 5-6 data. the issue of whether the resistance is "acquired" or "selected" for is evaluated. It's too quick for new mutations, which would require that it was selection for a small sub clone with existing mutations. But since they don't see mutation it must be acquired. They later posit that it is the transcriptional and protein profiles that show the adaptation, arguing that it is not mutation causing resistance, but rather is the protein signature.

(Please see attached figure)

In Figure 2B. They claim (page 7 line 154-158) that this figure shows that I-E cluster separately from V-E V-L and I-L samples, but then say that I-L (Line 157-158, "loss of responsiveness to I" = I-L) look like Vehicle V-E and V-L treated cells. I disagree that VE, VL and IL look alike. I looked at the heatmap and by eyeball boxed areas in which the expression was much higher in I-E vs. V-E (Magenta Boxes ABCDE) then areas in VE- that were higher than I-E, but that looked like V-L (Light blue boxes H & F), and G which is common to V-E, V-L and I-E but not V-L, then those that were high only in V-L relative to I-E (gray boxes I,J,K,L, K). And finally for V-L identifying two boxes of high and low expressed genes (yellow box M,N). It is clear that none of these four look like each other.

I agree that I-E is unique as it shares only region G with V-E, only B and E with V-L and virtually nothing with V-L.

B
It's also clear that V-L looks nothing like V-E cells, which implies that either time in culture, or the vehicle is inducing many changes. This is also true from Supp fig2 PCA where VE and VL are in different spaces. VE is close to the reference cells in Sup2I, but V-L are far away, again suggesting a strong effect of the time in culture, or a vehicle induced effect.

Furthermore it is overwhelmingly clear that I-L look nothing like the other three groups, and most importantly, nothing like the V-E cells.

This poses a major unexplained dilemma: If the protein pattern, achieved either by induction or selection, is crucial for resistance in I-E cells, and if the I-L cells are maintained under the same selection criteria, by continued exposure to Ibrutinib, until they were retested, then how can the I-L cells look nothing like (and in most cases, the exact opposite of) the I-E samples? This defies logic. If it is important for resistance, and exposure continues, why doesn't the expression pattern continue?

Response: We would like to thank the referee for the detailed analysis of our data and the helpful comments. We agree that the heatmap in Figure 2B (shown above) does not allow to conclude that I-early samples cluster separately from the rest. The figure that supports this statement is Figure S2D which presents the unsupervised cluster and PCA of the data. The heatmap in Figure 2B instead displays the most variable transcripts between the groups, and here it becomes clear that all four groups are

distinct with a set of differentially-expressed genes distinguishing them from the other groups, and for I-late cells the number of DEGs are highest. We have adapted the text to refer now to the right figures for our conclusions.

As explained above, it is not unexpected that the TCL1-CLL cells change their transcriptome with the stage of disease (as observed for many features in CLL development in this model). This likely explains the differences in gene expression between V-early and V-late groups.

We do not see a dilemma in the differences observed between I-late and I-early cells, but these differences are based on the extremely different functional properties of the cells. Due to the inhibition of BTK-dependent pathways (BCR, chemokine, TLR and likely others) by ibrutinib in I-early cells, they do not proliferate but are rather quiescent and non-expanding cells. I-late cells lost the dependency on BTK and proliferate and expand even in the presence of ibrutinib. This difference in proliferation and likely other functional properties is the reason for the large amounts of differentially expressed genes and proteins. The differences are acquired in I-late cells as a consequence of the adaptation and non-responsiveness, and are not present in I-early cells which are not resistant to ibrutinib.

We hope this clarifies the concerns of the referee. We have adapted the text on page 7 to improve the description of these data and findings.

Figure 2E shows a complete disconnect between protein expression and mRNA expression. There are as many completely inversely correlated proteins in I-L-Pro w.r.t. I-L-RNA as there are strongly positively correlated proteins. The other three comparisons also seem to show the same low correlation. (the correlation coefficient for these should be stated). In further support of this fig S2D (PCA of protein transcription) shows that V-L and I-L are closest, while in S2I (protein PCA) V-L and I-L are far apart. Furthermore, S2I shows that I-L revert to be close to the reference and V-E profiles, suggesting that the I-E changes are lost. This argues against the selection for, or induction of a protein expression pattern that confers resistance. So there is neither mRNA-Protein correlation, nor PCA pattern correlation between the samples in mRNA vs. protein. There is ample evidence in the literature that mRNA is not equal to protein expression, so this reaffirms what others have seen. But, since it is the protein that acts, not the mRNA, this then makes the mRNA derived data for the rest of the manuscript suspect as we cannot know if high transcription means high protein, or the inverse. Furthermore there is the extremely puzzling observation on PCA, that the early changes leading to the I-E protein pattern needed for resistance is lost with additional time, despite continued "pressure" from Ibrutinib and reverts back towards baseline with additional time.

Response: As explained above for the transcriptome data, the differences in I-early and I-late cells are due to their different cell states (quiescent vs proliferative). I-late cells are likely closer to V-early cells because both of these cells are proliferative, whereas I-early cells are not.

The most important point to clarify the criticism of the referee is that we do not expect to see changes associated with resistance in I-early cells. These cells are non-proliferating and quiescent as a consequence of BTK inhibition and therefore, likely very different from the other groups. Changes that are linked ibrutinib resistance are expected in I-late cells that do not respond to ibrutinib.

We agree with the referee that mRNA is frequently not equal to protein expression (which is confirmed in our data) and that protein data is more important than mRNA in terms of cellular behavior. This is why we did not continue in our manuscript using the transcriptome data, but rather focused on the proteome analyses.

Related to the correlation between transcriptome and proteome data, the calculated Pearson correlation of all matched features ($\sim 4,000$ as from Fig. 2D) shows a linear correlation of ~ 0.3 between the expression level of the transcripts and the respective protein abundances (average values of expression or protein abundance per condition are shown in the multi-scatterplot below, left panel). Similar values are obtained for all tested comparisons. This is in line with data reported in a number of other studies where, because of post transcriptional and translational processes, the linear correlation between transcriptome and proteome is expected to be around 0.3 – 0.4.

Regarding the correlation between the expression level and the protein abundance in ibrutinib-late condition, for features significantly deregulated at the transcript level (up- or down-regulated transcripts), the Pearson correlation is around 0.2, thus suggesting a prominent post transcriptional impact on deregulated transcripts (please see right panel below).

For what concerns the discrepancy between Fig. S2D and S2I, this is ascribable to the different number of features between transcriptome and proteome analysis, as well as at the expression level of the respective features, resulting in a global Pearson correlation of about 0.3. This results in different components being selected in PCA analysis, for the separation of the sample groups. As demonstration, the figure below shows the PCA plot for the $\sim 23,000$ transcripts as from Fig. 2. The left panel uses the component 1 and component 2, and resembles the PCA plot represented in Fig. 2D, where component 2

separates I-early, while I-late and V-late are the closest along the component 1. The PCA on the right uses component 2 and component 3 (subsequent in power of discrimination), and shows I-late being closer to V-early, corroborating Fig. 2I and, very importantly, Fig. 3A. We can therefore speculate that the dissimilarity between expression level and protein abundance might have guided towards the selection of a different set of components for the data compression and visualization of transcriptome and proteome data, thereby giving rise to two slightly different visualizations as from Fig. S2D and S2I.

The authors then look to show that the proteome Page * line 183. Sup3A PCA now has IE and VE, which were very distant in S2I, close together, and far from the two late samples, which curiously site close to the reference samples. If I exposure leads to the massive transcriptional changes seen in Fig 2B, and the very different protein profiles seen in Fig 2E, then why are the two early clustering together in S3A? This again makes me think that there is a lot of time in culture effect. Harder to explain, if exposure to I is leading to selection of a resistance profile, is why the VL and IL should be close to each other in S3A, and closer to the reference samples. This further argues against the idea that there is a unique I induced protein expression pattern being detected here.

The lack of consistency between the trends in the data from the different omics raises serious concerns.

Response: PCA and uMAP are two dimensionality reduction techniques that capture the most important pattern either by finding new variables (principal components), or by preserving the relationships between data points. In light of this, visualization of the very same data with these two strategies may differ. In our analyses, we use the uMAP and PCA as sanity check ahead of the hierarchical clustering reported in Fig. 3A. In particular, both uMAP and PCA were adopted to ensure that the batch effect observed in the raw data from the two TMT-labeled proteomes (please see figure below, top panels)

was cancelled out with our Limma-based batch correction algorithm (figure below, bottom panels). Although both PCA (especially component 2) and uMAP show a segregation of early and late samples, the difference between uMAP and PCA is purely linked to their dimensionality reduction algorithm.

It is general consensus that dimensionality reduction techniques are very valuable tools for data visualization, but unsupervised hierarchical clustering are superior in exploring multiple levels of data at once, and identifying data clusters more in details without recurring to data compression. In view of this, throughout our manuscript, we rely on unsupervised hierarchical clustering where all data are equally taken into consideration when reconstructing sample groups. In addition, the dendrogram visualization provides a very useful information on clusters distance or similarity.

Batch corrected proteomic data, visualized via either PCA (Fig. S2I) or uMAP (Fig. S3A), give rise to the unsupervised hierarchical clustering (Fig. 3A) visualized as heatmap representation. Dendrogram of the heatmap in Fig. 3A quite accurately resembles the PCA below (and Fig. S2I) with I-early being distant from the rest, as well as V-early and I-late being close to each other.

Part 2: Proteasome inhibition in Ibrutinib resistant cells.

Page 9, line 219-235 & Figure 4. I agree with the interpretation. CZL alone had minimal effect on I naïve cells, but did inhibit growth (spleen, node size) in the I-resistant cells.

Page 10-11 lines 237-261. Figure 5. Proteomics in CZL treated -I-resistant cells.

Need to note line 240 that these are total proteins and that phospho, or other PTM are not assessed.

Response: We added a note to clarify that we analyzed only total proteome and also added this as limitation of the study in the discussion.

For Figure 5A. Per the legends at the lower left, this is total protein abundance. The authors confirmed in the “response to reviewers” that this was indeed normalized based on the expression of the resistant cells, but I still don’t see that mentioned in either the methods or figure legend. I find the unnormalized figure in the RTR (page19) superior to the one used in the manuscript. For one the samples are labelled at the top (Naïve 1 2 3 , Resistant 1 2 3 4, UT/CZL) so that we can compare how a sample did/did not change with CZL treatment. That cannot be done with the current fig 5B. 5. Since the areas below the arms of a dendrogram can be freely spun to improve visual analysis, without changing the statistical implication of the clustering I would suggest flipping the order to move the left most ((NAÏVE_CZL_R3 to the far right next to NAÏVE_UT-R1. I would also reverse From NAÏVE-CZL-R2 to RES_UT_r1. This would put the Res together on the left and the Naïve on the R. Once reordered, it is clearer that there is not a lot of change with CZL tx in either set, but that the Res cells seem to maintain their starting point (compare UT vs CZL) and the UT tend to maintain their starting profile (UT vs CZL) , supporting the contention from part 1 that resistance selection/induction induces a protein expression profile and that this is maintained over time Fig 5B. The purpose of this figure (line 244-245) is to ID a set of proteins that were specifically deregulated I the I-res cells upon CZL treatment.

Response: Starting from the unsupervised hierarchical clustering visualized in Fig. 5A (please see figure below, left panel), we generated the corresponding “guided” visualization, where the order of the columns has been pre-defined according to the reviewer comment (figure below, central panel). As expected, a side-by-side comparison between these two heatmaps shows how the row nest clusters are significantly changed between fully unsupervised and “guided” visualization. In addition, we firmly believe that the suggested visualization lacks an important piece of evidence about the internal variability observed in the naïve samples. The dendrogram of the columns in the heatmap from Fig. 5A (left heatmap in the figure below) captures this important information.

As asked by the referee, in Fig. S5D and Fig. 5A we report the volcano plot showing the effect of carfilzomib-treatment in naïve and resistant cells, respectively. Upon carfilzomib-treatment, a number of deregulated proteins were identified as significantly deregulated upon t-test statistics. These proteins correspond to a fraction of proteins showing a differential abundance, as it can be deduced from the unsupervised hierarchical clustering heatmap visualization below (right panel, colored boxes) comparing resistant untreated (UT) and carfilzomib-treated samples. This result shows how carfilzomib is able to induce a protein deregulation also in resistant samples, and corroborates phenotypic in vivo data provided in this manuscript.

Page 11-12 Line 263-278 CZL as therapy for CLL. I agree with the laboratory work and interpretation until line 277. In fig 4 it is claimed that there was a significant decrease in growth (Fig4b ****) and in Spleen and node size (Fig 4 C E * and D, F **** and ***) in ibrutinib naïve cells, in some cases a statistical magnitude equal to that of the I-resistant cells. Yes in fig 6B & C adding CZL to V treated cells had no effect. So this is a discrepancy to be commented on. Why should CZL only work in I-res cells and not V in the fig6 experiments when it seemed to work in the fig 4 experiments? The lack of consistency needs some explanation.

Response: We apologize for an unclear presentation of the results that resulted in this misunderstanding. CFZ was only added to I-treated cells in this study, but not to vehicle-treated cells. This is why we do not see a response in the vehicle group. We have adapted the presentation of this data to make this clear, and the color code has been indicated in the figure legend.

Page 12-13, lines 281-337 Proteomics of patient derived I-res. Small numbers, but I don't think this can be helped. There is a question of what is treatment time related vs I-resistant related, similar to the issue with the mouse models (Line 310). This data might be stronger if you had a comparator from patients that were I-sensitive and maintained sensitivity (obviously you can probably only get the pre treatment and maybe 3 month timepoint samples, as they were unlikely to have many circulating cells by 6 months).

Response: We agree with the referee that a comparative proteome analysis of samples before and during treatment with ibrutinib, as well as upon resistance development is highly interesting. That is why we included such data for 3 cases. As outlined by the referee, the time point during treatment (ibrutinib-sensitive cells) was chosen within the first 6 months of treatment, as at later time points, numbers of CLL cells in blood are too low to perform such analyses.

Discussion

As they note on lines 344-345 on page 14, the time course of their model and clinical resistance are very different, so their own comments raise questions about the relevance of their model to the clinic. I think they need to stress this more in the discussion, perhaps as a lead in to line 372 when they argue for alternative mechanisms.

Because there is conflicting data (Fig4 and 6) regarding whether PI works in I-naïve cells, its unclear if PI should be tried in all CLL cases, or only those with their variant form of I-resistance. Their data using I-res cells from patients with the typical BTK mutations suggest that it could be all I-res cases.

Response: We hope that we could emphasize with the explanations above and the changes in the manuscript the relevance of the TCL1-AT model for CLL patients with aggressive disease who develop ibrutinib resistance early and without mutations in BTK and PLCG2. As suggested by the referee, we have added a paragraph to the discussion to explain this.

In addition, we hope that we have clarified with our response above that there is no conflict in data shown in Fig. 4 and 6. PI work both in I-naïve and I-resistant cells, independently of the mutational status in BTK or PLCG2.

SUMMARY:

We have some interesting data that supports looking at PI in CLL, possibly in I-resistant cases, which supports the findings of others. But this is based on multiomic data that is confusing, and inconsistent in places, using a model with uncertain clinical relevance.

Minor comments.

Some color choices in the figures are still hard to discern.

Response: We have now a homogenous color-code throughout the paper to better discriminate sample groups.

Lines 145. It remains difficult to say that these mutations are crucial when they are present at a low VAF. Supplemental 1C seems to claim that 4 deleterious SNV were seen in all 3 cases. Why is that not mentioned here?

Response: We have improved the description on page 7 on the deleterious SNVs detected in all cases.

Supp fig 2D seems to be missing a I-lates dots. Others have 4, but only 3 for I-late. +

Response: We thank the reviewer for this observation. There are no missing dots in the figure. The three Ibrutinib late samples from which we generated transcriptome data are shown in the figure.

Reviewer #4 (Remarks to the Author):

Comments to authors

Authors have revised their manuscript by rebuttal comments from all four reviewers. Want to thank them for adding additional patients' data in the current work. Revised manuscript is improved and strengthened to support conclusions. However, there are a few comments that need to be addressed.

Major comments

1. Throughout the manuscript, including title, authors state proteasome inhibition and proteasome inhibitor for ibrutinib-resistant CLL. While omics data indicate proteasome pathway in ibrutinib-resistance, in preclinical treatment mostly carfilzomib (CFZ) was tested. Hence it is more appropriate to replace proteasome inhibitor by carfilzomib throughout the manuscript except for Figure 7K.

Response: We would like to thank the referee for the thorough review and helpful comments. We have changed the wording throughout the text to make clear that we used mainly carfilzomib in our study.

2. Authors have added seven additional patient samples (Yellow-highlighted in Table 1) to the original three patients. This enhances strength of the manuscript. Clinical features are provided for each patient however it is not uniformly described or presented, and important information is missing for some individuals. For example, VAF value should be added for first four patients. BTK and/or PLCG2 mutation information should be provided for patients 4, 7 and 8. For some genes, chromosomal location is included for some patients, but it is not incorporated for other patients.

Response: We thank the reviewer for this observation. We agree that the information provided was not uniformly presented, so we rearranged the whole table to report relevant and coherent details for all the samples. VAF for the first four patients and BTK/PLCG2 mutations for patients 4, 7 and 8 have been provided, as requested.

3. Is there statistical difference between ibrutinib naïve versus ibrutinib-resistant CLL cells, spleen, and liver that are treated with carfilzomib in Figure 4 C, D, E, F? That is to compare data represented by blue solid squares and brown solid squares in panels C-F for Figure 4. If it is not significantly different, then these data would suggest that ibrutinib and carfilzomib sequential combination is effective in ibrutinib-sensitive as well as ibrutinib-resistant CLL.

Response: There is no statistically significant difference between the data of ibrutinib naïve versus ibrutinib-resistant samples. Therefore, we agree with the referee that the sequential treatment of ibrutinib and carfilzomib works both for ibrutinib-sensitive and -resistant CLL. We have adapted the text of the manuscript to better highlight this observation.

4. For SF6A, is ibrutinib (I) treatment alone or carfilzomib (CFZ) treatment alone different than I+CFZ? A p value and statistical analysis should be included.

Response: We would like to thank the referee for pointing this out. We included the statistics of more comparisons. Information on the statistical analyses is provided in the figure legends. For readability

purposes, we decided to include in the figure only the significant comparisons. This information has been included in the figure legend.

5. References 15 and 26 are same (Lamothe et al. Blood 2015). Based on the text of the manuscript it appears that authors are referring to single agent carfilzomib in CLL (one reference) and carfilzomib addition after ibrutinib therapy in CLL (another reference). Authors should appropriately cite the following two references

Single agent CFZ in CLL.

Lamothe B, Wierda WG, Keating MJ, et al. Carfilzomib triggers cell death in chronic lymphocytic leukemia by inducing proapoptotic and endoplasmic reticulum stress responses. Clin Cancer Res 22(18):4712-26, 2016.

Ibrutinib and CFZ combination in CLL.

Lamothe B, Cervantes-Gomez F, Sivina M, et al. Proteasome inhibitor carfilzomib complements ibrutinib's action in chronic lymphocytic leukemia. Blood 125(2):407-10, 2015.

Response: Thanks for pointing this out. We corrected the citing of these two studies.

Minor comments

1. References 7 and 41 are same. Reference 41 needs to be deleted.

2. References 16 and 30 are same. Reference 30 needs to be deleted.

3. References 29 and 51 are same. Reference 51 needs to be deleted.

4. Reference 45, 46, 49, 64 and many others are incomplete. Authors should check all references for completeness and duplication.

Response: We have checked for duplications and incomplete references and adjusted the manuscript.

5. SF4 title should be changed from 'Proteasome inhibitor treatment on '...to 'CFZ treatment on'.

Response: The title has been changed.

6. For SF5B, Naïve CZF should be Naïve CFZ.

Response: We thank the reviewer for this point, we revised the text.

RESPONSE TO REVIEWERS' COMMENTS (R3)

Reviewer #3 (Remarks to the Author):

The comments are not numbered in my prior critique or in the authors response to reviewers, so I'll count the response # to ID it and give a summary sentence and the page # from the response to reviewers to clarify what I'm referring to.

As before, my concerns center on the multi omic analysis and its applicability to routine BTK resistance, and not on the attempts to overcome this with proteasome inhibition.

Response: We would like to thank you for revising again our new version of the manuscript and for your helpful and constructive comments, that help us to improve the presentation and description of our data.

1st Response. Regarding model suitability for CLL. The authors agree that their model is "therefore resembling an advanced or aggressive form of CLL more similar to accelerated or transformed disease". This affirms my concern that the findings, while intriguing, may have little relevance to BTK resistance in non-advanced forms of CLL. This cannot be remedied. It doesn't make the findings incorrect, but raises concerns regarding their relevance to the typical BTK resistance they are trying to study. As they note the "frequency of this type of disease may be low, but represents patients with the highest clinical need." I concur, that it reflects a rare but difficult patient population. So, the findings may have value, but to a smaller subset of patients than implied, and not to general BTKi resistance.

Response: We agree with the referee, that our data are of most relevance for patients that develop resistance to ibrutinib due to accelerated and maybe transformed disease. And we are happy to hear that the referee finds our findings as correct and likely of value for this subset of cases. And even though these are not the majority of resistant cases, they are the ones that likely do not respond to other treatment options currently available. That is why our findings that carfilzomib has a high likelihood to be effective in these cases is important to consider for improving therapy options for these hard-to-treat patients. We want to make the point that we did not try to study typical BTK resistance as assumed by the referee. The resistance mechanism of patients who develop ibrutinib resistance late seems to be a clonal selection of cells that have mutations in BTK or PLCG2 and these patients can now be successfully treated with venetoclax. As we did not find such mutations in our mouse model and the resistance also developed much faster, it was clear to us that our model does not mimic this group of patients, but rather the before mentioned small subset of cases with accelerated and transformed disease, which we found highly interesting as the resistance mechanisms was completely unclear for those cases.

2nd Response. Regarding comparing 3 vs 6 week time points and "time in mouse" changes. The authors have adequately clarified that they are using the peripheral blood CLL cell count to align the samples that they are comparing, and the BTK treated mice have a lag before outgrowth. I can accept this. I think they need to add more text about this model and this why this endpoint was selected for disease state alignment to the manuscript and most readers will not be intimately familiar with this model.

Response: We were happy to hear that we could clarify this open question. To improve the description of our mouse model and the selection of the time points of sampling, we have added more text on

pages 6-7 and a Figure S2B to show that the late samples in the veh and ibr groups were similar in CLL cell counts in the spleen.

3rdnd Response. Regarding lack of phosphoproteomics. Adequately addressed in revised manuscript.

Response: We are glad that we could address this criticism adequately.

4th Response & 5th Response. Regarding former fig 2B interpretation and the different protein patterns of the different condition, and the lack of mRNA and protein correlation. The authors responded that the prior manuscript referred to the wrong figure, and redirect me to S2D. The text here and elsewhere makes the point that the I-E cells differ from the I-L cells, in that the I-E are not proliferative, and the I-L, having escaped BTKi suppression are now proliferative again. In the prior review I was concerned that we couldn't tell what changes were truly related to resistance vs. time in mouse and other effects. The response shifts my concern to be that now we cannot tell what changes are related to suppressed proliferation (I-E) and which to actual resistance in the I-L cells. The data (fig S2D for RNA-GEP and Response page 7 for protein) shows that the I-E cells, which decrease their proliferation rate in their initial response to BTKi, clearly move away from the V-E cells and that the V-L cells drift a bit from the V-E over time (a time in culture effect for the control cells), and that the I-L cells, once they have regained proliferation, move to a PCA space close to that of V-L cells. So, to me this says that the majority of the detected differences are reflective of proliferation state, and possibly buried within this are changes related to resistance. The analysis as performed did not try to separate what are proliferation state related changes, which are likely unrelated to BTKi resistance, and which DEG or protein are likely related to resistance. This would be a lot of work to perform, but I think that is crucial to teasing out the changes that they want to identify. Perhaps there are changes in I-E vs V-E that are also present in I-L vs V-L, as opposed to those common to I-L and V-L, and maybe these represent resistance, rather than proliferation?

I did not understand what point the authors were trying to make with the up and down regulated transcript/protein correlation plots on page 6 of the review when they say that this suggests "a prominent post transcriptional impact on deregulated transcripts". To me, there is no significant correlation in either.

Response: We are happy to hear that we could clarify the referee's concern regarding the "time in mouse" effect. And we agree that we have a major difference in the proteome of cells depending on whether they are cycling or not. This is mainly reflected in the separate clustering of I-early cells compared to the other 3 groups as shown in Fig. 2A. To decipher differences that are independent from the proliferative state of the cells, we compared the proteome of I-late and V-late cells (Fig. 3B), which are both cycling at a similar degree (Fig. 1D). In addition, we specifically searched for groups of proteins with a similar pattern of expression across the four sample groups (Fig. 3D). We identified here proteins that were differently expressed in I-late cells and all 3 other groups (cluster 1), only V-late and I-early cells (cluster 8 and 12), or only I-early cells (cluster 5). This later group of proteins might be mainly changed due to the proliferation state of cells as also evident by the GO term "Cell cycle". To look at these patterns helped us to discern differences that are mainly driven by the cell cycle activity versus those that are likely associated to resistance. Both the results shown in Fig. 3B and 3D, pointed towards a dysregulation of the proteasome activity in cells that are resistant to ibrutinib.

Concerning the correlation plots shown on page 6 of our last response letter, this shows that there is no correlation between the transcript level and protein level of all up- or down-regulated transcripts in the I-late group. That means that the observed difference in mRNA levels is not translated to a difference in the corresponding proteins. A possible explanation for that is the existence of post-transcriptional regulation, e.g. impacting on protein stability or degradation. Such observations are well-known from other published studies and they highlight the necessity to quantify proteins instead of transcripts, as proteins rather than transcripts are the mediators of biological processes.

6th Response. Regarding PCA plots for protein (Response page 7 lower). Generally these responses raise the same concern that most of what we are being shown are proliferation related changes, and that they have not teased out what are related to actual resistance.

Response: We are not completely sure to which part the referee is referring to with this concern. On page 7 of our last response letter, we show 2 PCA plots of transcripts using either principal components 1 and 2 (left) or principal components 2 and 3 (right). Both plots show that the 4 sample groups cluster as expected, showing that they transcriptionally distinct from each other. We use this information as a quality check for the robustness of the transcriptome data.

On the lower part of this page, we describe how we normalized the proteome data via Limma-based batch correction algorithm to overcome batch effects that are regularly obtained when analyzing samples independently via mass spectrometry. After this batch correction, the obtained data shows that all 4 sample groups cluster correctly both in uMAP and PCA visualization (lower plots left and right). In both parts of our response letter, we see no evidence for the statement of the referee, that we are mostly showing proliferation-related changes in the proteome.

7th Response. Regarding total proteome vs phosphoproteome. (Response bottom 8/top page 9) . Adequately addressed in revised manuscript.

Response: We are happy that we were able to adequately address this criticism.

8th Response. Regarding figure 5A and rearranging it. page 9. I think the “guided” heatmap (middle figure top of page 10) better shows both the variability in the 3 naïve UT and 3 naïve CZL samples, and better shows that Res UT and Res CZL are heterogeneous within each group, and share many commonalities across groups, than the prior and current figure 5A. Is there overlap between the proteins in the yellow, magenta and green boxes in the right most figure on page 10, and the proteins that are appearing in the volcano plot fig 5B ? Minor comment: Fig5B volcano plot has 5 colors, but the description for red is not included in the legend.

Response: Even though we agree with the referee that the “guided” heatmap has advantages in its display, a very important information, the dendrogram on top of the plot is missing in this version. This information shows us how the samples cluster in an unsupervised analysis. Therefore, we suggest keeping the unsupervised version of the heatmap as Figure 5A. Concerning the overlap of proteins, candidates with higher abundance in untreated conditions (i.e. Cacybp, Dusp3, Uchl3) lie in the yellow or green (i.e Prdx3, Bcap31) box, while the magenta box contains proteins showing lower expression in untreated conditions (i.e. Sod1, and Serp1). As shown in the heatmap below, significantly deregulated

proteins (t-test statistics-based volcano plot representation reported in Figure 5B of carfilzomib-treated cells vs untreated) span different levels of protein abundance.

We have added the description for the proteins labelled in red to the legend of Figure 5B. Thanks for pointing this out.

9th Response. Regarding figure 6B. page 10. Corrected in revised manuscript.

10th Response. Regarding I-sensitive comparators. page 10. As expected, this is not a sample they can access, so this cannot be further addressed. No remaining issue.

11th Response. Regarding discussion. page 11. Corrected in revised manuscript.

Minor comments were addressed.

Response: We are glad that the points above were all addressed with the last revision of our manuscript.

Reviewer #4 (Remarks to the Author):

Authors have addressed my prior comments.

Response: Thanks for the second revision of our manuscript. We are glad that all your concerns are resolved in the new version of the paper.

B